# LEARNING STOCHASTIC DYNAMICS FROM SNAPSHOTS THROUGH REGULARIZED UNBALANCED OPTIMAL TRANSPORT

**Zhenyi Zhang**[1], **Tiejun Li**[1,2,*], **and  Peijie Zhou** [2,3,4,5,*]

[1]LMAM and School of Mathematical Sciences, Peking University.
[2]Center for Machine Learning Research, Peking University. [3]NELBDA, Peking University.
[4]Center for Quantitative Biology, Peking University. [5]AI for Science Institute, Beijing.
zhenyizhang@stu.pku.edu.cn, {tieli, pjzhou}@pku.edu.cn

## ABSTRACT

Reconstructing dynamics using samples from sparsely time-resolved snapshots is an important problem in both natural sciences and machine learning. Here, we introduce a new deep learning approach for solving regularized unbalanced optimal transport (RUOT) and inferring continuous unbalanced stochastic dynamics from observed snapshots. Based on the RUOT form, our method models these dynamics without requiring prior knowledge of growth and death processes or additional information, allowing them to be learned directly from data. Theoretically, we explore the connections between the RUOT and Schrödinger bridge problem and discuss the key challenges and potential solutions. The effectiveness of our method is demonstrated with a synthetic gene regulatory network, high-dimensional Gaussian Mixture Model, and single-cell RNA-seq data from blood development. Compared with other methods, our approach accurately identifies growth and transition patterns, eliminates false transitions, and constructs the Waddington developmental landscape. Our code is available at: https://github.com/zhenyiizhang/DeepRUOT.

## 1 INTRODUCTION

In machine learning and natural sciences, a key challenge is coupling high-dimensional distributions from observed samples, exemplified by Variational Autoencoders (VAEs) (Kingma & Welling, 2013) which map complex data to simpler latent spaces. It is also important in multi-modal analysis for integrating diverse data types (Lahat et al., 2015), particularly in biology through aligning multi-omics data into unified cellular state representations (Cang & Zhao, 2024; Cao et al., 2022; Demetci et al., 2022; Gao et al., 2024b). Recently, there has been growing interest in understanding the dynamics of how distributions are coupled over time, such as diffusion models (Ho et al., 2020; Sohl-Dickstein et al., 2015) and stochastic differential equations (SDEs) (Song et al., 2021). The task is useful for interpolating paths between arbitrary distributions and learning the underlying dynamics (De Bortoli et al., 2021; Neklyudov et al., 2023; Tong et al., 2024b; Wang et al., 2021).

Due to the destructive nature of technology, the analysis of time series data in single-cell RNA sequencing (scRNA-seq) provides an important application scenario for the high dimensional probability distribution coupling and dynamical inference problem (Bunne et al., 2023b; 2024; Jiang & Wan, 2024; Jiang et al., 2022; Lavenant et al., 2024; Peng et al., 2024; Schiebinger et al., 2019; Sha et al., 2024; Tong et al., 2023; Zhang et al., 2021). Trajectory inference in scRNA-seq data have been extensively studied (Saelens et al., 2019), and optimal transport (OT)-based methods has emerged as a central tool for datasets with temporal resolution (Bunne et al., 2024; Klein et al., 2023a;b; Schiebinger et al., 2019). Often, there is a need to learn the continuous dynamics of cells over time and fit the mechanistic model that transforms the initial cell distributions into the distributions at later temporal points. This could be solved through the dynamical formulation of OT (Benamou & Brenier, 2000), also known as the B-B form. However, the formulation has not fully taken the

---

*Corresponding author.

stochastic dynamical effects into account, especially the intrinsic noise in gene expression and cell differentiation (Zhou et al., 2021a;b), which is prevalent in biological processes on single-cell level (Elowitz et al., 2002).

By incorporating stochastic dynamics, the Schrödinger bridge (SB) problem aims to identify the most likely stochastic transition path between two arbitrary distributions relative to a reference stochastic process (Léonard, 2014), and has been applied in a wide range of contexts, including scRNA-seq analysis and generative modeling (Liu et al., 2022a; Pariset et al., 2023). Meanwhile, recent **regularized unbalanced optimal transport (RUOT)** offers a promising approach for modeling both stochastic unbalanced continuous dynamics (Baradat & Lavenant, 2021; Buze & Duong, 2023; Chen et al., 2022b; Janati et al., 2020), which can be viewed as an unbalanced relaxation of the dynamic formulation of Schrödinger bridge problem. However, computational methods for learning RUOT or such high-dimensional unbalanced stochastic dynamics from snapshots are relatively lacking, especially when there is no prior knowledge of unbalanced effect.

Here we develop a new deep learning method (DeepRUOT) for learning general RUOT and inferring continuous unbalanced stochastic dynamics from samples based on the derived Fisher regularization form without requiring prior knowledge. We demonstrate the effectiveness of DeepRUOT on both synthetic and real-world datasets. Compared to the common SB method, our approach accurately identifies growth and transition patterns, eliminates false transitions, and constructs the Waddington developmental landscape of scRNA-seq data. Overall, our main contributions can be summarized as follows:

- We reformulate RUOT with a Fisher regularization form and explore the connections between RUOT and unbalanced SB. The formulation transforms the SDE into the ordinary differential equation (ODE), which is computationally more tractable.

- We propose DeepRUOT, the neural network algorithm for learning high dimensional unbalanced stochastic dynamics from snapshots. Through the neural network modeling for growth and death, our framework does not require prior knowledge of these processes.

- We validate the effectiveness of DeepRUOT on both synthetic data and real scRNA-seq datasets, showing its promising performance compared with existing approaches.

## 2 RELATED WORKS

**Deep Learning Solver for Dynamical OT** To tackle the dynamical OT (i.e. B-B form) in high dimensions, many methods (Albergo & Vanden-Eijnden, 2023; Cheng et al., 2024a;b; Chow et al., 2020; Gao et al., 2024a; Huguet et al., 2022; Jiao et al., 2024b; Jin et al., 2024; Lipman et al., 2023; Liu et al., 2021; 2022b; 2023b; Pooladian et al., 2024; Ruthotto et al., 2020; Tong et al., 2020; 2024a; Wan et al., 2023; Wu et al., 2023; Zhang et al., 2024a) have been developed based on continuous normalizing flow and neural ODE formulation either in original or latent space. To account for sink and source terms in unnormalized distributions, (Peng et al., 2024; Sha et al., 2024; Tong et al., 2023) formulated the neural-network based solver in the unbalanced dynamical OT setup. However, the appropriate formulation along with an effective deep learning solver to **simultaneously account for unbalanced term and stochastic effects** in dynamical OT remains largely lacking.

**Computational Methods for Schrödinger Bridge Problem** Many methods have recently been developed to solve the static SB problem (Chizat et al., 2022; De Bortoli et al., 2021; Gu et al., 2024; Lavenant et al., 2024; Liu et al., 2022a; Pariset et al., 2023; Pooladian & Niles-Weed, 2024; Shi et al., 2024; Ventre et al., 2023). To tackle the dynamical Schrödinger Bridge, methods based on neural SDE solver, neural ODE solver with fisher information regularization or flow matching (Albergo et al., 2023; Bunne et al., 2023a; Chen et al., 2022a; Jiao et al., 2024a; Koshizuka & Sato, 2023; Liu et al., 2023a; Maddu et al., 2024; Neklyudov et al., 2023; 2024; Tong et al., 2024b; Wang et al., 2021; Zhang et al., 2024b; Zhou et al., 2024a;b) have been proposed. However, these methods either fail to account for **unnormalized distributions** resulting from cell growth and death, or require **prior knowledge** of the processes (growth/death rate) (Chizat et al., 2022; Lavenant et al., 2024; Pariset et al., 2023; Schiebinger et al., 2019) or additional information (e.g.cell lineage) (Ventre et al., 2023).

**Study of RUOT**   The RUOT has recently formulated (Chen et al., 2022b), also known as unbalanced Schrödinger Bridge. The existing studies are mostly focused on **the theoretical side**. For instance, (Janati et al., 2020) derived a closed-form formula for entropic OT between unbalanced Gaussian measures. (Baradat & Lavenant, 2021) investigates relations between RUOT with branching Schrödinger bridge. (Buze & Duong, 2023) investigates different formulations of the problem.

## 3   PRELIMINARIES AND BACKGROUNDS

In this section, we provide an overview of stochastic effects and unbalanced forms within the dynamical framework. Specifically, considering only stochastic effects leads to the Schrödinger bridge problem (Section 3.1), whereas addressing solely the unbalanced aspect results in unbalanced dynamic optimal transport (Section 3.2). By integrating these two perspectives, we motivate the formulation of the Regularized Unbalanced Optimal Transport (RUOT) framework.

### 3.1   STOCHASTIC EFFECT: SCHRÖDINGER BRIDGE PROBLEM

The Schrödinger bridge problem aims to identify the most likely evolution between a given initial distribution $\nu_0$ and a terminal distribution $\nu_1$ (assumed to have density in this paper), relative to a given reference stochastic process. Formally, this problem can be formulated as the minimization of the Kullback-Leibler (KL) divergence in the optimal control perspective (Dai Pra, 1991) as below:

$$\min_{\mu_0^{\boldsymbol{X}}=\nu_0,\ \mu_1^{\boldsymbol{X}}=\nu_1} \mathcal{D}_{\mathrm{KL}}\left(\mu_{[0,1]}^{\boldsymbol{X}}\|\mu_{[0,1]}^{\boldsymbol{Y}}\right), \tag{1}$$

where $\mu_{[0,1]}^{\boldsymbol{X}}$ denotes the probability measure induced by stochastic process $\boldsymbol{X}_t$ $(0 \leq t \leq 1)$ defined on the space of all continuous paths $C([0,1],\mathbb{R}^d)$, with the distribution of $\boldsymbol{X}_t$ at given time $t$ characterized by the measure $\mu_t^{\boldsymbol{X}}$ with density function $p(\boldsymbol{x},t)$. In this work, we consider $\boldsymbol{X}_t$ the stochastic process characterized by the following stochastic differential equation (SDE):

$$\mathrm{d}\boldsymbol{X}_t = \boldsymbol{b}\left(\boldsymbol{X}_t,t\right)\mathrm{d}t + \boldsymbol{\sigma}\left(\boldsymbol{X}_t,t\right)\mathrm{d}\boldsymbol{W}_t, \tag{2}$$

and the reference measure $\mu_{[0,1]}^{\boldsymbol{Y}}$ is chosen as the probability measure induced by the process $\mathrm{d}\boldsymbol{Y}_t = \boldsymbol{\sigma}(\boldsymbol{Y}_t,t)\mathrm{d}\boldsymbol{W}_t$, where $\boldsymbol{W}_t \in \mathbb{R}^d$ is the standard multidimensional Brownian motion defined on a probability space $(\Omega,\mathcal{F},\mathbb{P})$ with $\mathbb{P}$ the Wiener measure for the coordinate process $\boldsymbol{W}_t(\omega) = \omega(t)$, and $\boldsymbol{\sigma}(\boldsymbol{x},t) \in \mathbb{R}^{d\times d}$ is the diffusion matrix which is typically assumed as bounded, coercive and invertible. Under this formulation, the solution to Eq. (1) is also referred to as the *diffusion Schrödinger bridge*. The diffusion Schrödinger bridge also possesses a dynamic formulation, which can be formally stated as the following theorem.

**Theorem 3.1.** *Consider the diffusion Schrödinger bridge problem (1) where $\mu_{[0,1]}^{\boldsymbol{Y}}$ the reference measure induced by $\mathrm{d}\boldsymbol{Y}_t = \boldsymbol{\sigma}(\boldsymbol{Y}_t,t)\mathrm{d}\boldsymbol{W}_t$. Then (1) is equivalent to*

$$\inf_{(p,\boldsymbol{b})} \int_0^1 \int_{\mathbb{R}^d} \left[\frac{1}{2}\boldsymbol{b}^{\boldsymbol{T}}(\boldsymbol{x},t)\boldsymbol{a}^{-1}(\boldsymbol{x},t)\boldsymbol{b}(\boldsymbol{x},t)\right] p(\boldsymbol{x},t)\mathrm{d}\boldsymbol{x}\mathrm{d}t, \tag{3}$$

*where the infimum is taken over all function pairs $(p,\boldsymbol{b})$ such that $p(\cdot,0)=\nu_0$, $p(\cdot,1)=\nu_1$, $p(\boldsymbol{x},t)$ absolutely continuous, and*

$$\partial_t p(\boldsymbol{x},t) = -\nabla_{\boldsymbol{x}} \cdot (p(\boldsymbol{x},t)\boldsymbol{b}(\boldsymbol{x},t)) + \frac{1}{2}\nabla_{\boldsymbol{x}}^2 : (\boldsymbol{a}(\boldsymbol{x},t)p(\boldsymbol{x},t)), \tag{4}$$

*where $\nabla_{\boldsymbol{x}}^2 : (\boldsymbol{a}(\boldsymbol{x},t)p(\boldsymbol{x},t)) = \sum_{ij} \partial_{ij}(a_{ij}p)$ and $\boldsymbol{a}(\boldsymbol{x},t) = \boldsymbol{\sigma}(\boldsymbol{x},t)\boldsymbol{\sigma}^T(\boldsymbol{x},t)$, coupled with vanishing boundary condition:* $\lim_{|\boldsymbol{x}|\to\infty} p(\boldsymbol{x},t) = 0$.

This theorem or its variants have been stated and proven in various forms, such as in (Chen et al., 2016; Dai Pra, 1991; Gentil et al., 2017). Here, we provide a simple direct proof for illustration in Appendix D.1. We note that in this dynamic formulation, it accounts for the stochastic aspects but does not incorporate unbalanced effects. As we will discuss below, RUOT can be motivated by incorporating unbalanced effects in the dynamic formulation of SB.

### 3.2 Unbalanced Effect: Unbalanced Dynamic Optimal Transport

The optimal transport problem has been extensively studied in various fields. Given two probability distributions $\boldsymbol{\alpha} \in \mathbb{R}_+^n$ and $\boldsymbol{\beta} \in \mathbb{R}_+^m$, its primary goal is to find the optimal coupling $\boldsymbol{\pi} \in \mathbb{R}_+^{n \times m}$ to transport a given distribution of mass (or resources) from one location to another while minimizing the cost associated with the transportation. The static optimal transport problem can be mathematically formulated as $\min_{\boldsymbol{\pi} \in \boldsymbol{\Pi}(\boldsymbol{\alpha}, \boldsymbol{\beta})} \langle \boldsymbol{\pi}, \boldsymbol{c} \rangle$, where $\boldsymbol{\Pi}(\boldsymbol{\alpha}, \boldsymbol{\beta}) = \left\{ \boldsymbol{\pi} \in \mathbb{R}_+^{n \times m} : \boldsymbol{\pi} \mathbf{1}_m = \boldsymbol{\alpha}, \boldsymbol{\pi}^T \mathbf{1}_n = \boldsymbol{\beta}, \boldsymbol{\pi} \geq 0 \right\}$. The cost matrix $\boldsymbol{c} \in \mathbb{R}^{n \times m}$ defines the transportation cost between each pair of points, where $c_{ij} := c(\boldsymbol{x}_i, \boldsymbol{y}_j)$ represents the cost of moving a unit mass from point $\boldsymbol{x}_i$ to point $\boldsymbol{y}_j$. We refer to (Peyré et al., 2019) for more details. Next we briefly state some well-known results on the dynamical formulation of the OT.

**Dynamical Optimal Transport** The formulation is also known as the Benamou-Brenier formulation (Benamou & Brenier, 2000), which can be stated as follows:

$$\mathcal{W}(\nu_0, \nu_1) = \inf_{(p(\boldsymbol{x}, t), \boldsymbol{b}(\boldsymbol{x}, t))} \int_0^1 \int_{\mathbb{R}^d} \frac{1}{2} \|\boldsymbol{b}(\boldsymbol{x}, t)\|_2^2 p(\boldsymbol{x}, t) \mathrm{d}\boldsymbol{x} \mathrm{d}t,$$
$$\text{s.t. } \partial_t p + \nabla \cdot (\boldsymbol{b}(\boldsymbol{x}, t) p) = 0, \ p|_{t=0} = \nu_0, p|_{t=1} = \nu_1.$$

Compared to SB, in OT the distributions are connected by deterministic transport equation instead of diffusion. It can be shown that this dynamical formulation corresponds to a static Kantorovich's optimal transport problem with cost function $c(\boldsymbol{x}, \boldsymbol{y}) = \|\boldsymbol{x} - \boldsymbol{y}\|_2^2$.

**Regularized Optimal Transport** The regularized optimal transport is defined by a dynamical form of the general Schrödinger bridge by taking diffusion rate as constant and scaled by $\sigma^2$ in (3):

$$\mathcal{W}(\nu_0, \nu_1) = \inf_{(p(\boldsymbol{x}, t), \boldsymbol{b}(\boldsymbol{x}, t))} \int_0^1 \int_{\mathbb{R}^d} \frac{1}{2} \|\boldsymbol{b}(\boldsymbol{x}, t)\|_2^2 p(\boldsymbol{x}, t) \mathrm{d}\boldsymbol{x} \mathrm{d}t,$$
$$\text{s.t. } \partial_t p + \nabla \cdot (\boldsymbol{b}(\boldsymbol{x}, t) p) - \frac{\sigma^2}{2} \Delta p = 0, \ p|_{t=0} = \nu_0, p|_{t=1} = \nu_1,$$

It can be demonstrated that as $\sigma^2$ approaches zero, the solution to this problem converges to that of the Benamou-Brenier problem (Mikami & Thieullen, 2008). And it is equivalent to the SB problem (Baradat & Lavenant, 2021; Gentil et al., 2017; Léonard, 2014).

**Unbalanced Dynamic Optimal Transport** To account for unnormalized marginal distributions and effects such as growth and death, an unbalanced optimal transport problem with Wasserstein–Fisher–Rao (WFR) metric has been proposed (Chizat et al., 2018a;b) or its extensions (Gangbo et al., 2019). Here we adopt the WFR unbalanced optimal transport:

$$\mathcal{W}(\nu_0, \nu_1) = \inf_{(p(\boldsymbol{x}, t), \boldsymbol{b}(\boldsymbol{x}, t), g(\boldsymbol{x}, t))} \int_0^1 \int_{\mathbb{R}^d} \left( \frac{1}{2} \|\boldsymbol{b}(\boldsymbol{x}, t)\|_2^2 + \alpha |g(\boldsymbol{x}, t)|_2^2 \right) p(\boldsymbol{x}, t) \mathrm{d}\boldsymbol{x} \mathrm{d}t,$$
$$\text{s.t. } \partial_t p + \nabla \cdot (\boldsymbol{b}(\boldsymbol{x}, t) p) = g(\boldsymbol{x}, t) p, \ p|_{t=0} = \nu_0, p|_{t=1} = \nu_1.$$

Here $g(\boldsymbol{x}, t)$ is a scalar function that denotes the growth or death rate of particles at the state $\boldsymbol{x}$ and time $t$, and is also optimized in the total energy term. $\alpha$ is the hyperparameter of weight. We should also note that in this case, $\nu_0$ and $\nu_1$ are not necessarily the normalized probability densities, but are generally densities of masses.

## 4 Regularized Unbalanced Optimal Transport

To simplify the notation and illustrate a commonly used setting, we take a special case considering $\boldsymbol{a}(\boldsymbol{x}, t) = \sigma^2(t) \boldsymbol{I}$, and the general case is left to Appendix D.4 for further discussions. Inspired by unbalanced dynamic optimal transport, the dynamical formulation Theorem 3.1 suggests a natural approach to relaxing the mass conservation constraint by introducing a growth/death term $gp$ in Eq. (4) as $\partial_t p = -\nabla_{\boldsymbol{x}} \cdot (p \boldsymbol{b}) + \frac{1}{2} \nabla_{\boldsymbol{x}}^2 : (\sigma^2(t) \boldsymbol{I} p) + gp$. Meanwhile, we also define a loss functional in Eq. (5) which incorporates the growth penalization and Wasserstein metric considered in Eq. (3) (with a rescaling by $\sigma^2(t)$). We refer to this formulation as the *regularized unbalanced optimal transport*.

**Definition 4.1** (Regularized unbalanced optimal transport). *Consider*

$$\inf_{(p,\boldsymbol{b},g)} \int_0^1 \int_{\mathbb{R}^d} \frac{1}{2} \|\boldsymbol{b}(\boldsymbol{x},t)\|_2^2 \, p(\boldsymbol{x},t)\mathrm{d}\boldsymbol{x}\mathrm{d}t + \int_0^1 \int_{\mathbb{R}^d} \alpha \Psi\left(g(\boldsymbol{x},t)\right) p(\boldsymbol{x},t)\mathrm{d}\boldsymbol{x}\mathrm{d}t, \tag{5}$$

*where* $\Psi : \mathbb{R} \to [0, +\infty]$ *corresponds to the growth penalty function, the infimum is taken over all pairs* $(p, \boldsymbol{b}, g)$ *such that* $p(\cdot, 0) = \nu_0, p(\cdot, 1) = \nu_1, p(\boldsymbol{x}, t)$ *absolutely continuous, and*

$$\partial_t p = -\nabla_{\boldsymbol{x}} \cdot (p\boldsymbol{b}) + \frac{1}{2}\nabla_{\boldsymbol{x}}^2 : \left(\sigma^2(t)\boldsymbol{I}p\right) + gp \tag{6}$$

*with vanishing boundary condition:* $\lim_{|\boldsymbol{x}|\to\infty} p(\boldsymbol{x}, t) = 0.$

Note that in the definition if $\Psi(g) = +\infty$ unless $g = 0$ and $\Psi(0) = 0$, then it implies $g(\boldsymbol{x}, t) = 0$ and the RUOT degenerates to the *regularized optimal transport* problem. If $\sigma(t) \to 0$ and $\Psi\left(g(\boldsymbol{x}, t)\right) = |g(\boldsymbol{x}, t)|^2$, this degenerates to the *unbalanced dynamic optimal transport* with WFR metrics. Meanwhile when $\sigma(t)$ is constant, it coincides with the definition of RUOT provided in (Baradat & Lavenant, 2021). Then, we can reformulate Definition 4.1 with the following Fisher information regularization.

**Theorem 4.1.** *The regularized unbalanced optimal transport problem* (5) *is equivalent to*

$$\inf_{(p,\boldsymbol{v},g)} \int_0^1 \int_{\mathbb{R}^d} \left[\frac{1}{2} \|\boldsymbol{v}\|_2^2 + \frac{\sigma^4(t)}{8} \|\nabla_{\boldsymbol{x}} \log p\|_2^2 - \frac{\sigma^2(t)}{2}\left(1 + \log p\right)g - \frac{1}{2}\frac{\mathrm{d}\sigma^2(t)}{\mathrm{d}t}\log p + \alpha\Psi\left(g\right)\right] p\mathrm{d}\boldsymbol{x}\mathrm{d}t, \tag{7}$$

*where the infimum is taken over all triplets* $(p, \boldsymbol{v}, g)$ *such that* $p(\cdot, 0) = \nu_0, p(\cdot, 1) = \nu_1, p(\boldsymbol{x}, t)$ *absolutely continuous, and*

$$\partial_t p = -\nabla_{\boldsymbol{x}} \cdot (p\boldsymbol{v}(\boldsymbol{x}, t)) + g(\boldsymbol{x}, t)p \tag{8}$$

*with vanishing boundary condition:* $\lim_{|\boldsymbol{x}|\to\infty} p(\boldsymbol{x}, t) = 0.$

Here $\boldsymbol{v}(\boldsymbol{x}, t)$ represents a new vector field. The proof is left to Appendix D.2. In (Baradat & Lavenant, 2021), they proposed another Fisher regularization form of RUOT, expressed in the formula Eq. (15). In their formulation, computing the cross term $\langle\nabla_{\boldsymbol{x}} \log p, \sigma^2(t)\boldsymbol{v}\rangle$ required calculating the derivative of the function $\log p$ with respect to $\boldsymbol{x}$ and performing vector multiplications with $\boldsymbol{v}$. The formulation here is equivalent to theirs but more computationally tractable, as we avoid differentiation and vector multiplication in our cross-term.

**Remark 4.1.** *When* $g = 0$, $\Psi(0) = 0$ *and* $\sigma(t)$ *is constant, then Eq.* (7) *is the same as the dynamic entropy-regularized optimal transport form as discussed in (Bunne et al., 2023a; Gentil et al., 2017; Léger & Li, 2021; Li et al., 2020; Neklyudov et al., 2024; Pooladian & Niles-Weed, 2024).*

**Remark 4.2.** *The term* $\mathcal{I}(p) = \int_{\mathbb{R}^d} \|\nabla_{\boldsymbol{x}} \log p(\boldsymbol{x}, t)\|_2^2 p(\boldsymbol{x}, t)\mathrm{d}\boldsymbol{x}$ *in Eq.* (7) *is referred to as the* Fisher information. *Notably, when considering growth/death factors, Eq.* (7) *includes not only the Fisher-Rao metric but also an additional cross-term* $\int_{\mathbb{R}^d} -\frac{1}{2}\sigma^2(t)\left(1 + \log p(\boldsymbol{x}, t)\right)g(\boldsymbol{x}, t)p(\boldsymbol{x}, t)\mathrm{d}\boldsymbol{x}.$

**Remark 4.3.** *From the Fokker-Plank equation and the proof of Theorem 4.1, the original SDE* $\mathrm{d}\boldsymbol{X}_t = \left(\boldsymbol{b}\left(\boldsymbol{X}_t, t\right)\right)\mathrm{d}t + \sigma\left(t\right)\mathrm{d}\boldsymbol{W}_t$ *can be transformed into the probability flow ODE*

$$\mathrm{d}\boldsymbol{X}_t = \underbrace{\left(\boldsymbol{b}\left(\boldsymbol{X}_t, t\right) - \frac{1}{2}\sigma^2(t)\nabla_{\boldsymbol{x}} \log p(\boldsymbol{X}_t, t)\right)}_{\boldsymbol{v}(\boldsymbol{X}_t, t)}\mathrm{d}t.$$

*Conversely, if the probability flow ODE's drift* $\boldsymbol{v}(\boldsymbol{x}, t)$, *the diffusion rate* $\sigma(t)$ *and the* score function $\nabla_{\boldsymbol{x}} \log p(\boldsymbol{x}, t)$ *are known, then the the drift term* $\boldsymbol{b}(\boldsymbol{x}, t)$ *of the SDE can be determined by* $\boldsymbol{b}(\boldsymbol{x}, t) = \boldsymbol{v}(\boldsymbol{x}, t) + \frac{1}{2}\sigma^2(t)\nabla_{\boldsymbol{x}} \log p(\boldsymbol{x}, t).$ *Thus, to specify an SDE is equivalent to specifying the probability flow ODE and the corresponding score function* $\nabla_{\boldsymbol{x}} \log p(\boldsymbol{x}, t)$ *(Tong et al., 2024b).*

In (Buze & Duong, 2023), the authors defined a RUOT problem with nonlinear Fokker-Planck equation constraints. In Appendix D.3, we show that the proposed form is indeed consistent with the RUOT defined here when $\sigma(t)$ is constant and $\Psi\left(g(\boldsymbol{x}, t)\right)$ takes the quadratic form (i.e., $\Psi\left(g(\boldsymbol{x}, t)\right) = |g(\boldsymbol{x}, t)|_2^2$). We then explore the connections between RUOT and SB problem in Appendix E.

## 5    LEARNING RUOT THROUGH NEURAL NETWORKS

Given unnormalized distributions at $T$ discrete time points, $\boldsymbol{X}_i \sim \mu_i$ for fixed timepoints $i \in \{0, \ldots, T-1\}$, we aim to learn continuous stochastic dynamics satisfying the RUOT from data. Similarly, to simplify the exposition and illustrate a commonly used case, we consider Definition 4.1 with the stochastic dynamics $\mathrm{d}\boldsymbol{X}_t = \boldsymbol{b}\left(\boldsymbol{X}_t, t\right)\mathrm{d}t + \sigma\left(t\right)\mathrm{d}\boldsymbol{W}_t$. As previously discussed, we approach this by transforming the problem into learning the drift $\boldsymbol{v}(\boldsymbol{x}, t)$ of the probability ODE and its score function $\frac{1}{2}\sigma^2(t)\nabla_{\boldsymbol{x}}\log p(\boldsymbol{x}, t)$ in Theorem 4.1. We parameterize $\boldsymbol{v}(\boldsymbol{x}, t)$, $g(\boldsymbol{x}, t)$, and $\frac{1}{2}\sigma^2(t)\log p(\boldsymbol{x}, t)$ using neural networks $\boldsymbol{v}_\theta$, $g_\theta$ and $s_\theta$ respectively (Fig. 1). To solve Theorem 4.1, the overall loss is composed of energy loss, reconstruction loss, and the Fokker-Planck constraint:

$$\mathcal{L} = \mathcal{L}_{\text{Energy}} + \lambda_r \mathcal{L}_{\text{Recons}} + \lambda_f \mathcal{L}_{\text{FP}}. \tag{9}$$

The $\mathcal{L}_{\text{Energy}}$ loss promotes the least action of kinetic energy Eq. (7). The reconstruction loss $\mathcal{L}_{\text{Recons}}$ promotes the dynamics to match data distribution at later time point (i.e. $p(\cdot, 1) = \nu_1$), and the $\mathcal{L}_{\text{FP}}$ promotes the three parameterized neural network to satisfy Fokker-Planck constraints Eq. (8).

### 5.1    ENERGY LOSS

To compute the integral in Eq. (7), the direct calculation is infeasible due to the high dimensionality. Thus, we need to transform it into an equivalent form that can be evaluated using Monte Carlo methods. Adopting the approach in (Sha et al., 2024) (see Appendix A.4), Eq. (7) is equivalent to the following form

$$\mathcal{L}_{\text{Energy}} = \mathbb{E}_{\boldsymbol{x_0} \sim p_0} \int_0^T \left[ \frac{1}{2}\|\boldsymbol{v}_\theta\|_2^2 + \frac{1}{2}\|\nabla_{\boldsymbol{x}} s_\theta\|_2^2 - \left(\frac{\sigma^2(t)}{2} + s_\theta\right)g_\theta - \frac{(\sigma^2(t))'}{\sigma^2(t)}s_\theta + \alpha\Psi\left(g_\theta\right) \right] w_\theta(t)\mathrm{d}t, \tag{10}$$

where $w_\theta(t) = e^{\int_0^t g_\theta(\boldsymbol{x}(t), s)\mathrm{d}s}$ and $\boldsymbol{x}(t)$ satisfy $\mathrm{d}\boldsymbol{x}/\mathrm{d}t = \boldsymbol{v}(\boldsymbol{x}, t)\mathrm{d}t$. We compute through Monte Carlo sampling and a Neural ODE solver.

### 5.2    RECONSTRUCTION LOSS

The reconstruction loss aims to match the final distribution in Theorem 4.1 (i.e., $p(\cdot, 1) = \nu_1$). Many works use the balanced optimal transport to evaluate the distance between two distributions. However due to the unnormalized effect, here we aim to use the unbalanced optimal transport instead. To realize this we need to tackle two parts:

$$\mathcal{L}_{\text{Recons}} = \lambda_m \mathcal{L}_{\text{Mass}} + \lambda_d \mathcal{L}_{\text{OT}} \tag{11}$$

where $\lambda_m$ and $\lambda_d$ are hyperparameters. The mass matching loss $\mathcal{L}_{\text{Mass}}$ promotes to align the number of cells. We then normalize the distributions according to the matched masses. The $\mathcal{L}_{\text{OT}}$ uses these weights to perform optimal transport matching.

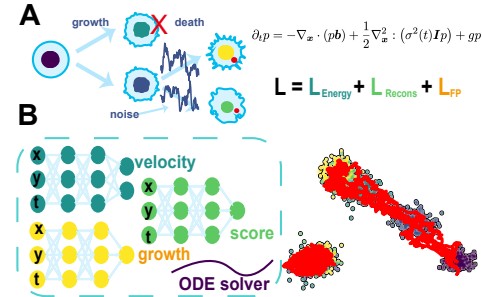

Figure 1: Overview of DeepRUOT.

For $\mathcal{L}_{\text{Mass}}$, we propose a local mass matching strategy. We denote $A_i$ ($i = 0, 1, \cdots T-1$) as the dataset observed at different time points, and define $\phi_\theta^{\boldsymbol{v}} : \mathbb{R}^d \times \mathcal{T} \to \mathbb{R}^{d|\mathcal{T}|}$ as the trajectory mapping function of the Neural ODE $\mathrm{d}\boldsymbol{x}/\mathrm{d}t = \boldsymbol{v}_\theta$, which inputs a given starting point as the initial condition and outputs the observed particle coordinates at $\mathcal{T}$ following ODE dynamics, with $\mathcal{T}$ being a set of time indices. Starting with an initial set $A_0$, the Neural ODE function $\phi_\theta^{\boldsymbol{v}}$ predicts the subsequent sets of data points over the time indices in $\mathcal{T}$. Specifically, the predicted sets $\widehat{A}_1, \ldots, \widehat{A}_{T-1}$ are obtained by applying the Neural ODE function to $A_0$ and the full range of time indices from 1 to $T-1$, i.e., $\widehat{A}_1, \ldots, \widehat{A}_{T-1} = \phi_\theta^{\boldsymbol{v}}\left(A_0, \{1, \ldots, T-1\}\right)$. Similarly, we define $\phi_\theta^g : \mathbb{R} \times \mathcal{T} \to \mathbb{R}^{|\mathcal{T}|}$ as the particle weight mapping function of the Neural ODE $\mathrm{d}\log w_i(t)/\mathrm{d}t = g_\theta(\boldsymbol{x}_i(t), t)$, which inputs a given weight of particle $i$ as the initial condition and outputs the sampled weights at $\mathcal{T}$ following ODE dynamics. For simplicity of notation in algorithm description Algorithm 1, we also use $w(A)$ to denote the set of weights of particles in set

$A$. Assume the sample size $N_0$ at the initial time point, with the relative total masses at subsequent time points denoted as $n_{t_k} = N_k/N_0$ for $k = 0, 1, \cdots, T-1$. Along the trajectory, each sampled particle $i$ has a weight $w_i(t)$ with the initial condition $w_i(0) = 1/N_0$. We then establish a mapping $h_k$ from points at time $t_k$ to the sampled particles (total number $N_0$) predicted at time $t_k$, which links the real data points in $A_k$ to its closest point in sampled particles at $\widehat{A}_k$, i.e., $h_k : A_k \to \widehat{A}_k$, $h_k(\boldsymbol{x}_{t_k}) = \operatorname{argmin}_{\boldsymbol{y} \in \widehat{A}_k} \|\boldsymbol{x}_{t_k} - \boldsymbol{y}\|_2^2$. The mass matching error at time $t_k$ is then defined as $M_k = \sum_{i=1}^{N_0} \left\| w_i(t_k) - \operatorname{card}\left(h_k^{-1}(\boldsymbol{x}_i(t_k))\right) \frac{1}{N_0} \right\|_2^2$. Here $\operatorname{card}(A)$ denotes the cardinality (i.e., number of elements for the finite set) of set $A$. This error metric ensures the matching of local masses. So the $\mathcal{L}_{\text{Mass}} = \sum_{k=1}^{T-1} M_k$ Furthermore, when $M_t = 0$, we have $\sum_{i=1}^{N_0} w_i(t_k) = \sum_{i=1}^{N_0} \operatorname{card}\left(h_t^{-1}(\boldsymbol{x}_i(t_k))\right)/N_0 = N_k/N_0 = n_{t_k}$. So the local matching loss also encourages the matching of total mass, since the left-hand side is the unbiased estimator of $\int_{\mathbb{R}^d} p(\boldsymbol{y}, t_k) d\boldsymbol{y}$. Once we have determined the weights $w_i(t)$, we utilize these weights to perform optimal transport matching of the distributions $\mathcal{L}_{\text{OT}} := \sum_{k=1}^{T-1} \mathcal{W}_2(\hat{\boldsymbol{w}}^k, \boldsymbol{w}(t_k))$, where $\hat{\boldsymbol{w}}^k = (1/N_k, 1/N_k, ..., 1/N_k)$ is the uniform distribution of $A_k$ at time $t$, $\boldsymbol{w}(t_k) = (w_1(t_k), w_2(t_k), ..., w_{N_0}(t_k))/\sum_{i=1}^{N_0} w_i(t_k)$ is the predicted weight distribution of sample particles at time $t_k$ and $\mathcal{W}_2$ represents the Wasserstein distance between $A_k$ and $\widehat{A}_k$ with normalized distribution defined by $\hat{\boldsymbol{w}}^k$ and $\boldsymbol{w}(t_k)$.

## 5.3 FOKKER-PLANCK CONSTRAINT AND TWO STAGE TRAINING

In addition to the energy loss defined in Eq. (10) and the reconstruction loss defined in Eq. (11), it is necessary to incorporate a physics-informed loss (PINN-loss) (Raissi et al., 2019) to constrain the relationships among the three neural networks, i.e., the Fokker-Planck constraint Eq. (8). Here we utilize a Gaussian mixture model to estimate the initial distribution (Appendix A.3), ensuring that it satisfies the initial conditions $p_0$, and the PINN-loss is defined as

$$\mathcal{L}_{\text{FP}} = \|\partial_t p_\theta + \nabla_{\boldsymbol{x}} \cdot (p_\theta \boldsymbol{v}_\theta) - g_\theta p_\theta\| + \lambda_w \|p_\theta(\boldsymbol{x}, 0) - p_0\|, \tag{12}$$

where $p_\theta = \exp \frac{2}{\sigma^2} s_\theta$. Therefore, the total loss function (9) is defined as the weighted sum of the energy, the reconstruction error and the Fokker-Planck loss with the loss weights as hyperparameters. This allows us to develop a neural network algorithm for solving the RUOT problem (Algorithm 1 and Appendix A.1).

We adopt a two-stage training approach to deal with multiple loss terms and stabilize the training process. For the pre-training stage, initially, we use reconstruction loss only to train $\boldsymbol{v}_\theta$ and $g_\theta$, ensuring a required matching as the initial value. Subsequently, we fix $\boldsymbol{v}_\theta$ and $g_\theta$ and employ flow-matching to learn an initial and well-optimized log density function ($s_\theta(\boldsymbol{x}, t)$). Specifically, it is conducted by conditional flow matching (Lipman et al., 2023; Tong et al., 2024a;b) (Appendix A.2). As part of this pre-training stage, we incorporate a hyperparameter scheduling strategy to further enhance stability (Appendix C.2). Based on these warmup steps in the pre-training stage, in the training stage, we use the obtained $\boldsymbol{v}_\theta$, $g_\theta$, and the optimized log density function as the starting point, then get the final result by minimizing the total loss (9). In summary, by integrating all the methodologies, we derive the Algorithm 1 for training the regularized unbalanced optimal transport (Appendices A.2 and C.3). We discuss the loss weighting strategy and settings in Appendix C.2.

# 6 NUMERICAL RESULTS

This section evaluates the DeepRUOT solver's ability to recover growth and transitions accurately and produce realistic stochastic dynamics for constructing a Waddington's developmental landscape. We take $\Psi(g(\boldsymbol{x}, t)) = |g(\boldsymbol{x}, t)|_2^2$ and $\sigma(t)$ is constant in the following computations.

**Synthetic Gene Regulatory Network** Inspired by (Sha et al., 2024), we adopt the same three-gene simulation model (Appendix B.1) to explore the stochastic dynamics of gene regulation, as illustrated in Fig. 2(a). The resulting gene regulatory and cellular dynamics are illustrated in Fig. 2(b), (c), where a quiescent region and an area exhibiting both transition and growth can be observed. We focus on the projection of these dynamics onto the two-dimensional space of $(X_1, X_2)$ since on $X_3$ it remains quiescent. In Fig. 2(d-e), we compare DeepRUOT with the balanced diffusion Schrödinger bridge method as described in (Tong et al., 2024b). We find that neglecting the growth

---

**Algorithm 1** Training Regularized Unbalanced Optimal Transport

---

**Require:** Datasets $A_0, \ldots, A_{T-1}$, batch size $N$, maximum ode iteration $n_{\text{ode}}$, maximum log density iteration $n_{\text{log-density}}$, initialized ODE $\boldsymbol{v_\theta}$, growth $g_\theta$ and log density $s_\theta$
**Ensure:** Trained neural ODE $\boldsymbol{v_\theta}$, growth function $g_\theta$ and log density function $s_\theta$.

1: **Pre-Training Stage:**
2: **for** $i = 1$ to $n_{\text{ode}}$ **do**          ▷ *Distribution Reconstruction training*
3:     **for** $t = 0$ to $T - 2$ **do**
4:        $\widehat{A}_{t+1} \leftarrow \phi_\theta^{\boldsymbol{v}}\left(\widehat{A}_t, t+1\right), w(\widehat{A}_{t+1}) \leftarrow \phi_\theta^g\left(w(\widehat{A}_t), t+1\right).$
5:        $\mathcal{L}_{\text{Recons}} \leftarrow \mathcal{L}_{\text{Recons}} + \lambda_m M_t + \lambda_d \mathcal{W}_2\left(\hat{\boldsymbol{w}}^t, \boldsymbol{w}(t)\right)$ (11), update $\boldsymbol{v_\theta}$ and $g_\theta$ w.r.t. $\mathcal{L}_{\text{Recons}}$ with hyperparameter scheduling (Appendix C.2).
6: **for** $t = 0$ to $T - 2$ **do**
7:     $\widehat{A}_{t+1} \leftarrow \phi_\theta^{\boldsymbol{v}}\left(\widehat{A}_t, t+1\right)$          ▷ *Generating samples from learned $\boldsymbol{v_\theta}$.*
8: **for** $i = 1$ to $n_{\text{log-density}}$ **do**          ▷ *CFM Score matching (Tong et al., 2024b)*
9:     $(\boldsymbol{x}_0, \boldsymbol{x}_1) \sim q(\boldsymbol{x}_0, \boldsymbol{x}_1); \quad t \sim \mathcal{U}(0,1); \quad \boldsymbol{x} \sim p(\boldsymbol{x}, t \mid (\boldsymbol{x}_0, \boldsymbol{x}_1))$ at generated datasets
       $\widehat{A}_0, \cdots, \widehat{A}_{T-1}, \mathcal{L}_{\text{score}} \leftarrow \|\lambda_s(t)\nabla_{\boldsymbol{x}} s_\theta(\boldsymbol{x}, t) + \boldsymbol{\epsilon_1}\|_2^2$ (13) , update $s_\theta$ w.r.t. the loss $\mathcal{L}_{\text{score}}$
10: **Training Stage**:
11: **for** $i = 1$ to $n_{\text{ode}}$ **do**
12:     Estimate the initial distribution through Gaussian Mixture Model (Appendix A.3).
13:     **for** $t = 0$ to $T - 2$ **do**
14:        $\widehat{A}_{t+1} \leftarrow \phi_\theta^{\boldsymbol{v}}\left(\widehat{A}_t, t+1\right), w(\widehat{A}_{t+1}) \leftarrow \phi_\theta^g\left(w(\widehat{A}_t), t+1\right)$
15:        $\mathcal{L}_{\text{Energy}} \leftarrow \mathbb{E}_{\boldsymbol{x}_t \sim p_t} \int_t^{t+1} \left[\frac{1}{2}\|\boldsymbol{v_\theta}\|_2^2 + \frac{1}{2}\|\nabla_{\boldsymbol{x}} s_\theta\|_2^2 - \left(\frac{\sigma^2}{2} + s_\theta\right) g_\theta - \frac{(\sigma^2(t))'}{\sigma^2(t)} s_\theta + \alpha \Psi(g_\theta)\right] w(z)\mathrm{d}z$
16:        $\mathcal{L}_{\text{Recons}} \leftarrow \mathcal{L}_{\text{Recons}} + \lambda_m M_t + \lambda_d \mathcal{W}_2(\hat{\boldsymbol{w}}^t, \boldsymbol{w}(t))$ (11)
17:        $\mathcal{L}_{\text{FP}} \leftarrow \|\partial_t p_\theta + \nabla_{\boldsymbol{x}} \cdot (p_\theta \boldsymbol{v_\theta}(\boldsymbol{x}, t)) - g_\theta(\boldsymbol{x}, t)p_\theta\| + \lambda_w \|p_\theta(\boldsymbol{x}, 0) - p_0\|$ (12)
18:        $\mathcal{L} \leftarrow \mathcal{L}_{\text{Energy}} + \lambda_r \mathcal{L}_{\text{Recons}} + \lambda_f \mathcal{L}_{\text{FP}}$ (9) , update $\boldsymbol{v_\theta}$, $g_\theta$ and $s_\theta$ w.r.t. $\mathcal{L}$

---

in the balanced diffusion Schrödinger bridge leads to a false transition, incorrectly attracting cells in the quiescent state to the transition and growth region. The underlying reason is that the growth factor causes an increase in the number of cells. If the growth factor is ignored, the regions with increasing cell numbers will attract cells to transition into them to maintain balance. Our approach, which explicitly incorporates growth dynamics, indeed eliminates false transitions and yields results consistent with the ground truth dynamics Fig. 2(b). Furthermore, the growth rates inferred by Deep-RUOT (Fig. 2(f)) closely match the ground truth after normalization, demonstrating the robustness and accuracy of the DeepRUOT solver (Fig. 2(c)). In Table 1, we present quantitative metrics ($\mathcal{W}_1$ and $\mathcal{W}_2$, Appendix C.1) to evaluate the performance of our proposed algorithm in comparison with several representative baseline methods (Balanced OT, Balanced SB, Unbalanced OT, Unbalanced Action Matching (AM), Unbalanced SB). The results demonstrate that DeepRUOT is quantitatively more accurate than the competing methods. Furthermore, we observe that our algorithm can benefit from the incorporation of stochasticity, outperforming the solver without diffusion (Unbalanced OT). We include the detail ablation studies in Appendix B.5.

Table 1: Wasserstein distance ($\mathcal{W}_1$ and $\mathcal{W}_2$) of predictions at different points across five runs on gene regulatory data (Appendix C.1). We show the mean value with one standard deviation.

| | $t=1$ | | $t=2$ | | $t=3$ | | $t=4$ | |
| --- | --- | --- | --- | --- | --- | --- | --- | --- |
| Model | $\mathcal{W}_1$ | $\mathcal{W}_2$ | $\mathcal{W}_1$ | $\mathcal{W}_2$ | $\mathcal{W}_1$ | $\mathcal{W}_2$ | $\mathcal{W}_1$ | $\mathcal{W}_2$ |
| MIOFlow (Huguet et al., 2022) | $0.098_{\pm 0.000}$ | $0.113_{\pm 0.000}$ | $0.250_{\pm 0.000}$ | $0.295_{\pm 0.000}$ | $0.421_{\pm 0.000}$ | $0.536_{\pm 0.000}$ | $0.614_{\pm 0.000}$ | $0.802_{\pm 0.000}$ |
| SF2M (Tong et al., 2024b) | $0.174_{\pm 0.010}$ | $0.303_{\pm 0.023}$ | $0.430_{\pm 0.027}$ | $0.719_{\pm 0.032}$ | $0.686_{\pm 0.054}$ | $1.050_{\pm 0.047}$ | $0.871_{\pm 0.072}$ | $1.242_{\pm 0.065}$ |
| Unbalanced SB (Pariset et al., 2023) | $0.729_{\pm 0.009}$ | $0.846_{\pm 0.012}$ | $0.747_{\pm 0.012}$ | $0.823_{\pm 0.012}$ | $0.612_{\pm 0.012}$ | $0.725_{\pm 0.018}$ | $0.572_{\pm 0.029}$ | $0.944_{\pm 0.031}$ |
| uAM (Neklyudov et al., 2023) | $0.448_{\pm 0.000}$ | $0.495_{\pm 0.000}$ | $0.650_{\pm 0.000}$ | $0.691_{\pm 0.000}$ | $0.661_{\pm 0.000}$ | $0.749_{\pm 0.000}$ | $0.672_{\pm 0.000}$ | $0.864_{\pm 0.000}$ |
| Unbalanced OT (Sha et al., 2024) | $0.047_{\pm 0.000}$ | $0.060_{\pm 0.000}$ | $0.059_{\pm 0.000}$ | $0.088_{\pm 0.000}$ | $0.073_{\pm 0.000}$ | $0.084_{\pm 0.000}$ | $0.107_{\pm 0.000}$ | $0.124_{\pm 0.000}$ |
| DeepRUOT (ours) | $\mathbf{0.044}_{\pm 0.001}$ | $\mathbf{0.058}_{\pm 0.001}$ | $\mathbf{0.056}_{\pm 0.002}$ | $\mathbf{0.084}_{\pm 0.002}$ | $\mathbf{0.071}_{\pm 0.002}$ | $\mathbf{0.083}_{\pm 0.002}$ | $\mathbf{0.104}_{\pm 0.001}$ | $\mathbf{0.121}_{\pm 0.000}$ |

**Learning Waddington Developmental Landscape** In biophysics, Waddington's landscape metaphor is a well-known model for representing the cell fate decision process. Constructing such a potential landscape has been widely studied (Bian et al., 2023; 2024; Li & Wang, 2013; 2014; Shi et al., 2022; Wang et al., 2010; Zhao et al., 2024; Zhou & Li, 2016; Zhou et al., 2024c;d), however, it still remains a challenging problem in the single-cell omics data. The energy landscape

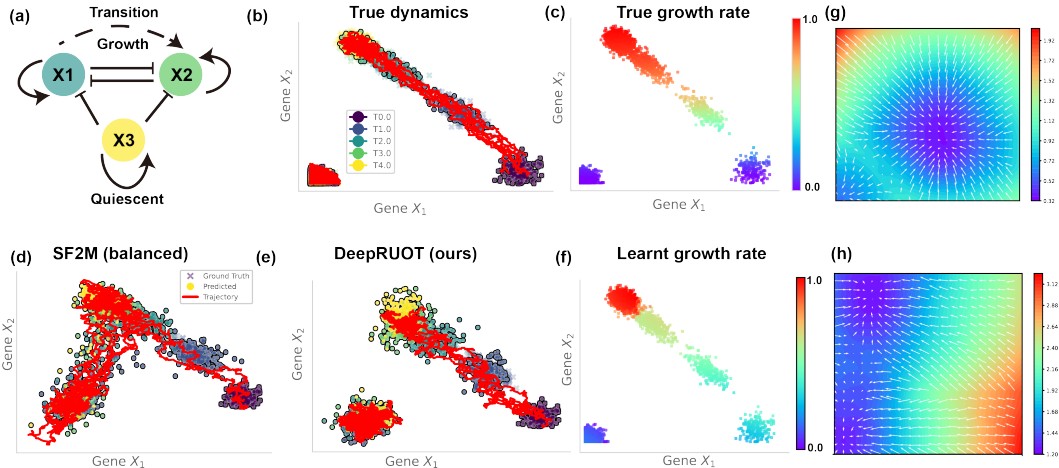

Figure 2: (a) Illustration of the synthetic gene regulatory dynamics. (b) The ground truth cellular dynamics project on $(X_1, X_2)$. (c) The ground truth growth rates. (d) The dynamics learned by balanced Schrödinger bridge (SF2M (Tong et al., 2024b), $\sigma = 0.25$). (e) The dynamics learned by our DeepRUOT solver ($\sigma = 0.25$). (f) The growth rates inferred by our DeepRUOT solver. (g) The Waddington developmental landscape learned at $t = 1$. (h) The constructed landscape at $t = 4$.

is defined by $U = -\sigma^2 \log p_{ss}/2$ where $p_{ss}$ is the steady-state PDF satisfying the Fokker-Planck equation $-\nabla \cdot (p_{ss}\boldsymbol{b}) + \frac{\sigma^2}{2}\Delta p_{ss} = g p_{ss}$. In this work, leveraging our model, we can naturally infer the time-varying potential energy landscape through the learned log density function, and use $-\sigma^2(t) \log p(\boldsymbol{x}, t)/2$ to represent the landscape at time $t$. The lower energy function indicates more stable cell fates in the landscape. In Fig. 2(g-h), we can observe the temporal evolution of the potential landscape in the synthetic gene regulation system. The quiescent cells always occupy a potential well, indicating its stability. Interestingly, the location of potential wells belonging to the transitional cell population is moving in a consistent direction with ground truth cell-state transition dynamics. Overall, the results suggest the accuracy and usefulness of DeepRUOT algorithm.

**Synthetic Gaussian Mixtures** Inspired by (Ruthotto et al., 2020), we employed a high-dimensional Gaussian mixture model to evaluate the scalability of DeepRUOT. Initially, we generated a 10-dimensional Gaussian mixture distribution. The initial density is represented as a Gaussian mixture derived from the average of two Gaussians, with their projections onto the $(x_1, x_2)$ plane depicted in Appendix B.2. The final density is also modeled as a mixture of three Gaussians, with its projection onto $(x_1, x_2)$ planes. Fig. 5 (Appendix B.2) demonstrates that our model effectively incorporates growth while learning the stochastic dynamics that transition from the initial to the target distribution. We observe that the cells in the upper region exhibit proliferation without transport. If the true cell dynamics indeed follow this pattern, models that neglect the growth factor, such as conventional balanced methods, would fail to capture these dynamics. Conversely, if the true dynamics involve transport from the lower to the upper cells, our model can still recover such dynamics, provided more detailed temporal resolution data is available. This highlights the robustness of our approach in capturing complex dynamical behaviors involving both growth and transport. Similarly, we visualized the energy landscapes at the initial and final time points Fig. 5 (Appendix B.2). The results reveal that the cells ultimately differentiate into three distinct fate states.

**Real Single-Cell Population Dynamics** Next, we evaluate our algorithm on a real scRNA-seq dataset. We use the same dataset as in (Sha et al., 2024; Weinreb et al., 2020), which involves mouse hematopoiesis analyzed by using a lineage tracing technique. After batch correction across different experiments, the data was projected onto 2D reduced-dimension force-directed layouts (SPRING plots). A clear bifurcation is observed where early-stage progenitor cells differentiate into two distinct fates (Fig. 3(a)). Using the RUOT-based approach, we learn the underlying stochastic dynamics of the data, along with the growth rates and developmental landscape at different times (Fig. 3(b),(c),(d)). We similarly evaluated quantitative metrics to compare our method with several

baseline approaches (Table 2). We find that our method outperforms others (Appendix B.3). We then test our method on a time-series single cell dataset from an A549 cancer cell line, where cells were exposed to TGFB1 to induce EM (Sha et al., 2024). We project the data with PCA on a ten-dimensional latent space as the input of our algorithm. The results are shown in Appendix B.4, which indicates that DeepRUOT remains effective and applicable in higher-dimensional settings, demonstrating its versatility and robustness in modeling complex single-cell dynamics.

Table 2: Wasserstein distance ($\mathcal{W}_1$ and $\mathcal{W}_2$) of predictions at different points across five runs on scRNA-seq data (Appendix C.1, $\sigma = 0.25$). We show the mean value with one standard deviation.

| Model | $t = 1$ | | $t = 2$ | |
|---|---|---|---|---|
| | $\mathcal{W}_1$ | $\mathcal{W}_2$ | $\mathcal{W}_1$ | $\mathcal{W}_2$ |
| MIOFlow (Huguet et al., 2022) | $0.276725_{\pm 0.000}$ | $0.312102_{\pm 0.000}$ | $0.307610_{\pm 0.000}$ | $0.402190_{\pm 0.000}$ |
| SF2M (Tong et al., 2024b) | $0.167477_{\pm 0.003}$ | $0.213489_{\pm 0.006}$ | $0.190020_{\pm 0.016}$ | $0.241516_{\pm 0.022}$ |
| Unbalanced SB (Pariset et al., 2023) | $0.387538_{\pm 0.009}$ | $0.460603_{\pm 0.008}$ | $\mathbf{0.128254}_{\pm 0.003}$ | $0.188339_{\pm 0.009}$ |
| uAM (Neklyudov et al., 2023) | $0.744918_{\pm 0.000}$ | $0.851704_{\pm 0.000}$ | $0.777237_{\pm 0.000}$ | $0.889527_{\pm 0.000}$ |
| Unbalanced OT (Sha et al., 2024) | $0.313522_{\pm 0.000}$ | $0.396947_{\pm 0.000}$ | $0.342230_{\pm 0.000}$ | $0.469342_{\pm 0.000}$ |
| DeepRUOT (ours) | $\mathbf{0.145026}_{\pm 0.002}$ | $\mathbf{0.172878}_{\pm 0.002}$ | $0.132411_{\pm 0.006}$ | $\mathbf{0.167328}_{\pm 0.010}$ |

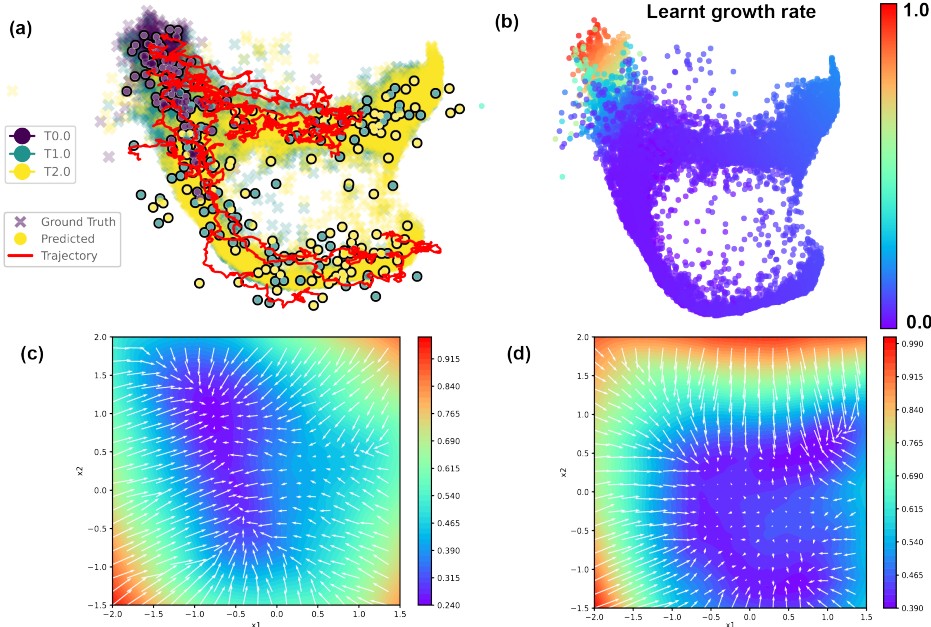

Figure 3: Application in hematopoiesis scRNA-seq data ($\sigma = 0.25$). (a) The stochastic dynamics learned by RUOT. (b) The growth rates learned by DeepRUOT. (c) The constructed Waddington developmental landscape at $t = 0$. (d) The landscape at $t = 2$.

## 7 Conclusion

We have introduced DeepRUOT for learning regularized unbalanced optimal transport (RUOT) and continuous unbalanced stochastic dynamics from time-series snapshot data. By leveraging Fisher regularization, our method transforms an SDE problem into an ODE constraint. Through the use of neural network modeling for growth and death, our framework models dynamics without requiring prior knowledge of these processes. We have demonstrated the effectiveness of our method on a synthetic gene regulatory network, high-dimensional Gaussian Mixture Model, and single-cell RNA-seq data, showing its ability to eliminate false transitions caused by neglecting growth processes. Future directions involve extending the methodology to learn the cell latent embedding space jointly with the dynamics and developing more computationally efficient algorithms. We expect future work to address these aspects and improve trajectory inference problems in various scenarios, including multi-omics data and other scenarios beyond biology in machine learning.

## ACKNOWLEDGMENTS

We thank Dr. Hugo Lavenant (Bocconi) for helpful discussions on Section E.1 and Qiangwei Peng (PKU) for helpful suggestions on the algorithm. We thank Prof. Qing Nie (UCI), Prof. Chunhe Li (Fudan), and Juntan Liu (Fudan) for their insightful discussions. This work was supported by the National Key R&D Program of China (No. 2021YFA1003301 to T.L.) and National Natural Science Foundation of China (NSFC No. 12288101 to T.L. & P.Z., and 8206100646, T2321001 to P.Z.). We acknowledge the support from the High-performance Computing Platform of Peking University for computation. We thank the anonymous referees for their valuable feedback and constructive suggestions.

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

## A  TRAINING REGULARIZED UNBALANCED OPTIMAL TRANSPORT

### A.1  TRAINING OF REGULARIZED UNBALANCED OPTIMAL TRANSPORT

After the pre-training stage, once we have obtained initial $v_\theta, g_\theta$ and $s_\theta$, then we can continue training by minimizing the total loss:

$$\mathcal{L} = \mathcal{L}_{\text{Energy}} + \lambda_r \mathcal{L}_{\text{Recons}} + \lambda_f \mathcal{L}_{\text{FP}}.$$

To compute the each component of the loss, the temporal integral and ODEs were numerically solved using the Neural ODE solver (Chen et al., 2018). The gradients of the loss function with respect to the parameters in the neural networks for $v(x, t)$, $g(x, t)$ and $s(x, t)$ were computed using neural ODEs with a memory-efficient implementation. For computing the Wasserstein distance between discrete distributions, we utilize the implementation provided by the Python Optimal Transport library (POT) (Flamary et al., 2021). The algorithm is detailed in Algorithm 1 (Training Stage).

## A.2 Training Log Density Function

To utilize conditional flow matching (CFM) to learn an initial log density, first we choose pair samples $(\boldsymbol{x}_0, \boldsymbol{x}_1)$ from the optimal transport plan $q(\boldsymbol{x}_0, \boldsymbol{x}_1)$ and then construct Brownian bridges between these pair samples. We first consider $\sigma(t) = \sigma$ is constant. The log density function is then matched with these Brownian bridges, i.e., $p(\boldsymbol{x}, t \mid (\boldsymbol{x}_0, \boldsymbol{x}_1)) = \mathcal{N}\left(\boldsymbol{x}; t\boldsymbol{x}_1 + (1-t)\boldsymbol{x}_0, \sigma^2 t(1-t)\right)$ and $\nabla_{\boldsymbol{x}} \log p(\boldsymbol{x}, t \mid (\boldsymbol{x}_0, \boldsymbol{x}_1)) = \frac{t\boldsymbol{x}_1 + (1-t)\boldsymbol{x}_0 - \boldsymbol{x}}{\sigma^2 t(1-t)}$, $t \in [0, 1]$. We then utilize the fully connected neural networks $s_\theta(\boldsymbol{x}, t)$ to approximate the log density function $\frac{1}{2}\sigma^2 \log p(\boldsymbol{x}, t)$ with a weighting function $\lambda_s$. Then we have

$$\mathcal{L}_{us} = \lambda_s^2 \|\nabla_{\boldsymbol{x}} s_\theta - \frac{1}{2}\sigma^2 \nabla_{\boldsymbol{x}} \log p(\boldsymbol{x}, t)\|_2^2.$$

And the correspongding CFM loss is

$$\mathcal{L}_{\text{score}} = \mathbb{E}_{Q'} \lambda_s^2 \|\nabla_{\boldsymbol{x}} s_\theta - \frac{1}{2}\sigma^2 \nabla_{\boldsymbol{x}} \log p(\boldsymbol{x}, t \mid (\boldsymbol{x}_0, \boldsymbol{x}_1))\|_2^2,$$

where $Q' = (t \sim \mathcal{U}(0, 1)) \otimes q(\boldsymbol{x}_0, \boldsymbol{x}_1) \otimes p(\boldsymbol{x}, t \mid (\boldsymbol{x}_0, \boldsymbol{x}_1))$. We take the weighting function as

$$\lambda_s(t) = \frac{2}{\sigma^2}\sigma\sqrt{t(1-t)} = \frac{2\sqrt{t(1-t)}}{\sigma}.$$

Then we have

$$\begin{aligned}
\mathcal{L}_{\text{score}} &= \lambda_s(t)^2 \left\|\nabla_{\boldsymbol{x}} s_\theta(\boldsymbol{x}, t) - \frac{\sigma^2}{2}\nabla_{\boldsymbol{x}} \log p\left(\boldsymbol{x} \mid \boldsymbol{x_0}, \boldsymbol{x_1}\right)\right\|_2^2, \\
&= \left\|\lambda_s(t)\nabla_{\boldsymbol{x}} s_\theta(\boldsymbol{x}, t) - \lambda_s(t)\frac{\sigma^2}{2}\nabla_{\boldsymbol{x}} \log p\left(\boldsymbol{x} \mid \boldsymbol{x_0}, \boldsymbol{x_1}\right)\right\|_2^2, \\
&= \left\|\lambda_s(t)\nabla_{\boldsymbol{x}} s_\theta(\boldsymbol{x}, t) + \boldsymbol{\epsilon}_1\right\|_2^2,
\end{aligned} \tag{13}$$

where $\boldsymbol{\epsilon}_1 \sim \mathcal{N}(0, \boldsymbol{I})$. This is also numerically stable. For the case where $\sigma(t)$ is not constant, a similar approach can be applied. One may refer to (Tong et al., 2024b) for further details.

## A.3 Estimating Initial Distribution for Fokker-Planck Equation

The initial distribution is estimated by a Gaussian mixture model to generate density

$$p(\boldsymbol{x}, 0) = p_0(\boldsymbol{x}) = \sum_{i=1}^{N_0} \frac{\exp\left(-\frac{1}{2}(\boldsymbol{x} - \boldsymbol{x}_i(t_0))^T \boldsymbol{\Sigma}^{-1}(\boldsymbol{x} - \boldsymbol{x}_i(t_0))\right)}{\sqrt{(2\pi)^d |\boldsymbol{\Sigma}|}}, \boldsymbol{\Sigma} = \sigma\boldsymbol{I} \in \mathbb{R}^{d \times d}.$$

## A.4 Energy Loss

In (Sha et al., 2024), it introduces the following theorem to transform the high-dimensional integral to the Monte Carlo integral.

**Theorem A.1.** *If smooth density $p(\boldsymbol{x}, t) : \mathbb{R}^d \times [0, 1] \to \mathbb{R}^+$, velocity field $\boldsymbol{v}(\boldsymbol{x}, t) : \mathbb{R}^d \times [0, 1] \to \mathbb{R}^d$ and growth rate $g(\boldsymbol{x}, t) : \mathbb{R}^d \times [0, 1] \to \mathbb{R}$ satisfy*

$$\begin{cases} \partial_t p(\boldsymbol{x}, t) + \nabla \cdot (\boldsymbol{v}(\boldsymbol{x}, t)p(\boldsymbol{x}, t)) = g(\boldsymbol{x}, t)p(\boldsymbol{x}, t), \\ p(\boldsymbol{x}, 0) = p_0(\boldsymbol{x}), \end{cases}$$

*for all $0 \le t \le 1$ with $\frac{d\boldsymbol{x}(t)}{dt} = \boldsymbol{v}(\boldsymbol{x}, t)$ and $\boldsymbol{x}(0) = \boldsymbol{x}_0$, then for any measurable function $f(\boldsymbol{x}, t) : \mathbb{R}^d \times [0, 1] \to \mathbb{R}^d$, we have $\int_0^1 \int_{\mathbb{R}^d} f(\boldsymbol{x}, t)p(\boldsymbol{x}, t)d\boldsymbol{x}dt = \mathbb{E}_{\boldsymbol{x_0} \sim p_0} \int_0^1 f(\boldsymbol{x}, t)e^{\int_0^t g(\boldsymbol{x}, s)ds}dt.$*

# B  ADDITIONAL RESULTS

## B.1  SYNTHETIC GENE REGULATORY NETWORK

The system dynamics are governed by the following set of stochastic ordinary differential equations (ODEs):

$$\frac{\mathrm{d}X_1}{\mathrm{d}t} = \frac{\alpha_1 X_1^2 + \beta}{1 + \alpha_1 X_1^2 + \gamma_2 X_2^2 + \gamma_3 X_3^2 + \beta} - \delta_1 X_1 + \eta_1 \xi_t,$$

$$\frac{\mathrm{d}X_2}{\mathrm{d}t} = \frac{\alpha_2 X_2^2 + \beta}{1 + \gamma_1 X_1^2 + \alpha_2 X_2^2 + \gamma_3 X_3^2 + \beta} - \delta_2 X_2 + \eta_2 \xi_t,$$

$$\frac{\mathrm{d}X_3}{\mathrm{d}t} = \frac{\alpha_3 X_3^2}{1 + \alpha_3 X_3^2} - \delta_3 X_3 + \eta_3 \xi_t.$$

Genes $X_1$ and $X_2$ mutually inhibit each other and self-activate, forming a toggle switch. An external signal $\beta$ uniformly activates both $X_1$ and $X_2$ independently of gene expression levels. Gene $X_3$ inhibits the expression of both $X_1$ and $X_2$. Here, $X_i(t)$ represents the concentration of gene $i$ at time $t$, with $\alpha_i$ and $\gamma_i$ denoting the strengths of self-activation and inhibition, respectively. The parameters $\delta_i$ represent gene degradation rates, while $\eta_i \xi_t$ accounts for the stochastic effects with additive white noise. The probability of cell division correlates positively with the expression of $X_2$, calculated as $g = \alpha_g \frac{X_2^2}{1 + X_2^2}\%$. Upon division, cells inherit the gene expression states $(X_1(t), X_2(t), X_3(t))$ of the parent cell, subject to independent perturbations $\eta_d \mathcal{N}(0, 1)$ for each gene, and transition independently thereafter. The hyper-parameters are listed at Table 3. The initial cells are chosen independently and identically distributed from two normal distributions $\mathcal{N}([2, 0.2, 0], 0.01)$ and $\mathcal{N}([0, 0, 2], 0.01)$. At each step, we corrected the negative expression value to 0.

Table 3: Simulation parameters on gene regulatory network.

| Parameter | Value | Description |
|---|---|---|
| $\alpha_1$ | 0.5 | Strength of self-activation for $X_1$ |
| $\gamma_1$ | 0.5 | Strength of inhibition by $X_3$ on $X_1$ |
| $\alpha_2$ | 1 | Strength of self-activation for $X_2$ |
| $\gamma_2$ | 1 | Strength of inhibition by $X_3$ on $X_2$ |
| $\alpha_3$ | 1 | Strength of self-activation for $X_3$ |
| $\gamma_3$ | 10 | Half-saturation constant for inhibition terms |
| $\delta_1$ | 0.4 | Degradation rate for $X_1$ |
| $\delta_2$ | 0.4 | Degradation rate for $X_2$ |
| $\delta_3$ | 0.4 | Degradation rate for $X_3$ |
| $\eta_1$ | 0.05 | Noise intensity for $X_1$ |
| $\eta_2$ | 0.05 | Noise intensity for $X_2$ |
| $\eta_3$ | 0.01 | Noise intensity for $X_3$ |
| $\eta_d$ | 0.014 | Noise intensity for cell perturbations |
| $\beta$ | 1 | External signal activating $X_1$ and $X_2$ |
| $dt$ | 1 | Time step size |
| Time Points | [0, 8, 16, 24, 32] | Time points at which data is recorded |

We conducted a comprehensive evaluation of the methodologies proposed by (Neklyudov et al., 2023; Pariset et al., 2023) utilizing our simulated dataset. As detailed in Table 1, our approach consistently outperforms the referenced methods across multiple quantitative metrics, thereby demonstrating its superior ability to capture the underlying dynamics of the system. Furthermore, Fig. 4 provides the outcomes produced by the UDSB (Pariset et al., 2023). While their method predicts an overall increase in cell population, a closer examination reveals the presence of false dynamics within the predicted transitions.

## B.2  SYNTHETIC GAUSSIAN MIXTURES

For the initial distribution, we generated 400 samples from the Gaussian located lower in the $(x_1, x_2)$ plane, and 100 samples from the Gaussian positioned higher. For the final distribution, we generated

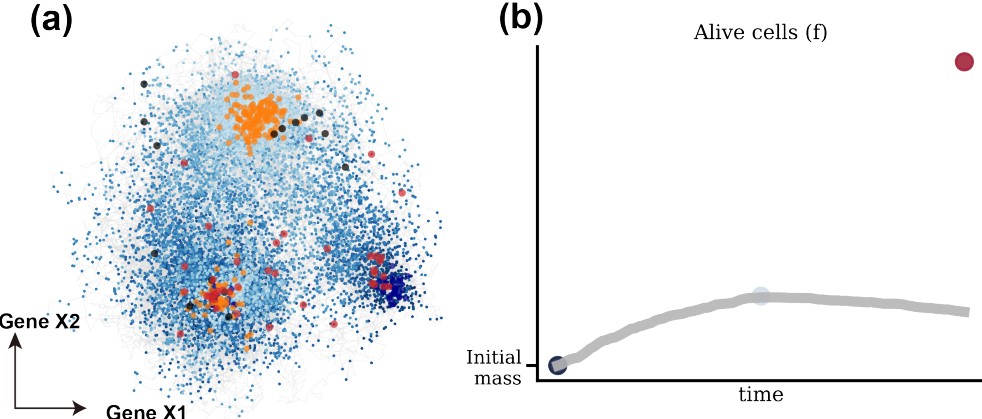

Figure 4: **Results obtained by UDSB ([Pariset et al., 2023](#)) on gene regulatory network.** (a) The trajectory learned by UDSB, where black dots indicate particle death, red dots signify particle growth, orange dots represent the target distribution, dark blue dots denote the source distribution, and gradient blue dots illustrate particle trajectories. (b) Predicted changes in cell population at intermediate time points, with dots representing the actual mass.

1,000 samples from the upper Gaussian, and 200 samples each from the two lower Gaussians. We then tested the RUOT-based model's ability to learn the stochastic dynamics using the samples generated in this manner.

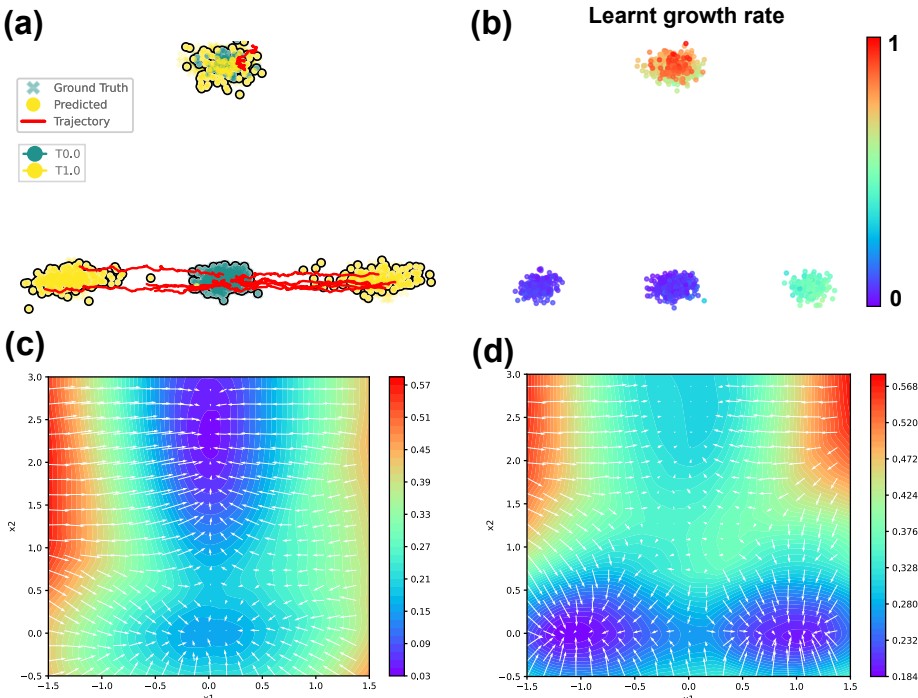

Figure 5: **Results of DeepRUOT on Gaussian mixtures ($\sigma = 0.1$, 10D).** (a) The learned trajectory by DeepRUOT ($\sigma = 0.1$). (b) The growth rate inferred by our model. (c) The Waddington developmental landscape at $t = 0$ ($\sigma = 0.1$). (d) The Waddington developmental landscape at $t = 1$ ($\sigma = 0.1$).

To further evaluate the scalability of our algorithm, we simulated Gaussian mixture models with 20, 50, and 100 dimensions, as illustrated in Fig. 6. We similarly visualize the results on the first and second dimensions. The results demonstrate that our model remains effective and applicable across these higher-dimensional settings.

Similar to the frameworks such as TrajectoryNet (Tong et al., 2020), MIOflow (Huguet et al., 2022) and (Koshizuka & Sato, 2023), which involve simulating an ODE/SDE and performing optimal transport for distribution matching, the computational cost and scalability of our method are comparable to these existing approaches. Notably, the score matching in our method is conducted via conditional flow matching (Lipman et al., 2023; Tong et al., 2024a;b), which is simulation-free and thus highly efficient. Consequently, our approach does not introduce significant additional computational overhead, since the cost of each component is similar to previous works. Meanwhile, when it extends to thousands of dimensions, our approach similarly encounters challenges, necessitating the development of simulation-free training methods akin to flow matching to handle higher-dimensional settings effectively (Lipman et al., 2023; Tong et al., 2024a;b). However, in the field of single-cell biology, dimensionality reduction of gene expression data is routinely employed, making hundreds of dimensional space sufficient for practical applications.

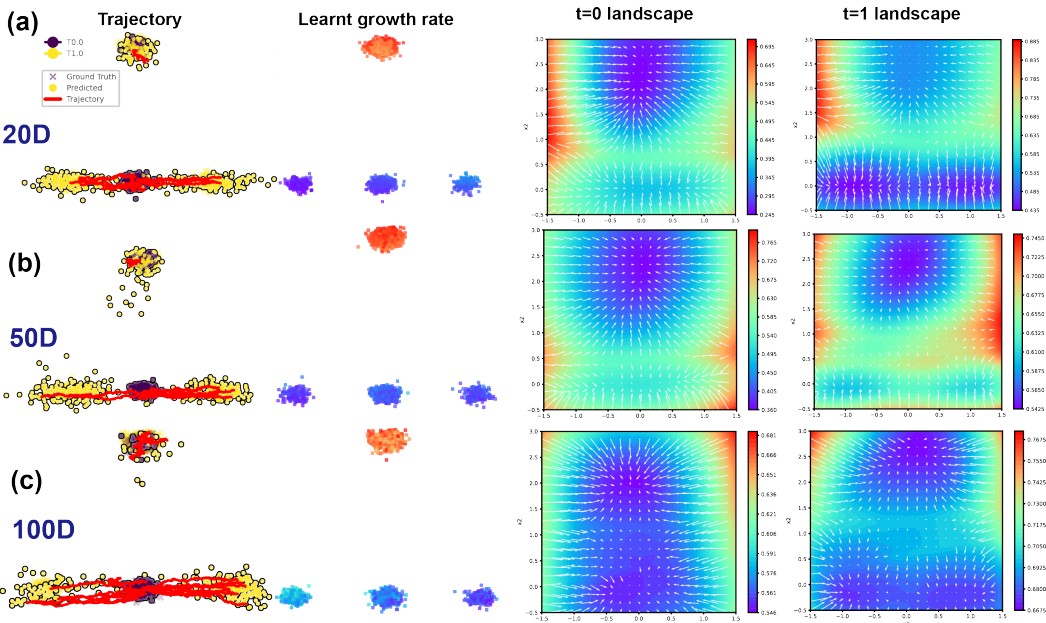

Figure 6: **Results of DeepRUOT on higher dimensional Gaussian mixtures ($\sigma = 0.1$).** The first column presents the trajectories learned by DeepRUOT ($\sigma = 0.1$), the second column displays the growth rates inferred by our model, the third column shows the Waddington developmental landscape at $t = 0$ ($\sigma = 0.1$), and the fourth column depicts the Waddington developmental landscape at $t = 1$ ($\sigma = 0.1$). (a) 20 dimensions, (b) 50 dimensions, (c) 100 dimensions.

### B.3 SINGLE CELL DYNAMICS IN MOUSE HEMATOPOIESIS

We conducted a comparative evaluation of the methodologies proposed by (Neklyudov et al., 2023; Pariset et al., 2023) using our mouse hematopoiesis dataset. As detailed in Table 1, our approach consistently outperforms the referenced methods across a range of quantitative metrics, underscoring its superior efficacy in capturing the underlying biological dynamics. In Fig. 7, we present the results generated by the UDSB (Pariset et al., 2023). Fig. 7(a) displays the learned trajectory, where black dots indicate particle death, red dots signify particle growth, orange dots represent the target distribution, dark blue dots denote the source distribution, and gradient blue lines illustrate the particle trajectories. This visualization highlights the dynamic trajectory of cell population changes over time. Fig. 7(b) illustrates the predicted changes in cell population at intermediate time points, with

each dot representing the actual mass observed. Finally, Fig. 7(c) shows the predicted cell distributions at various time points in red, while the blue distribution at the final time point represents the initial distribution. This panel underscores the model's ability to track and predict the evolution of cell distributions from the initial state to subsequent stages.

Notably, although this method achieves a marginally higher $\mathcal{W}_1$ metric at the final time point, a closer examination reveals that the predicted distributions at intermediate time points, as illustrated in Fig. 7(c), deviate from the true data distributions. This indicates that the (Pariset et al., 2023) method may have challenges to accurately model the transitional dynamics that occur between the initial and final stages of hematopoiesis.

Furthermore, Fig. 7(a) and Fig. 7(b) demonstrate that the growth locations predicted by the UDSB algorithm are consistent with those identified by our model. This consistency provides a form of cross-validation, affirming that our algorithm effectively captures both the growth and migration dynamics inherent in the cell population. So the ability of our model to maintain accurate predictions across all time points, including the intermediate stages, demonstrates its enhanced capacity for modeling unbalanced dynamics within complex biological systems.

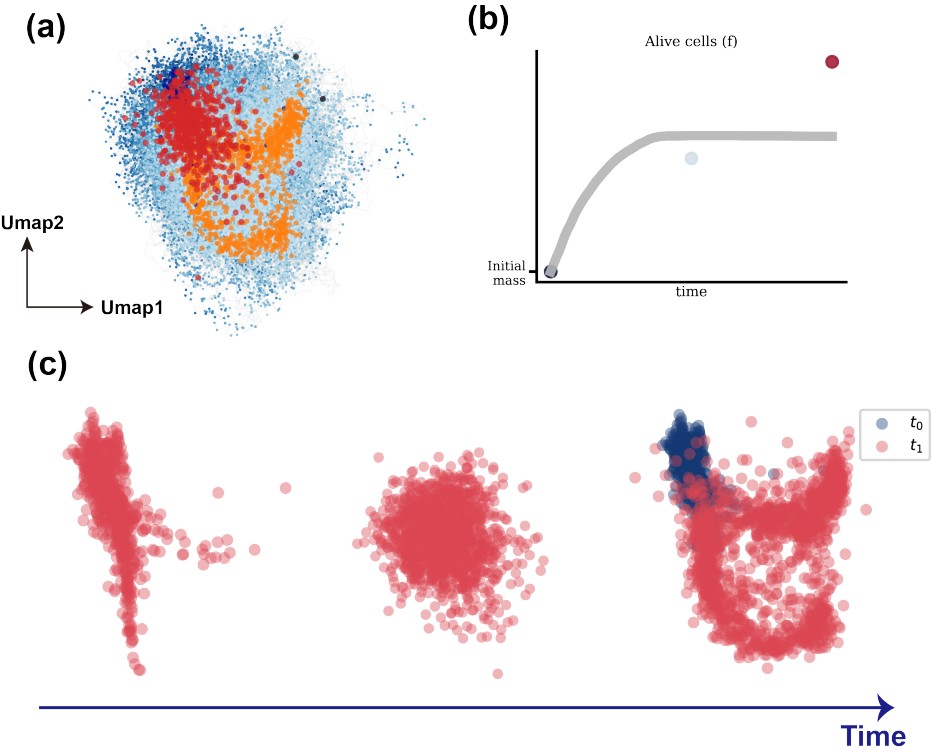

Figure 7: **Results obtained by UDSB (Pariset et al., 2023) on mouse hematopoiesis.** (a) The trajectory learned by UDSB, where black dots indicate particle death, red dots signify particle growth, orange dots represent the target distribution, dark blue dots denote the source distribution, and gradient blue dots illustrate particle trajectories. (b) Predicted changes in cell population at intermediate time points, with dots representing the actual mass. (c) Red denotes the predicted cell distributions at various time points, while the blue distribution at the final time point represents the initial distribution.

## B.4    SINGLE CELL DYNAMICS IN EMT

Subsequently, we applied DeepRUOT to a time-series single-cell RNA sequencing dataset derived from the A549 cancer cell line. In this study, cells were treated with TGFB1 to induce epithelial-mesenchymal transition (EMT) during the initial four-time points (Sha et al., 2024). Cells harvested

at each time point were cultured in vitro with identical initial cell numbers, ensuring that the cell counts directly reflect the dynamics of the cell population over time.

We employed Principal Component Analysis (PCA) to reduce the dimensionality of our single-cell RNA sequencing (scRNA-seq) data to a ten-dimensional latent space. This latent representation was subsequently utilized as the input for DeepRUOT. To aid in the interpretation of the results, we projected the algorithm's outputs onto the first and second principal components, which capture the majority of the variance in the data and thus provide meaningful insights into the underlying biological phenomena.

The trajectories inferred by DeepRUOT exhibited consistent and biologically plausible transition dynamics, effectively mapping the progression of cells through different developmental states. Furthermore, we visualized the growth rates estimated by the algorithm alongside the developmental landscapes at various time points, providing a comprehensive view of both the quantitative and qualitative aspects of cellular dynamics.

Notably, the growth patterns inferred by DeepRUOT displayed elevated values during the initial and intermediate stage of epithelial-mesenchymal transition (EMT) compared to the epithelial (E) and mesenchymal (M) stages, as illustrated in Fig. 8. This observation aligns with previous studies that have reported enhanced stemness and proliferative capacity in cells at the intermediate stage.

Moreover, our landscape analysis revealed that the inferred developmental landscapes are consistent with the observed data distributions at corresponding time points. The developmental landscapes provide a representation of the cell states, illustrating how cells traverse through different regions of the latent space as they progress through various stages of differentiation and migration.

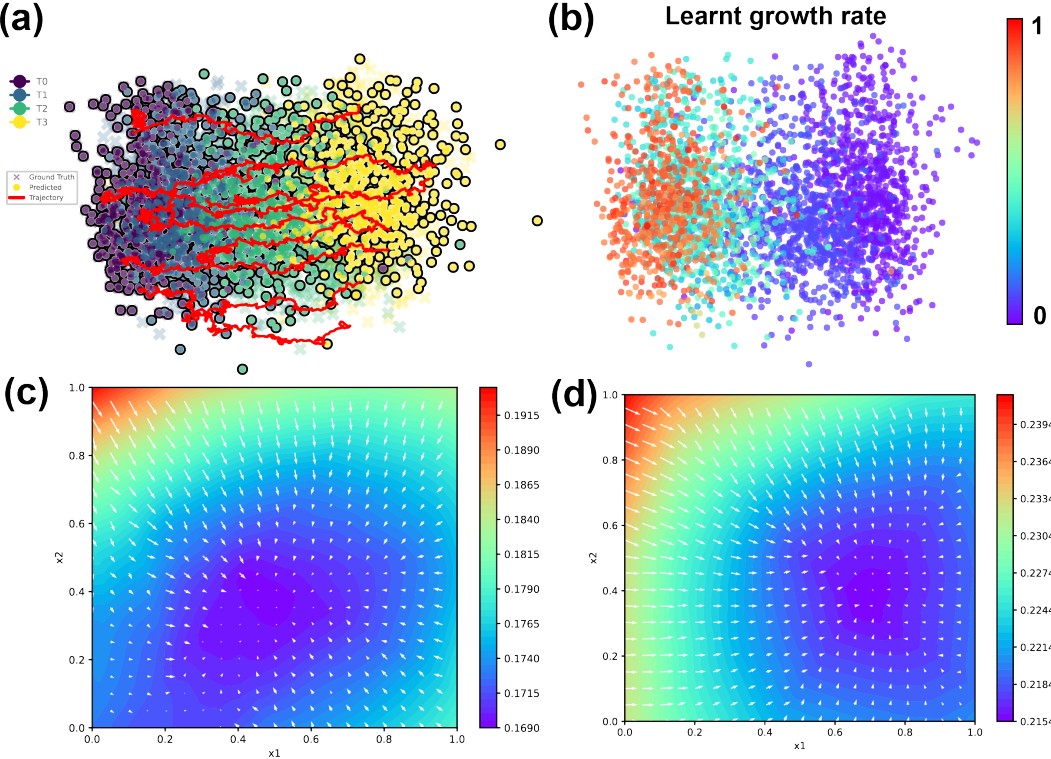

Figure 8: **Results of DeepRUOT on EMT scRNA-seq data** ($\sigma = 0.05$). (a) The stochastic dynamics learned by RUOT ($\sigma = 0.05$). (b) The growth rates learned by DeepRUOT ($\sigma = 0.05$). (c) The constructed Waddington developmental landscape at $t = 1$ ($\sigma = 0.05$). (d) The landscape at $t = 2$ ($\sigma = 0.05$).

## B.5 ABLATION STUDIES

We conducted ablation experiments on the gene regulatory dataset to evaluate the contributions of key components in our algorithm. Specifically, we examined the impact of the growth term $g(\boldsymbol{x}, t)$. Then we assessed the necessity of the mass loss term $\mathcal{L}_{\text{Mass}}$ within the reconstruction loss function $\mathcal{L}_{\text{Recons}}$ and $\mathcal{L}_{\text{FP}}$ in the total loss (9). These analyses demonstrate the critical roles that the growth term, the mass loss component, and the Fokker-Planck constraint play in enhancing the performance and accuracy of our model.

Table 4: Ablation studies across five runs on gene regulatory data ($\sigma = 0.25$). We show the mean value with one standard deviation.

| Model | $t = 1$ | | $t = 2$ | | $t = 3$ | | $t = 4$ | |
|---|---|---|---|---|---|---|---|---|
| | $\mathcal{W}_1$ | $\mathcal{W}_2$ | $\mathcal{W}_1$ | $\mathcal{W}_2$ | $\mathcal{W}_1$ | $\mathcal{W}_2$ | $\mathcal{W}_1$ | $\mathcal{W}_2$ |
| SF2M (Tong et al., 2024b) | $0.174_{\pm 0.010}$ | $0.303_{\pm 0.023}$ | $0.430_{\pm 0.027}$ | $0.719_{\pm 0.032}$ | $0.686_{\pm 0.054}$ | $1.050_{\pm 0.047}$ | $0.871_{\pm 0.072}$ | $1.242_{\pm 0.065}$ |
| DeepRUOT w/o growth | $0.173_{\pm 0.005}$ | $0.198_{\pm 0.004}$ | $0.324_{\pm 0.007}$ | $0.371_{\pm 0.005}$ | $0.481_{\pm 0.010}$ | $0.576_{\pm 0.004}$ | $0.772_{\pm 0.009}$ | $0.877_{\pm 0.011}$ |
| DeepRUOT w/o $\mathcal{L}_{\text{Mass}}$ | $0.159_{\pm 0.006}$ | $0.186_{\pm 0.006}$ | $0.308_{\pm 0.005}$ | $0.349_{\pm 0.007}$ | $0.465_{\pm 0.011}$ | $0.553_{\pm 0.011}$ | $0.690_{\pm 0.007}$ | $0.816_{\pm 0.009}$ |
| DeepRUOT w/o $\mathcal{L}_{\text{FP}}$ | $0.090_{\pm 0.004}$ | $\mathbf{0.107}_{\pm 0.005}$ | $0.108_{\pm 0.003}$ | $0.132_{\pm 0.002}$ | $0.133_{\pm 0.006}$ | $\mathbf{0.156}_{\pm 0.007}$ | $0.165_{\pm 0.008}$ | $0.197_{\pm 0.011}$ |
| DeepRUOT w/o pre-training | $0.635_{\pm 0.006}$ | $0.737_{\pm 0.009}$ | $1.113_{\pm 0.017}$ | $1.310_{\pm 0.017}$ | $1.329_{\pm 0.020}$ | $1.545_{\pm 0.025}$ | $1.366_{\pm 0.023}$ | $1.565_{\pm 0.028}$ |
| DeepRUOT w/o score matching | $1.291_{\pm 0.007}$ | $1,426_{\pm 0.009}$ | $2.051_{\pm 0.003}$ | $2.187_{\pm 0.004}$ | $2.675_{\pm 0.010}$ | $2.798_{\pm 0.009}$ | $3.290_{\pm 0.009}$ | $3.391_{\pm 0.007}$ |
| DeepRUOT w/o training | $\mathbf{0.089}_{\pm 0.004}$ | $0.108_{\pm 0.005}$ | $\mathbf{0.102}_{\pm 0.002}$ | $0.132_{\pm 0.012}$ | $0.134_{\pm 0.007}$ | $0.158_{\pm 0.013}$ | $0.163_{\pm 0.009}$ | $0.200_{\pm 0.015}$ |
| DeepRUOT | $0.095_{\pm 0.005}$ | $0.115_{\pm 0.006}$ | $0.104_{\pm 0.002}$ | $\mathbf{0.130}_{\pm 0.007}$ | $\mathbf{0.130}_{\pm 0.003}$ | $0.157_{\pm 0.003}$ | $\mathbf{0.140}_{\pm 0.007}$ | $\mathbf{0.168}_{\pm 0.007}$ |

We observe that when our algorithm does not account for growth, i.e., by setting $g(\boldsymbol{x}, t) = 0$, it reduces to regularized optimal transport, which is equivalent to the Schrödinger bridge problem. Consequently, SF2M (Tong et al., 2024b) corresponds to the scenario where $g(\boldsymbol{x}, t) = 0$ within our approach. Our experiments have demonstrated that omitting the growth factor leads to inaccurate trajectory reconstructions, underscoring the necessity of modeling unbalanced dynamics. To further substantiate this finding, we conducted an explicit ablation study with $g(\boldsymbol{x}, t)$ (Table 4). The results from this experiment align with those of the SF2M, confirming that disabling the growth rate adversely affects performance by neglecting essential growth and death processes. This comparison reinforces the importance of the growth rate component in accurately capturing unbalanced stochastic dynamics, thereby validating the enhanced performance of DeepRUOT.

Next, we assess the impact of the mass loss term, $\mathcal{L}_{\text{Mass}}$, within the reconstruction loss function $\mathcal{L}_{\text{Recons}} = \lambda_m \mathcal{L}_{\text{Mass}} + \lambda_d \mathcal{L}_{\text{OT}}$ during all training stage. By setting $\lambda_m = 0$, we observed that the exclusion of this component results in inaccurate trajectory reconstructions (see Table 4). This finding underscores the crucial role of incorporating local mass matching in unbalanced settings, demonstrating that the omission of $\mathcal{L}_{\text{Mass}}$ significantly impairs the model's ability to accurately capture the system's dynamics.

Subsequently, we investigate the training stage using the loss function $\mathcal{L} = \mathcal{L}_{\text{Energy}} + \lambda_r \mathcal{L}_{\text{Recons}} + \lambda_f \mathcal{L}_{\text{FP}}$. By setting $\lambda_f = 0$, thereby excluding the Fokker-Planck constraint, the results presented in Table 4 reveal that incorporating this loss term enhances our model's performance, particularly at the final time point. These results collectively highlight the importance of both mass matching and the Fokker-Planck constraint in improving the accuracy and reliability of trajectory reconstructions within our framework.

Furthermore, we investigated the impact of different stages in our training process on the final results. Our training algorithm (Algorithm 1) comprises two stages: the pre-training stage and the training stage. To assess their respective contributions, we conducted experiments under three scenarios: (1) omitting the pre-training stage, (2) excluding score matching during the pre-training stage, and (3) removing the training stage. As presented in Table 4, we observed that the algorithm failed to converge in the first two scenarios. Specifically, scenario (2) yielded poorer performance due to inaccurate score matching, which caused particle trajectories to deviate from the correct dynamical paths. In scenario (1), the necessity to balance multiple loss functions resulted in the vector field and score not being adequately learned, leading particles are inclined to remain near their initial positions. In scenario (3), the absence of the training stage slightly degraded the performance metrics, especially at later time points. These ablation studies demonstrate that the pre-training stage is critical within our framework, while the training stage contributes to the refinement of final results, ensuring we are addressing the regularized optimal transport problem, which is theoretically necessary. Empirical results indicate that the pre-training stage significantly enhances the performance of our algorithm.

## C    EXPERIENTIAL DETAILS

### C.1    EVALUATION METRICS

We evaluate the empirical 1-Wasserstein distance ($\mathcal{W}_1$) and 2-Wasserstein distance ($\mathcal{W}_2$) on the gene data and scRNA-seq data. It is defined as follows:

$$\mathcal{W}_2(p,q) = \left( \min_{\pi \in \mathbf{\Pi}(p,q)} \int \|\boldsymbol{x} - \boldsymbol{y}\|_2^2 d\pi(\boldsymbol{x}, \boldsymbol{y}) \right)^{1/2},$$

and

$$\mathcal{W}_1(p,q) = \left( \min_{\pi \in \mathbf{\Pi}(p,q)} \int \|\boldsymbol{x} - \boldsymbol{y}\|_2 d\pi(\boldsymbol{x}, \boldsymbol{y}) \right),$$

where $p$ and $q$ represent empirical distributions.

We evaluate our method by first learning the dynamics using data from all time points. We then apply these inferred dynamics to the initial data point to generate data for subsequent time points. Next, we compute the $\mathcal{W}_1$ and $\mathcal{W}_2$ distances between the generated data and the real data. For the gene regulatory network data, we ensure that the number of generated data points matches that of the real data since we have the true dynamics. In the scRNA-seq dataset, since the true dynamics are unknown, we apply the inferred dynamics to the initial data and compute the $\mathcal{W}_1$ and $\mathcal{W}_2$ distances between the generated data and all the data at that time point. Because our method could assign weights to the sample points, we compute the weighted $\mathcal{W}_1$ and $\mathcal{W}_2$ distances accordingly in the evaluation of our methods. We take the weight computed by DeepRUOT without diffusion to use for simplicity. For other methods we used the default uniform distribution to evaluate.

To evaluate the performance of SF2M and DeepRUOT on scRNA-seq datasets we take the same diffusion coefficient as $\sigma = 0.25$. To evaluate the performance on synthetic gene regulatory data, we choose $\sigma = 0.05$ for our DeepRUOT method and $\sigma = 0.25$ for SF2M. This is due to we observe some numerical instability when using SF2M with $\sigma = 0.05$. So to ensure a fair comparison, we choose the parameter as default in SF2M.

To evaluate the performance of (Sha et al., 2024) on gene regulatory network data and single-cell RNA sequencing (scRNA-seq) mouse hematopoiesis data, as we observed certain instabilities on the datasets, to ensure a fair comparison, we independently reimplemented the method and conducted additional evaluations (by setting $\sigma = 0$ in DeepRUOT).

To evaluate the performance of (Neklyudov et al., 2023) and (Pariset et al., 2023) on gene regulatory network data and single-cell RNA sequencing (scRNA-seq) mouse hematopoiesis data, we utilized their default parameter settings. Specifically, for (Pariset et al., 2023), whose default configuration involves three time points, we selected samples at time points 0, 2, and 4 from the gene regulatory network dataset as inputs for the algorithms.

### C.2    HYPERPARAMETERS SELECTION AND LOSS WEIGHTING

We use one A100 Nvidia GPU along with 16 CPU cores for computation at a shared high-performance computing cluster. For the two-stage training of DeepRUOT, each stage necessitates the selection of hyperparameters tailored to its specific requirements, and here we will provide a guideline based on empirical studies.

In the pre-training phase, we need to select the parameters in $\mathcal{L}_{\text{Recons}}$, i.e., $\lambda_m$ and $\lambda_d$ in $\mathcal{L}_{\text{Recons}} = \lambda_m \mathcal{L}_{\text{Mass}} + \lambda_d \mathcal{L}_{\text{OT}}$. Parameter selection is important during this stage, as in the ablation studies we have shown that both components are important. To effectively manage these parameters, we implement a hyperparameter scheduling strategy during the pre-training stage. Initially, we assign predefined values to $\lambda_m$ and $\lambda_d$ and train the model for a specified number of epochs. Subsequently, we adjust $\lambda_m$ to zero ($\lambda_m = 0$) and conduct additional epochs of training. This strategy is grounded in the rationale of first aligning growth patterns and subsequently refining the transition process based on the established growth rates. The parameter scheduling procedure is elucidated in Table 5, where the arrow notation ($\rightarrow$) signifies the rescheduling of hyperparameter values during the pre-training stage. Specifically, each arrow indicates the change from an initial value to a subsequent value, reflecting the strategic adjustment of hyperparameters during the pre-training stage.

During the training stage, the pre-training phase provides robust initial values, thereby reducing the sensitivity to parameter variations in $\mathcal{L} = \mathcal{L}_{\text{Energy}} + \lambda_r \mathcal{L}_{\text{Recons}} + \lambda_f \mathcal{L}_{\text{FP}}$. As a result, we utilize the set of parameters and train the model for 10 epochs consistently across different datasets.

As shown in Table 5, the primary hyperparameters requiring adjustment during the pre-training stage are mainly the number of training epochs selected during the $\lambda_m$ resetting process, specifically the number of epochs trained before $\lambda_m$ resetting and the number of epochs trained after $\lambda_m$ resetting. As previously discussed, the pre-training stage is crucial for stabilizing the training process. Determining the number of epochs relies on the complexity and scale of the problem, including factors such as the number of time points and the size of the dataset. For the first epoch number, a default of 30 epochs is recommended. However, in the case of the Gene Regulatory Network dataset, we opted for 20 epochs due to its larger number of time points and greater data volume. Similarly, for the EMT dataset, 20 epochs were chosen as it had already achieved convergence.

Table 5: Parameter Settings for Different Datasets Across Two Training Stages (Regulatory Network ($\sigma = 0.25$), Mouse Hematopoiesis ($\sigma = 0.25$), EMT ($\sigma = 0.05$), Gaussian Mixtures ($\sigma = 0.1$)). The arrow ($\rightarrow$) indicates the scheduling of parameter values within the Pre-Training Stage, where the numbers on the left indicate the hyperparameter values before resetting, and the right ones indicate the values after resetting. For example, the $(1.0, 0.1, 20) \rightarrow (0.0, 0.1, 10)$ entry in $(\lambda_m, \lambda_d, \text{Epochs})$ row denotes 20 epochs training when $\lambda_m$ was 1.0, $\lambda_d$ was 0.1, and another 10 epochs training after $\lambda_m$ was reset to 0.0.

| Parameter | Datasets | | | | | | |
| --- | --- | --- | --- | --- | --- | --- | --- |
| | Regulatory Network | Mouse Hematopoiesis | EMT | Gaussian Mixtures | | | |
| | | | | 10D | 20D | 50D | 100D |
| **Pre-Training Stage (With Hyperparameters Scheduling)** | | | | | | | |
| $(\lambda_m, \lambda_d, \text{Epochs})$ | $(1.0,0.1,20) \rightarrow (0.0,0.1,10)$ | $(1.0,0.1,30) \rightarrow (0.0,0.1,30)$ | $(1.0,0.1,20) \rightarrow (0.0,0.1,0)$ | $(1.0,0.1,35) \rightarrow (0.0,0.1,80)$ | $(1.0,0.1,30) \rightarrow (0.0,0.1,90)$ | $(1.0,0.1,35) \rightarrow (0.0,0.1,120)$ | $(1.0,0.1,35) \rightarrow (0.0,0.1,100)$ |
| **Training Stage** | | | | | | | |
| $\lambda_m$ | $1 \times 10^3$ | $1 \times 10^3$ | $1 \times 10^3$ | $1 \times 10^3$ | $1 \times 10^3$ | $1 \times 10^3$ | $1 \times 10^3$ |
| $\lambda_d$ | 1.0 | 1.0 | 1.0 | 1.0 | 1.0 | 1.0 | 1.0 |
| $\lambda_r$ | 1.0 | 1.0 | 1.0 | 1.0 | 1.0 | 1.0 | 1.0 |
| $\lambda_f$ | 1.0 | 1.0 | 1.0 | 1.0 | 1.0 | 1.0 | 1.0 |
| $\lambda_w$ | 0.1 | 0.1 | 0.1 | 0.1 | 0.1 | 0.1 | 0.1 |
| Epochs | 10 | 10 | 10 | 10 | 10 | 10 | 10 |

In Table 6, we evaluated the GRN dataset by maintaining the second epoch number (trained after $\lambda_m$ resetting) as 10 while varying the first epoch number (trained before resetting) among 10, 20, and 30, with the other hyperparameters chosen as Table 5. The results indicated that the first epoch number value of 10, which was insufficient, deteriorated performance, whereas the first epoch number values of 20 and 30 showed negligible differences. This suggests that increasing the number of pre-training epochs does not adversely affect the training process; hence, we recommend training for a higher number of epochs when feasible. The selection of the second epoch number varies across datasets based on convergence criteria. For instance, in the GRN dataset, training with the second epoch number set to 10 was sufficient for convergence, leading us to terminate further training. Conversely, for the EMT dataset, convergence was achieved during the first stage, negating the need for additional second epoch number training. Generally, a higher number of epochs in this stage enhances training efficacy. In the case of Gaussian Mixtures, the number of the second epoch number increases with dimensionality. However, for 50D and 100D, the second epoch number decreases because, for 100D, training can extend beyond 120 epochs for improved results, while 50D converges around 100 epochs, making further training unnecessary. Therefore, for the adjustment of the second epoch number, we recommend that users train until convergence based on their specific data requirements.

Table 6: Results of different epoch scheduling during the pre-training stage on gene regulatory dataset. Here we use the arrow to represent the number of epochs trained before(left) and after(right) the $\lambda_m$ resetting, for example, $10 \rightarrow 20$ denoting 10 epochs of training when $\lambda_m = 1$ and 20 epochs of training when $\lambda_m = 0$.

| Epoch Scheduling | $t = 1$ | | $t = 2$ | | $t = 3$ | | $t = 4$ | |
| --- | --- | --- | --- | --- | --- | --- | --- | --- |
| | $\mathcal{W}_1$ | $\mathcal{W}_2$ | $\mathcal{W}_1$ | $\mathcal{W}_2$ | $\mathcal{W}_1$ | $\mathcal{W}_2$ | $\mathcal{W}_1$ | $\mathcal{W}_2$ |
| $10 \rightarrow 10$ | $0.138_{\pm0.003}$ | $0.173_{\pm0.002}$ | $0.243_{\pm0.005}$ | $0.278_{\pm0.004}$ | $0.325_{\pm0.011}$ | $0.353_{\pm0.013}$ | $0.406_{\pm0.014}$ | $0.455_{\pm0.011}$ |
| $20 \rightarrow 10$ | $0.095_{\pm0.005}$ | $0.115_{\pm0.006}$ | $0.104_{\pm0.002}$ | $0.130_{\pm0.007}$ | $0.130_{\pm0.003}$ | $0.157_{\pm0.003}$ | $0.140_{\pm0.007}$ | $0.168_{\pm0.007}$ |
| $30 \rightarrow 10$ | $0.107_{\pm0.008}$ | $0.128_{\pm0.010}$ | $0.120_{\pm0.05}$ | $0.152_{\pm0.08}$ | $0.130_{\pm0.02}$ | $0.164_{\pm0.004}$ | $0.154_{\pm0.012}$ | $0.188_{\pm0.113}$ |

In the training stage, we empirically identified $\lambda_w$ as a hyperparameter requiring tuning. We found that $\lambda_w$ may relate to the noise amplitude $\sigma$. Specifically, for the Gene Regulatory Network (GRN)

and Mouse Hematopoiesis datasets, where $\sigma$ is set to 0.25, a smaller $\lambda_w$ is necessary to maintain computational stability; setting $\lambda_w$ to 1 in these cases resulted in instability. For the remaining datasets, we have found that selecting $\lambda_w$ within the range of [0.1, 1] is feasible, as variations within this range do not significantly impact the training outcomes. In Table 7, we present the results of training with $\lambda_w$ values of 0.1, 0.4, 0.7, and 1.0 on EMT dataset. We computed the values of the trained results ($\boldsymbol{v}(\boldsymbol{x}, t)$, $g$, $s$) at each data point, using the results obtained with $\lambda_w = 1$ as the baseline. The distances between the outcomes of other $\lambda_w$ settings and the baseline were measured using Euclidean distance for inferred $\boldsymbol{v}(\boldsymbol{x}, t)$ and mean squared error (MSE) for $g$ and $s$. As illustrated in Table 7, we observed that as long as $\lambda_w$ ensures computational stability, the outcomes for different $\lambda_w$ values are consistent. Therefore, we recommend selecting a smaller $\lambda_w$, with a default value of 0.01 (for smaller $\sigma$) or 0.001 (for larger $\sigma$), to achieve stable and consistent training results.

Table 7: Results with different $\lambda_w$ settings on the EMT dataset ($\sigma = 0.25$). Distances for $\boldsymbol{v}$ are measured using Euclidean distance, while $g$ and $s$ are evaluated using mean squared error (MSE).

| $\lambda_w$ | $\boldsymbol{v}$ Dist | $g$ MSE | $s$ MSE |
|---|---|---|---|
| 0.1 | $2.64 \times 10^{-8}$ | $6.61 \times 10^{-16}$ | $1.63 \times 10^{-9}$ |
| 0.4 | $2.20 \times 10^{-8}$ | $3.80 \times 10^{-16}$ | $7.23 \times 10^{-10}$ |
| 0.7 | $2.11 \times 10^{-8}$ | $3.46 \times 10^{-16}$ | $1.80 \times 10^{-10}$ |
| 1.0 | $0.00$ | $0.00$ | $0.00$ |

### C.3 TRAINING INITIAL LOG DENSITY FUNCTION

In our gene regulatory network example training, we utilize the following steps to stabilize training. Here since we parameterize $\frac{\sigma^2}{2} \log p(\boldsymbol{x}, t)$ using $s_\theta(\boldsymbol{x}, t)$ when we compute Fokker-Planck constrained loss, we need to do the exponential operations, which may cause numerical instability during computations. To mitigate this issue, we augment the CFM loss with an additional penalty term defined as $L = \alpha_{\text{penalty}} \max(s_\theta(\boldsymbol{x}, t), 0)$ prior to training $s_\theta(\boldsymbol{x}, t)$. This penalty encourages $s_\theta(\boldsymbol{x}, t)$ to adopt negative values, thereby preventing instability caused by the subsequent exponential function. Our training procedure is conducted in two stages:

1. **Initial Training Phase:** We optimize the combined CFM loss and penalty term for 3,000 epochs. This stage ensures that $s$ remains negative, stabilizing the training process by avoiding potential numerical issues associated with positive values of $s$.

2. **Secondary Training Phase:** After the initial phase, we remove the penalty term and continue training solely with the CFM loss for an additional 6,000 epochs. This allows the model to fine-tune the parameter $s$ without the constraint imposed by the penalty, facilitating more accurate learning.

For the last two examples, such stabilization measures are unnecessary, as the dynamics of these cases do not induce the same level of numerical instability. Consequently, we proceed with training using only the CFM loss without incorporating the penalty term.

This issue may be avoided if we use the Hamilton–Jacobi–Bellman (HJB) equations to set the constraints rather than the direct Fokker-Planck equations (Jiang & Wan, 2024; Meng et al., 2024; Zhao et al., 2024; Zhou et al., 2024b).

## D TECHNICAL DETAILS FOR SCHRÖDINGER BRIDGE AND RUOT

### D.1 PROOF OF DYNAMICAL FORMULATION OF SCHRÖDINGER BRIDGE

*Proof.* For Schrödinger bridge problems, the optimal solution can be found within the class of SDEs:
$$\boldsymbol{X}_t \sim \mu_t^{\boldsymbol{X}}: \quad \mathrm{d}\boldsymbol{X}_t = \boldsymbol{b}(\boldsymbol{X}_t, t)\,\mathrm{d}t + \boldsymbol{\sigma}(\boldsymbol{X}_t, t)\,\mathrm{d}\boldsymbol{W}_t,$$
The Fokker-Planck equation for the SDE is
$$\partial_t p(\boldsymbol{x}, t) = -\nabla_{\boldsymbol{x}} \cdot (p(\boldsymbol{x}, t)\boldsymbol{b}(\boldsymbol{x}, t)) + \frac{1}{2}\nabla_{\boldsymbol{x}}^2 : (\boldsymbol{a}(\boldsymbol{x}, t)p(\boldsymbol{x}, t)),$$

where $\boldsymbol{a}(\boldsymbol{x}, t) = \boldsymbol{\sigma}(\boldsymbol{x}, t)\boldsymbol{\sigma}^T(\boldsymbol{x}, t)$. Then we need to compute the KL divergence of these two stochastic processes. Define the auxiliary variable

$$\gamma(\boldsymbol{x}, t) = \boldsymbol{\sigma}(\boldsymbol{x}, t)^{-1}\boldsymbol{b}(\boldsymbol{x}, t)$$

and apply the Girsanov's theorem, we get

$$\mathcal{D}_{\mathrm{KL}}\left(\mu_{[0,1]}^{\boldsymbol{X}} || \mu_{[0,1]}^{\boldsymbol{Y}}\right) = \mathbb{E}_{\mathbb{P}}\left[\mathcal{E}_1\left(\int_0^1 \gamma(\boldsymbol{Y}_t, t)\mathrm{d}\boldsymbol{W}_t - \frac{1}{2}\int_0^1 \|\gamma(\boldsymbol{Y}_t, t)\|_2^2 \,\mathrm{d}t\right)\right], \qquad (14)$$

where $\{\mathcal{E}_t\}_{t \in [0,1]}$ is an exponential martingale with respect to the Brownian filtrations $\{\mathcal{F}_t\}$ defined as

$$\mathcal{E}_t = \exp\left(\int_0^t \gamma(\boldsymbol{Y}_t, t)\mathrm{d}\boldsymbol{W}_t - \frac{1}{2}\int_0^t \|\gamma(\boldsymbol{Y}_t, t)\|_2^2 \,\mathrm{d}t\right).$$

It is well-known that the exponential martingale $\mathcal{E}_t$ satisfies the SDE

$$\mathrm{d}\mathcal{E}_t = \mathcal{E}_t \gamma(\boldsymbol{Y}_t, t)\mathrm{d}\boldsymbol{W}_t, \quad \mathcal{E}_0 = 1.$$

So we obtain

$$\mathcal{E}_1 = \int_0^1 \mathcal{E}_s \gamma(\boldsymbol{Y}_s, s)\mathrm{d}\boldsymbol{W}_s + 1.$$

Since $\mathbb{E}\int_0^1 \gamma(\boldsymbol{Y}_t, t)\mathrm{d}\boldsymbol{W}_t = 0$, we get

$$\mathbb{E}\left[\mathcal{E}_1 \int_0^1 \gamma(\boldsymbol{Y}_t, t)\mathrm{d}\boldsymbol{W}_t\right] = \mathbb{E}\left[\int_0^1 \mathcal{E}_t \gamma(\boldsymbol{Y}_t, t)\mathrm{d}\boldsymbol{W}_t \cdot \int_0^1 \gamma(\boldsymbol{Y}_t, t)\mathrm{d}\boldsymbol{W}_t\right],$$

$$= \mathbb{E}\int_0^1 \mathcal{E}_t \|\gamma(\boldsymbol{Y}_t, t)\|_2^2 \,\mathrm{d}t,$$

$$= \mathbb{E}\int_0^1 \mathbb{E}\left(\|\gamma(\boldsymbol{Y}_t, t)\|_2^2 \mathcal{E}_1 \mid \mathcal{F}_t\right)\mathrm{d}t,$$

$$= \int_0^1 \mathbb{E}\left(\mathcal{E}_1 \|\gamma(\boldsymbol{Y}_t, t)\|_2^2\right)\mathrm{d}t,$$

$$= \mathbb{E}\left[\mathcal{E}_1 \int_0^1 \|\gamma(\boldsymbol{Y}_t, t)\|_2^2 \,\mathrm{d}t\right].$$

Combine with Eq. (14), we obtain that

$$\mathcal{D}_{\mathrm{KL}}(\mu_{[0,1]}^{\boldsymbol{X}} || \mu_{[0,1]}^{\boldsymbol{Y}}) = \mathbb{E}\left[\mathcal{E}_1\left(\int_0^1 \frac{1}{2}\|\gamma(\boldsymbol{Y}_t, t)\|_2^2 \,\mathrm{d}t\right)\right],$$

$$= \mathbb{E}\int_0^1 \frac{1}{2}\|\gamma(\boldsymbol{X}_t, t)\|_2^2 \,\mathrm{d}t,$$

$$= \int_0^1 \int_{\mathbb{R}^d}\left[\frac{1}{2}\boldsymbol{b}^T(\boldsymbol{x}, t)\boldsymbol{a}(\boldsymbol{x}, t)^{-1}\boldsymbol{b}(\boldsymbol{x}, t)\right]p(\boldsymbol{x}, t)\mathrm{d}\boldsymbol{x}\mathrm{d}t.$$

The proof is done. $\qquad\qquad\qquad\qquad\qquad\qquad\qquad\qquad\qquad\qquad\qquad\qquad\qquad\qquad\square$

## D.2 PROOF OF FISHER REGULARIZATION OF RUOT

*Proof.* From (6) we obtain

$$\partial_t p = -\nabla_{\boldsymbol{x}} \cdot (p\boldsymbol{b}) + \frac{1}{2}\nabla_{\boldsymbol{x}}^2 : \left(\sigma^2(t)\boldsymbol{I}p\right) + g(\boldsymbol{x}, t)p,$$

$$= -\nabla_{\boldsymbol{x}} \cdot \left(\left(\boldsymbol{b} - \frac{1}{2}\sigma^2(t)\nabla_{\boldsymbol{x}}\log p\right)p\right) + g(\boldsymbol{x}, t)p.$$

Using the change of variable $\boldsymbol{v}(\boldsymbol{x}, t) = \boldsymbol{b}(\boldsymbol{x}, t) - \frac{1}{2}\sigma^2(t)\nabla_{\boldsymbol{x}}\log p$, we see that it is equivalent to

$$\partial_t p = -\nabla_{\boldsymbol{x}} \cdot (p\boldsymbol{v}(\boldsymbol{x}, t)) + g(\boldsymbol{x}, t)p.$$

Correspondingly, the integrand in the objective functional becomes

$$\int_0^1 \int_{\mathbb{R}^d} \left[ \frac{1}{2} \|\boldsymbol{v}(\boldsymbol{x},t)\|_2^2 + \frac{\sigma^4(t)}{8} \|\nabla_{\boldsymbol{x}} \log p\|_2^2 + \frac{1}{2} \langle \boldsymbol{v}(\boldsymbol{x},t), \sigma^2(t) \nabla_{\boldsymbol{x}} \log p \rangle + \alpha \Psi(g) \right] p(\boldsymbol{x},t) \mathrm{d}\boldsymbol{x} \mathrm{d}t. \tag{15}$$

Letting $H(p(\boldsymbol{x},t)) := \int_{\mathbb{R}^d} p(\boldsymbol{x},t) \log p(\boldsymbol{x},t) \mathrm{d}\boldsymbol{x}$ be the entropy, we have

$$\sigma^2(1) H(p(\boldsymbol{x},1)) - \sigma^2(0) H(p(\boldsymbol{x},0)) = \int_0^1 \partial_t \left( \sigma^2(t) H(p(\boldsymbol{x},t)) \right) \mathrm{d}t,$$

$$= \int_0^1 \int_{\mathbb{R}^d} \frac{\mathrm{d}\sigma^2(t)}{\mathrm{d}t} p(\boldsymbol{x},t) \log p(\boldsymbol{x},t) + \sigma^2(t) (1 + \log p(\boldsymbol{x},t)) \partial_t p(\boldsymbol{x},t) \mathrm{d}\boldsymbol{x} \mathrm{d}t,$$

$$= \int_0^1 \int_{\mathbb{R}^d} \sigma^2(t) (1 + \log p(\boldsymbol{x},t)) \cdot (-\nabla_{\boldsymbol{x}} \cdot (p(\boldsymbol{x},t) \boldsymbol{v}(\boldsymbol{x},t)) + g p(\boldsymbol{x},t)) + \frac{\mathrm{d}\sigma^2(t)}{\mathrm{d}t} p(\boldsymbol{x},t) \log p(\boldsymbol{x},t) \mathrm{d}\boldsymbol{x} \mathrm{d}t,$$

$$= \int_0^1 \int_{\mathbb{R}^d} \sigma^2(t) p(\boldsymbol{x},t) \langle \nabla_{\boldsymbol{x}} \log p(\boldsymbol{x},t), \boldsymbol{v}(\boldsymbol{x},t) \rangle + \sigma^2(t) (1 + \log p(\boldsymbol{x},t)) g p + \frac{\mathrm{d}\sigma^2(t)}{\mathrm{d}t} p(\boldsymbol{x},t) \log p(\boldsymbol{x},t) \mathrm{d}\boldsymbol{x} \mathrm{d}t.$$

Therefore,

$$\int_0^1 \int_{\mathbb{R}^d} \left[ \langle \sigma^2(t) \nabla_{\boldsymbol{x}} \log p, \boldsymbol{v}(\boldsymbol{x},t) \rangle \right] p(\boldsymbol{x},t) \mathrm{d}\boldsymbol{x} \mathrm{d}t = \left( \sigma^2(1) H(p(\boldsymbol{x},1)) - \sigma^2(0) H(p(\boldsymbol{x},0)) \right)$$

$$- \int_0^1 \int_{\mathbb{R}^d} \sigma^2(t) (1 + \log p(\boldsymbol{x},t)) g p \, \mathrm{d}\boldsymbol{x} \mathrm{d}t$$

$$- \int_0^1 \int_{\mathbb{R}^d} \frac{\mathrm{d}\sigma^2(t)}{\mathrm{d}t} p(\boldsymbol{x},t) \log p(\boldsymbol{x},t) \mathrm{d}\boldsymbol{x} \mathrm{d}t.$$

$\square$

### D.3 RUOT WITH NONLINEAR FOKKER-PLANCK EQUATION CONSTRAINTS

Note that when $\Psi(g(\boldsymbol{x},t))$ take the quadratic form i.e. $\Psi(g(\boldsymbol{x},t)) = |g(\boldsymbol{x},t)|_2^2$, then by canceling the cross term in Theorem 4.1 we will get another Fisher information regularisation form as below.

**Theorem D.1.** *When $\Psi(g(\boldsymbol{x},t)) = |g(\boldsymbol{x},t)|_2^2$ and $\sigma(t) = \sigma$ is constant, the regularized unbalanced optimal transport problem* (7) *is equivalent to*

$$\inf_{(p,\boldsymbol{v},\widetilde{g})} \int_0^1 \int_{\mathbb{R}^d} \left[ \frac{1}{2} \|\boldsymbol{v}(\boldsymbol{x},t)\|_2^2 + \frac{\sigma^4}{8} \|\nabla_{\boldsymbol{x}} \log p\|_2^2 + \alpha |\widetilde{g}(\boldsymbol{x},t)|^2 - \frac{\sigma^4}{16\alpha} (1 + \log p)^2 \right] p \, \mathrm{d}\boldsymbol{x} \mathrm{d}t, \tag{16}$$

*where the infimum is taken all pairs $(p, \boldsymbol{v}, \widetilde{g})$ such that $p(\cdot, 0) = \nu_0, p(\cdot, 1) = \nu_1, p(\boldsymbol{x},t)$ absolutely continuous, and*

$$\partial_t p = -\nabla_{\boldsymbol{x}} \cdot (p \boldsymbol{v}(\boldsymbol{x},t)) + \widetilde{g}(\boldsymbol{x},t) p + \frac{\sigma^2}{4\alpha} (1 + \log p) p. \tag{17}$$

*coupled with vanishing boundary condition:* $\lim_{|\boldsymbol{x}| \to \infty} p(\boldsymbol{x},t) = 0.$

*Proof.* Based on the formulation of Theorem 4.1, we proceed by introducing the following change of variables:

$$\widetilde{g} = g - \frac{\sigma^2}{4\alpha} (1 + \log p). \tag{18}$$

Substituting this variable into Eq. (7), then it follows immediately. $\square$

From Theorem D.1, we can find that if we introduce a new term in the continuity equation Eq. (6) as:

$$\partial_t p = -\nabla_{\boldsymbol{x}} \cdot (p \boldsymbol{b}) + \frac{1}{2} \nabla_{\boldsymbol{x}}^2 : \left( \sigma^2 \boldsymbol{I} p \right) + g p - \frac{\sigma^2}{4\alpha} (1 + \log p) p.$$

Then Eq. (17) now yields the standard continuity equation:

$$\partial_t p = -\nabla_{\boldsymbol{x}} \cdot (p \boldsymbol{v}(\boldsymbol{x},t)) + \widetilde{g}(\boldsymbol{x},t) p.$$

So we introduce a new definition.

**Definition D.1.** *We introduce*

$$\mathrm{UOT}^{\mathrm{D}}\left(\nu_0, \nu_1\right) := \inf_{(p, \boldsymbol{b}, g) \in \mathcal{C}(\nu_0, \nu_1)} \int_0^1 \int_{\mathbb{R}^d} \frac{1}{2} \|\boldsymbol{b}(\boldsymbol{x}, t)\|_2^2 \, p \mathrm{d}\boldsymbol{x} \mathrm{d}t + \int_0^1 \int_{\mathbb{R}^d} \alpha \left|g(\boldsymbol{x}, t)\right|_2^2 p \mathrm{d}\boldsymbol{x} \mathrm{d}t,$$

*where, for $(p, \boldsymbol{b}, g)$ belonging to appropriate functions spaces,*

$$\mathcal{C}\left(\nu_0, \nu_1\right) := \left\{p, \boldsymbol{b}, g \,\middle|\, \partial_t p = -\nabla_{\boldsymbol{x}} \cdot (p\boldsymbol{b}) + \frac{1}{2}\nabla_{\boldsymbol{x}}^2 : (\sigma^2 \boldsymbol{I} p) + gp - \frac{\sigma^2}{4\alpha}(1 + \log p)p \middle|\right\},$$

*where $p_0 = \nu_0$ and $p_1 = \nu_1$.*

**Remark D.1.** *Note that Definition D.1 is consistent with the form provided in (Buze & Duong, 2023), and it has been conjectured that it is the higher order approximation scheme to the unbalanced optimal transport problem as $\sigma \to 0$.*

### D.4 RUOT IN THE GENERAL CASE

For the general case, we consider the following equation

$$\partial_t p = -\nabla_{\boldsymbol{x}} \cdot (p\boldsymbol{b}) + \frac{1}{2}\nabla_{\boldsymbol{x}}^2 : (\boldsymbol{a}(\boldsymbol{x}, t)p) + gp, \tag{19}$$

with the corresponding metric

$$\int_0^1 \int_{\mathbb{R}^d} \frac{1}{2} \|\boldsymbol{b}(\boldsymbol{x}, t)\|_2^2 \, p(\boldsymbol{x}, t)\mathrm{d}\boldsymbol{x}\mathrm{d}t + \int_0^1 \int_{\mathbb{R}^d} \alpha \Psi\left(g(\boldsymbol{x}, t)\right) p(\boldsymbol{x}, t)\mathrm{d}\boldsymbol{x}\mathrm{d}t, \tag{20}$$

where $\alpha$ and $\Psi : \mathbb{R} \to [0, +\infty]$ is the growth penalization. Then we can define the *regularized unbalanced optimal transport* problem in the same way.

**Definition D.2** (Regularized unbalanced optimal transport II). *Consider*

$$\inf_{(p, \boldsymbol{b}, g)} \int_0^1 \int_{\mathbb{R}^d} \frac{1}{2} \|\boldsymbol{b}(\boldsymbol{x}, t)\|_2^2 \, p(\boldsymbol{x}, t)\mathrm{d}\boldsymbol{x}\mathrm{d}t + \int_0^1 \int_{\mathbb{R}^d} \alpha \Psi\left(g(\boldsymbol{x}, t)\right) p(\boldsymbol{x}, t)\mathrm{d}\boldsymbol{x}\mathrm{d}t, \tag{21}$$

*where the infimum is taken all pairs $(p, \boldsymbol{b})$ such that $p(\cdot, 0) = \nu_0, p(\cdot, 1) = \nu_1, p(\boldsymbol{x}, t)$ absolutely continuous, and*

$$\partial_t p = -\nabla_{\boldsymbol{x}} \cdot (p\boldsymbol{b}) + \frac{1}{2}\nabla_{\boldsymbol{x}}^2 : (\boldsymbol{a}(\boldsymbol{x}, t)p) + gp. \tag{22}$$

*coupled with vanishing boundary condition:* $\lim_{|\boldsymbol{x}| \to \infty} p(\boldsymbol{x}, t) = 0.$

We similarly can reformulate Definition D.2 with the following Fisher information regularization.

**Theorem D.2.** *Consider the regularized unbalanced optimal transport problem Definition D.2, it is equivalent to*

$$\inf_{(p, \boldsymbol{v}, g)} \int_0^1 \int_{\mathbb{R}^d} \frac{1}{2} \|\boldsymbol{v}(\boldsymbol{x}, t)\|_2^2 \, p(\boldsymbol{x}, t)\mathrm{d}\boldsymbol{x}\mathrm{d}t + \frac{1}{8} \|\boldsymbol{\sigma}(\boldsymbol{x}, t)\nabla_{\boldsymbol{x}} \cdot \log\left(\boldsymbol{a}(\boldsymbol{x}, t)p\right)\|_2^2 \, p(\boldsymbol{x}, t)\mathrm{d}\boldsymbol{x}\mathrm{d}t$$
$$+ \frac{1}{2} \langle\boldsymbol{v}(\boldsymbol{x}, t), \boldsymbol{a}(\boldsymbol{x}, t)\nabla_{\boldsymbol{x}} \cdot \log\left(\boldsymbol{a}(\boldsymbol{x}, t)p\right)\rangle \, p(\boldsymbol{x}, t)\mathrm{d}\boldsymbol{x}\mathrm{d}t + \alpha\Psi\left(g(\boldsymbol{x}, t)\right) p(\boldsymbol{x}, t)\mathrm{d}\boldsymbol{x}\mathrm{d}t, \tag{23}$$

*where the infimum is taken all pairs $(p, \boldsymbol{v}, g)$ such that $p(\cdot, 0) = \nu_0, p(\cdot, 1) = \nu_1, p(\boldsymbol{x}, t)$ absolutely continuous, and*

$$\partial_t p = -\nabla_{\boldsymbol{x}} \cdot (p\boldsymbol{v}(\boldsymbol{x}, t)) + g(\boldsymbol{x}, t)p. \tag{24}$$

*coupled with vanishing boundary condition:* $\lim_{|\boldsymbol{x}| \to \infty} p(\boldsymbol{x}, t) = 0.$

*Proof.* From Definition D.2 we know that the Fokker-Planck equation for the SDE is

$$\partial_t p = -\nabla_{\boldsymbol{x}} \cdot (p\boldsymbol{b}) + \frac{1}{2}\nabla_{\boldsymbol{x}}^2 : (\boldsymbol{a}(\boldsymbol{x}, t)p) + g(\boldsymbol{x}, t)p,$$
$$= -\nabla_{\boldsymbol{x}} \cdot \left(\left(\boldsymbol{b} - \frac{1}{2}\boldsymbol{a}(\boldsymbol{x}, t)\nabla_{\boldsymbol{x}} \cdot \log\left(\boldsymbol{a}(\boldsymbol{x}, t)p\right)\right)p\right) + g(\boldsymbol{x}, t)p.$$

with minimization problem

$$\inf_{(p,\boldsymbol{b})} \int_0^1 \int_{\mathbb{R}^d} \frac{1}{2} \|\boldsymbol{b}(\boldsymbol{x},t)\|_2^2 \, p(\boldsymbol{x},t) \mathrm{d}\boldsymbol{x}\mathrm{d}t + \int_0^1 \int_{\mathbb{R}^d} \alpha \Psi\left(g(\boldsymbol{x},t)\right) p(\boldsymbol{x},t) \mathrm{d}\boldsymbol{x}\mathrm{d}t.$$

Using the change of variable $\boldsymbol{v}(\boldsymbol{x},t) = \boldsymbol{b}(\boldsymbol{x},t) - \frac{1}{2}\boldsymbol{a}(\boldsymbol{x},t)\nabla_{\boldsymbol{x}} \cdot \log\left(\boldsymbol{a}(\boldsymbol{x},t)p\right)$, we see that it is equivalent to

$$\partial_t p = -\nabla_{\boldsymbol{x}} \cdot (p\boldsymbol{v}(\boldsymbol{x},t)) + g(\boldsymbol{x},t)p.$$

On the other hand, since $\|\boldsymbol{b}(\boldsymbol{x},t)\|_2^2 = \|\boldsymbol{v}(\boldsymbol{x},t)\|_2^2 + \frac{1}{4}\|\boldsymbol{a}(\boldsymbol{x},t)\nabla_{\boldsymbol{x}} \cdot \log\left(\boldsymbol{a}(\boldsymbol{x},t)p\right)\|_2^2 + 2\left\langle \boldsymbol{v}(\boldsymbol{x},t), \frac{1}{2}\boldsymbol{a}(\boldsymbol{x},t)\nabla_{\boldsymbol{x}} \cdot \log\left(\boldsymbol{a}(\boldsymbol{x},t)p\right)\right\rangle$, the integrand in the objective then becomes

$$\int_0^1 \int_{\mathbb{R}^d} \frac{1}{2} \|\boldsymbol{v}(\boldsymbol{x},t)\|_2^2 \, p(\boldsymbol{x},t)\mathrm{d}\boldsymbol{x}\mathrm{d}t + \frac{1}{8}\|\boldsymbol{\sigma}(\boldsymbol{x},t)\nabla_{\boldsymbol{x}} \cdot \log\left(\boldsymbol{a}(\boldsymbol{x},t)p\right)\|_2^2 \, p(\boldsymbol{x},t)\mathrm{d}\boldsymbol{x}\mathrm{d}t$$

$$+ \frac{1}{2}\left\langle \boldsymbol{v}(\boldsymbol{x},t), \boldsymbol{a}(\boldsymbol{x},t)\nabla_{\boldsymbol{x}} \cdot \log\left(\boldsymbol{a}(\boldsymbol{x},t)p\right)\right\rangle p(\boldsymbol{x},t)\mathrm{d}\boldsymbol{x}\mathrm{d}t + \alpha\Psi\left(g(\boldsymbol{x},t)\right) p(\boldsymbol{x},t)\mathrm{d}\boldsymbol{x}\mathrm{d}t,$$

$\square$

Similarly, from the proof of Theorem D.2, the original SDE $\mathrm{d}\boldsymbol{X}_t = \left(\boldsymbol{b}\left(\boldsymbol{X}_t,t\right)\right)\mathrm{d}t + \boldsymbol{\sigma}\left(\boldsymbol{X}_t,t\right)\mathrm{d}\boldsymbol{W}_t$ can be transformed into the probability flow ODE

$$\mathrm{d}\boldsymbol{X}_t = \underbrace{\left(\boldsymbol{b}\left(\boldsymbol{X}_t,t\right) - \frac{1}{2}\boldsymbol{a}(\boldsymbol{X}_t,t)\nabla_{\boldsymbol{x}} \cdot \log\left(\boldsymbol{a}(\boldsymbol{X}_t,t)\boldsymbol{p}\left(\boldsymbol{X}_t,t\right)\right)\right)}_{\boldsymbol{v}(\boldsymbol{X}_t,t)} \mathrm{d}t.$$

Conversely, if the probability flow ODE's drift $\boldsymbol{v}(\boldsymbol{x},t)$ the diffusion rate $\boldsymbol{a}(\boldsymbol{x},t)$ and the function $\nabla_{\boldsymbol{x}} \cdot \log\left(\boldsymbol{a}(\boldsymbol{x},t)p(\boldsymbol{x},t)\right)$ are known, then the the SDE's drift term $b(\boldsymbol{x},t)$ can be determined by

$$b(\boldsymbol{x},t) = \boldsymbol{v}(\boldsymbol{x},t) + \frac{1}{2}\boldsymbol{a}(\boldsymbol{x},t)\nabla_{\boldsymbol{x}} \cdot \log\left(\boldsymbol{a}(\boldsymbol{x},t)p(\boldsymbol{x},t)\right).$$

So to specify an SDE is equivalent to specifying the probability flow ODE and the corresponding log density function.

## E    CONNECTIONS WITH SCHRÖDINGER BRIDGE

Although the RUOT problem can be derived from the dynamical formulation of a Schrödinger bridge relaxation, the relationship between the RUOT problem and the original static Schrödinger bridge problem, the choice of the growth penalty form in RUOT as well as its microscopic interpretation, remains unsolved. In this section, we will discuss two potential microscopic interpretations with the choice of the growth penalty form and identify the persisting problems and challenges associated with each.

### E.1    CONNECTIONS WITH BRANCHING SCHRÖDINGER BRIDGE

In (Baradat & Lavenant, 2021), the authors explore the relationship between the RUOT and the branching Schrödinger bridge, with the corresponding reference process identified as branching Brownian motion. We will begin by reviewing their principal findings and subsequently highlight the associated issues and challenges.

A Branching Brownian Motion (BBM) can be formally described by the diffusivity $\varepsilon$, branching rate $\lambda$ (Each particle is associated with an independent exponential distribution with parameter $\lambda$), and the offspring distribution denoted by $\boldsymbol{p} = (p_k)_{k\in\mathbb{N}} \in \mathcal{P}(\mathbb{N})$ (The probability measures on $\mathbb{N}$), where $p_k$ is the probability of producing $k$ offspring at a branching event. We define $q_k = \lambda p_k$ and introduce the generating function of $\boldsymbol{q}$ as $\psi_{\boldsymbol{q}}$.

$$\psi_{\boldsymbol{q}}(z) := \sum_{k\in\mathbb{N}} q_k z^k = \lambda \sum_{k\in\mathbb{N}} p_k z^k.$$

The branching mechanism is characterized by $\boldsymbol{q}$, since $\sum_{k \in \mathbb{N}} p_k = 1$ and $\lambda$, $\boldsymbol{p}$ can be recovered from $\boldsymbol{q}$ by $\lambda = \sum_{k \in \mathbb{N}} q_k$ and $p_k = \lambda^{-1} q_k$ for $k \in \mathbb{N}$. Therefore, it is sufficient to fully determine the BBM with $\nu$, $\boldsymbol{q} = \lambda \boldsymbol{p}$, and initial distribution $\nu_0$. So $R \sim \mathrm{BBM}(\varepsilon, \boldsymbol{q}, \nu_0)$ represent such BBM. Their core result indicates that consider Definition 4.1 where $\sigma(t) = \sigma$ is constant and $\varepsilon = \sigma^2$ then if the growth penalty term is defined as follows, it can correspond to a branching Schrödinger process.

$$\Psi^*_{\varepsilon, \boldsymbol{q}}(s) = \varepsilon \left( \psi_{\boldsymbol{q}} \left( e^{s/\varepsilon} \right) e^{-s/\varepsilon} - \psi_{\boldsymbol{q}}(1) \right) = \varepsilon \sum_{k=0}^{+\infty} q_k \left\{ \exp \left( (k-1) \frac{s}{\varepsilon} \right) - 1 \right\}, \qquad (25)$$

and we define $\Psi_{\varepsilon, \boldsymbol{q}}(g) = \sup_{s \in \mathbb{R}} gs - \Psi^*_{\varepsilon, \boldsymbol{q}}(s)$. So by following their result, we can present some examples.

**The case of only loss mass** This example has been included in their work. We consider $\varepsilon = \lambda = 1$, $p_0 = 1$, and $\sum_{k \geq 2} p_k = 0$, which means the particle can only die at the branching event. Substituting it into Eq. (25), we have

$$\Psi^*_{1, \boldsymbol{q}}(s) = \exp(-s) - 1.$$

Then by computing its Legendre transform we obtain

$$\Psi_{1, \boldsymbol{q}}(g) = 1 + g - g \log(-g),$$

where $g < 0$. This form is consistent with the formula discussed in (Chen et al., 2022b).

**The case of only gain mass** Similarly, if we consider $\varepsilon = \lambda = 1$, $p_2 = 1$, which means the particle can only divide into two parts at the branching event. Then by a similar calculation, we obtain

$$\Psi_{1, \boldsymbol{q}}(g) = 1 - g + g \log(g),$$

which $g > 0$.

**The case of both gain mass and loss mass** This is also presented in their work. We consider $\varepsilon = \lambda = 1$, $p_0 = p_2 = \frac{1}{2}$, which means the particle can divide into two parts or die at the same probability at the branching event, which we have

$$\Psi^*_{1, \boldsymbol{q}}(s) = \cosh(s) - 1.$$

However in this case the Legendre transform would be more complicated but it has some asymptotic properties discussed in their work.

**General case** Although we have illustrated several examples, challenges persist, particularly in the context of more general scenarios. Specifically, when the growth penalty term cannot be expressed as $\Psi_{\varepsilon, \boldsymbol{q}}$, the connection to the corresponding stochastic process remains unclear. For instance, this ambiguity arises in cases where in the RUOT the $\Psi(g)$ takes the quadratic form i.e., the WFR metric, or when $\Psi(g)$ is in a linear form, such as $|g|$. These issues still require specific mathematical treatment and to our knowledge, remain an open problem.

## E.2 CONNECTIONS WITH SCHRÖDINGER BRIDGE ON WEIGHTED PATH

From an alternative perspective on the microscopic interpretation of RUOT, we aim to consider particles on the weighted path space, namely:

$$\mathrm{d}\mu^{\boldsymbol{G}}_{[0,1]} = \exp \left( \int_0^1 g(\boldsymbol{X}_s, s) \mathrm{d}s \right) \mathrm{d}\mu^{\boldsymbol{X}}_{[0,1]},$$

where $\mu^{\boldsymbol{X}}_{[0,1]}$ represents the probability measure induced by the SDE (Eq. (2)). Here for example we consider a simple case when $g(\boldsymbol{x}, t) = C$. Then following the procedure of Theorem 3.1.

$$\mathcal{D}_{\mathrm{KL}}(\mu^{\boldsymbol{G}}_{[0,1]} \| \mu^{\boldsymbol{Y}}_{[0,1]}) = \mathbb{E}_{\mu^{\boldsymbol{G}}_{[0,1]}} \left( \int_0^1 \gamma(\boldsymbol{Y}_t, t) \mathrm{d}\boldsymbol{W}_t - \frac{1}{2} \| \gamma(\boldsymbol{Y}_t, t) \|_2^2 \, \mathrm{d}t + C \right),$$

$$= \mathbb{E} \left( \mathcal{E}_1 \exp(C) \right) \left( \int_0^1 \frac{1}{2} \| \gamma(\boldsymbol{Y}_t, t) \|_2^2 \, \mathrm{d}t \right) + C \exp(C), \qquad (26)$$

$$= \mathbb{E} \exp(C) \int_0^1 \frac{1}{2} \| \gamma(\boldsymbol{X}_t, t) \|_2^2 \, \mathrm{d}t + C \exp(C).$$

Here $\mu_{[0,1]}^{\boldsymbol{Y}}$ is the distribution of stochastic process induced by $\mathrm{d}\boldsymbol{Y}_t = \boldsymbol{\sigma}(\boldsymbol{Y}_t, t)\mathrm{d}\boldsymbol{W}_t$. Our derivation reveals that, from a path weighting perspective, even when the growth term is a constant, it still does not correspond to the form of RUOT. This suggests that the microscopic interpretation from this angle may need further investigation to determine whether a RUOT problem could be recovered from this starting point.

