# OpenReview forum: "Learning stochastic dynamics from snapshots through regularized unbalanced optimal transport"
_ICLR.cc/2025/Conference — ICLR 2025 Oral_

### Official Review · Reviewer_FFk6 · 2024-11-03

**Soundness:** 3
**Presentation:** 2
**Contribution:** 2
**Rating:** 8
**Confidence:** 5

**Summary:**

This paper introduces DeepRUOT, a method for reconstructing dynamics from snapshots using regularized unbalanced optimal transport. This work tackles both the unbalanced and stochastic settings simultaneously, which few works have attempted thus far. An algorithm is proposed to learn drift, score, and growth functions from snapshot data including both synthetic and single-cell transcriptomics data.

**Strengths:**

* Provided code provides additional clarity on experimental setting.
* Tackles an important problem in trajectory inference from population level data.

**Weaknesses:**

* I found it extremely difficult to understand the multistage training procedure for DeepRUOT. There are some stages described in the main text, but then initial log density training also in the appendix? While there are a number of losses and initializations proposed to stabilize training, it is unclear to me what effects these have on final performance and how difficult RUOT is to train in practice. The paper would be much stronger if it was made clear which parts of the training procedure are important practically potentially through an ablation study on different components of the training procedure.
* For a largely empirical work proposing a new algorithm to tackle this problem, the empirical studies are limited. The DeepRUOT is only benchmarked against balanced transport methods. It would greatly strengthen this paper if it was also compared to other unbalanced transport methods such as Action matching [Neklyudov et al. 2023] and unbalanced diffusion Schrödinger bridge [Pariset et al. 2023]. It would also be very useful to know how the many different weighting parameters affect performance, and how sensitive training is to them.
* It is difficult to tell what parts of the theory are novel and which are minor adaptations from prior work. Theorem 4.1 seems like a subclass of Baradat & Lavenant for the Fischer information case. I don’t believe varying sigma(t) over time changes anything theoretically and as far as I understand it is not used in practice.

**Questions:**

It is unclear to me exactly what the computational / numerical cost for allowing varying growth rates is in this framework. From my understanding both the deterministic and stochastic setting are extremely efficient, but it has thus far been numerically challenging to incorporate varying growth rates. How scalable is DeepRUOT? Can it be applied to the higher dimensional settings (20,50,100,1000D) in previous works?

I don’t understand the motivation behind the reconstruction loss R_d. What is its purpose and why does it correspond to a reconstruction loss vs. say an OT loss?

Comments:

\Psi(g) is not explicitly defined (although is fairly clear from context).

> So to specify an SDE is equivalent to specify the probability flow
ODE and its score function.

Probably missing an article here.

### Overall

Overall this work has great potential, however I believe further empirical study would greatly improve its value to the community. I would be eager to reconsider my evaluation given further empirical study relative to existing unbalanced dynamic OT methods and a better empirical understanding of the various components of training and loss weighting.

---

> ### Author Response · Authors · 2024-11-20
> **Response to Reviewer FFk6 (1/5)**
>
> > This paper introduces DeepRUOT, a method for reconstructing dynamics from snapshots using regularized unbalanced optimal transport. This work tackles both the unbalanced and stochastic settings simultaneously, which few works have attempted thus far. An algorithm is proposed to learn drift, score, and growth functions from snapshot data including both synthetic and single-cell transcriptomics data.
>
> > Strengths:
> Provided code provides additional clarity on experimental setting.
> Tackles an important problem in trajectory inference from population level data.
>
> - Thank you for your careful reading and insightful suggestions. Thank you for your clear summary and positive comments.  Trajectory inference is indeed a pivotal area in both machine learning and single-cell biology. In our study, we address the challenge of learning stochastic dynamics from snapshots while accounting for unbalanced effects—an aspect that is critically important in both single-cell analysis and broader machine learning applications. We believe that DeepRUOT offers a flexible tool for modeling and inferring continuous unbalanced stochastic dynamics from observed snapshots without requiring prior knowledge. Your recognition of the importance and innovation of our work is greatly appreciated.
>
>
> - Below you will find our detailed point-by-point responses. We have revised our manuscript accordingly to address the comments, and all changes made to the manuscript are indicated in **blue** throughout the revised main text and appendix.   The major revisions we have made are listed as follows:
>
>   - We have **restructured the explanations within Section 5** to better introduce and motivate the overall loss function (Section 5.1& Section 5.2 & Section 5.3, Page 5-7, Line 266-377).
>   - We have **revised the description of our algorithm** to make it more accessible and easier to understand (Section 5.3 & Algorithm 1, Page 7, Line 334-377; Appendix A & Algorithm 2, Page 16-17, Line 835-844, 864-886).
>   - We have **included ablation studies** to systematically evaluate the contributions of different components of our model (Appendix B.5 & Table 4, Page 23, Line 1190-1226).
>   - We have **expanded our experimental comparisons** to include **five** baseline algorithms, specifically Balanced OT, Unbalanced OT, Balanced SB,  Unbalanced Action Matching (Unbalanced AM) and Unbalanced SB (Section 6 & Table 1, Page 8, Line 423-428, 432-439; Table 2, Page 10, Line 487-494; Appendix B.1 & Figure 4, Page 18, Line 966-992; Appendix B.3, Page 19, Line 1019-1025; Figure 7, Page 21, Line 1095-1128).
>   - We have **expanded the discussion on loss function weighting** (Appendix C.2, Page 24, Line 1265-1280; Table 5, Page 25, Line 1296-1316). We provide detailed values (Table 5, Page 25, Line 1296-1316) and the rationale behind the selection of these weights (Page 24, Line 1270-1276).
>   - We have **included the discussion of scability** in the revised manuscript (Page 19, Line 1006-1017).
>   - We have **extended our tests to Gaussian mixture models in 20D, 50D, and 100D** (Section B.2, Page 19, Line 1002-1005; Figure 6, Page 20, Line 1054-1079).
>
>
>
>
>
> Thank you once again for your insightful review and encouraging feedback.
>
> > Weaknesses:
> I found it extremely difficult to understand the multistage training procedure for DeepRUOT. There are some stages described in the main text, but then initial log density training also in the appendix? While there are a number of losses and initializations proposed to stabilize training, it is unclear to me what effects these have on final performance and how difficult RUOT is to train in practice. The paper would be much stronger if it was made clear which parts of the training procedure are important practically potentially through ablation studies on different components of the training procedure.
>
>
> - Thank you for the insightful suggestions. We apologize for any confusion caused by the original presentation of our methodology.
> To address your concerns, **we have undertaken a comprehensive restructuring of our manuscript to enhance clarity and coherence.** Specifically, we have revised the original algorithm description to provide a more detailed and comprehensive overview. (Section 5.1& Section 5.2 & Section 5.3, Page 5-7, Line 266-377; Section 5.3 & Algorithm 1, Page 7, Line 334-377; Appendix A & Algorithm 2, Page 16-17, Line 835-844, 864-886) These revisions include explicit step-by-step procedures and direct references to corresponding sections within the text, ensuring that the multistage training process is both transparent and easy to follow.

---

> ### Author Response · Authors · 2024-11-20
> **Response to Reviewer FFk6 (2/5)**
>
> - Furthermore, **we have reorganized the explanations within Section 5 to better introduce and motivate the overall loss function.** The detailed discussions in Sections 5.1, 5.2, and 5.3 (Page 5-7, Lines 266-377) now present a logical and progressive methodological framework. This restructuring ensures that each component of the loss function and its role in stabilizing training are clearly articulated, improving the accessibility of our core method.
>
> - In response to the need for elucidating the impact of various training components on the final performance, **we have incorporated ablation studies** in Appendix B.5 (Appendix B.5 & Table 4, Page 23, Line 1190-1226). These studies systematically evaluate the contributions of different elements of our model, including the mass loss term $\mathcal{L}\_{\text{Mass}}$ and the Fokker-Planck constraint $\mathcal{L}\_{\text{FP}}$. Our findings demonstrate that both the mass matching component and the Fokker-Planck constraint play important roles in enhancing the accuracy and reliability of trajectory reconstructions. Specifically, the inclusion of $\mathcal{L}\_{\text{Mass}}$ is crucial for accurate local mass matching in unbalanced settings, while the Fokker-Planck constraint improves performance, particularly at the final time point of the trajectory.
>
>
> > For a largely empirical work proposing a new algorithm to tackle this problem, the empirical studies are limited. The DeepRUOT is only benchmarked against balanced transport methods. It would greatly strengthen this paper if it was also compared to other unbalanced transport methods such as Action matching [Neklyudov et al. 2023] and unbalanced diffusion Schrödinger bridge [Pariset et al. 2023]. It would also be very useful to know how the many different weighting parameters affect performance, and how sensitive training is to them.
>
> - Thank you for your valuable suggestion. We fully agree that comparing our method with a broader range of baselines is essential to thoroughly demonstrate its effectiveness.  **In our original manuscript, we indeed included a comparison with Unbalanced Optimal Transport**, as our algorithm reduces to the framework proposed by Sha et al. (2024) when $\sigma = 0$. Since we observed that there exhibited certain instabilities in our dataset of their method, to ensure a fair comparison, we independently reimplemented their method and conducted additional evaluations. **We have clarified this point in the revised manuscript (Page 24, Line 1255-1259).** Building on your recommendation, we have now extended our experimental comparisons to include Unbalanced Action Matching (Unbalanced AM, [Neklyudov et al. 2023]) and Unbalanced Schrödinger Bridge (Unbalanced SB, [Pariset et al. 2023]). **These additions bring our comparisons to five baseline algorithms: Balanced OT, Unbalanced OT, Unbalanced AM, Balanced SB, and Unbalanced SB** (Section 6 & Table 1, Page 8, Line 423-428, 432-439; Table 2, Page 10, Line 487-494; Appendix B.1 & Figure 4, Page 18, Line 966-992; Appendix B.3, Page 19, Line 1019-1025; Figure 7, Page 21, Line 1095-1128). Our comprehensive evaluation demonstrates that DeepRUOT consistently outperforms all these baselines, reaffirming the robustness and superiority of our approach in modeling unbalanced stochastic dynamics.
>
> - For parameter selection, **we have expanded the discussion on loss function weighting** (Appendix C.2, Page 24, Line 1265-1280; Table 5, Page 25, Line 1296-1316). Specifically, we provide detailed values (Table 5, Page 25, Line 1296-1316) and the rationale behind the selection of these weights (Page 24, Line 1270-1276). During different training stages, we employ varying parameters to optimize performance; however, we have observed that these parameters exhibit robustness across diverse datasets, thereby minimizing the need for extensive adjustments.
>
>
> > It is difficult to tell what parts of the theory are novel and which are minor adaptations from prior work. Theorem 4.1 seems like a subclass of Baradat & Lavenant for the Fischer information case. I don’t believe varying sigma(t) over time changes anything theoretically and as far as I understand it is not used in practice.

---

> ### Author Response · Authors · 2024-11-20
> **Response to Reviewer FFk6 (3/5)**
>
> - Thank you for your insightful comments. We agree that Baradat & Lavenant (2023) have introduced another Fisher information regularization form of RUOT. We appreciate this important clarification. However, we would like to point out that the formulation presented in our work is **equivalent** to there rather than being a subclass (Page 5, Line 239-242; Appendix D.2, Page 26, Line 1378-1412). Moreover,  we consider our formulation to be more computationally tractable because it avoids the differentiation and vector multiplication operations present in the cross-term. In the original formulation, computing the cross term $\langle \nabla_{\boldsymbol{x}} \log p, \sigma^2(t) \boldsymbol{v} \rangle$ required calculating the derivative of the function $\log p$ with respect to $\boldsymbol{x}$ and performing vector multiplications with $\boldsymbol{v}$. **We have clarified this distinction in the revised manuscript to highlight the computational advantages of our formulation** (Page 5, Line 239-242; Appendix D.2, Page 26, Line 1378-1412).
>
> - Moreover, in Buze et al. (2023), the authors also define a RUOT problem incorporating nonlinear Fokker-Planck equation constraints. **We have elucidated the consistency of their proposed formulation with our RUOT Fisher information.** Specifically, we demonstrate that when the function $\Psi(g)$ adopts the Fisher-Rao metric, the resulting formulation aligns seamlessly with the Fisher information derived in our study (Appendix D.3, Page 27, Line 1413-1457), thereby providing a comprehensive understanding of RUOT problem.
>
> - Regarding Theorem 4.1, we agree with your insightful observation that varying $\sigma(t)$ over time does not theoretically alter the properties of the RUOT framework. If it turns to be  $\boldsymbol{\sigma}(\boldsymbol{x}, t)$, things will be more complex (Appendix D.4, Page 28-29, Line 1458-1524).
>
> - Additionally, we explore the connections between RUOT and SB problems and discuss some existing challenges within the field (Chen et al., 2022; Baradat & Lavenant, 2021) (Appendix E, Page 29-30, Line 1534-1608).
>
> > Questions:
> It is unclear to me exactly what the computational / numerical cost for allowing varying growth rates is in this framework. From my understanding both the deterministic and stochastic setting are extremely efficient, but it has thus far been numerically challenging to incorporate varying growth rates. How scalable is DeepRUOT? Can it be applied to the higher dimensional settings (20,50,100,1000D) in previous works?
>
> - Thank you for the insightful questions. We agree that in balanced scenarios, both deterministic and stochastic settings are highly efficient. Recent advancements, particularly those based on conditional flow matching (Lipman et al., 2022; Tong et al., 2023a, 2023b;), have enabled simulation-free transport between arbitrary distributions via ODEs or SDEs.  Regarding computational cost, our approach is comparable to existing frameworks such as TrajectoryNet (Tong et al., 2020), MIOflow (Huguet et al., 2022), and Koshizuka et al. (2023), which involve simulating ordinary or stochastic differential equations (ODEs/SDEs) and performing optimal transport for distribution matching or integration along trajectories. In  TrajectoryNet (Tong et al., 2023), the method has been tested up to 30 dimensions.  MIOflow (Huguet et al., 2022)  has been tested in 100 dimensions, (Koshizuka et al., 2023) have demonstrated efficacy in settings like 5-dimensional PCA in gene space. Incorporating varying growth rates does not introduce notable additional computational costs. Consequently, our method maintains scalability comparable to these existing frameworks and is currently best suited for relatively low-dimensional problems. **We have included the discussion of scability in the revised manuscript** (Page 19, Line 1006-1017).
>
> - **In our experiments, we have extended our tests to Gaussian mixture models in 20D, 50D, and 100D, demonstrating that our algorithm remains effective within these dimensions** (Section B.2, Page 19, Line 1002-1005;  Figure 6, Page 20, Line 1054-1079). However, when it extends to thousands of dimensions, our approach similarly encounters challenges like the previous methods, necessitating the development of simulation-free training methods akin to flow matching to handle higher-dimensional settings effectively. Indeed, we found the calculation of Wasserstein distance in datasets of thousands of dimensionalities in the loss function would also bring significant inaccuracy. Future work will explore such advancements to enhance the applicability of our framework to more complex, high-dimensional problems.

---

> ### Author Response · Authors · 2024-11-20
> **Response to Reviewer FFk6 (4/5)**
>
> > I don’t understand the motivation behind the reconstruction loss R_d. What is its purpose and why does it correspond to a reconstruction loss vs. say an OT loss?
>
> - Thank you for the insightful question regarding reconstruction loss  $\mathcal{L}\_{\text{Recons}}$ (revised notation to $R\_d$). The primary purpose of $\mathcal{L}\_{\text{Recons}}$ is to ensure that the dynamics of our model accurately match the target data distribution at the final time point, specifically  $p(\cdot, 1) = \nu_1$ . While many existing works utilize balanced optimal transport (OT) to measure the distance between two distributions, our approach addresses scenarios with unnormalized data distributions. Consequently, we employ unbalanced optimal transport to accommodate differences in total mass between distributions.
>
> - To achieve this,  $\mathcal{L}\_{\text{Recons}}$ is decomposed into two components:  $\mathcal{L}\_{\text{Mass}}$ and  $\mathcal{L}\_{\text{OT}}$ . The first component,  $\mathcal{L}\_{\text{Mass}}$, is responsible for aligning the number of cells by matching the mass of particles. This ensures that the particle masses evolve through the dynamics  $g$ to correspond with the actual data masses. This is why we introduced a mapping to link these two aspects, allowing us to match the corresponding masses via this mapping.
>
> - The second component,  $\mathcal{L}\_{\text{OT}}$, performs the optimal transport matching using the normalized distributions based on the matched masses. Importantly, unlike previous works, our OT loss incorporates sample-specific weights or masses, enhancing the fidelity of the transport process under unbalanced conditions. **We have clarified this methodology in the manuscript** (Section 5.2, Page 6, Line 295-333).
>
> > Comments:
> $\Psi(g)$ is not explicitly defined (although is fairly clear from context).
>
> - Thank you for the valuable feedback. **We have included the definition of $\Psi(g)$ in Definition 4.1** (Page 5, Line 216-217).
>
> > So to specify an SDE is equivalent to specify the probability flow ODE and its score function.
> Probably missing an article here.
>
> - Thank you for your careful attention. **We have revised it to read: “Thus, to specify an SDE is equivalent to specifying the probability flow ODE and the corresponding score function $\nabla_{\boldsymbol{x}} \log p(x,t)$ ” and have included the related references**, ensuring grammatical correctness and clarity (Page 5, Line 257-259).
>
> >Overall, this work has great potential, however I believe further empirical study would greatly improve its value to the community. I would be eager to reconsider my evaluation given further empirical study relative to existing unbalanced dynamic OT methods and a better empirical understanding of the various components of training and loss weighting.
>
>
>
> - Thank you for your insightful review and encouraging feedback. We have thoroughly addressed each of your points in the detailed responses above. We sincerely hope that the revisions will meet your expectations and that you find our manuscript suitable for publication. We are committed to addressing any additional comments and will gladly make further revisions if you find them needed.

---

> ### Author Response · Authors · 2024-11-20
> **Response to Reviewer FFk6 (5/5)**
>
> References
>
> - Pariset, M., Hsieh, Y. P., Bunne, C., Krause, A., & De Bortoli, V. (2023). Unbalanced Diffusion Schrödinger Bridge. arXiv preprint arXiv:2306.09099.
>
> - Neklyudov, K., Brekelmans, R., Severo, D., & Makhzani, A. (2023, July). Action matching: Learning stochastic dynamics from samples. In International conference on machine learning (pp. 25858-25889). PMLR.
>
> - Sha, Y., Qiu, Y., Zhou, P., & Nie, Q. (2024). Reconstructing growth and dynamic trajectories from single-cell transcriptomics data. Nature Machine Intelligence, 6(1), 25-39.
>
> - Baradat, A., & Lavenant, H. (2021). Regularized unbalanced optimal transport as entropy minimization with respect to branching brownian motion. arXiv preprint arXiv:2111.01666.
>
> - Tong, A., Huang, J., Wolf, G., Van Dijk, D., & Krishnaswamy, S. (2020, November). Trajectorynet: A dynamic optimal transport network for modeling cellular dynamics. In International conference on machine learning (pp. 9526-9536). PMLR.
>
> - Huguet, G., Magruder, D. S., Tong, A., Fasina, O., Kuchroo, M., Wolf, G., & Krishnaswamy, S. (2022). Manifold interpolating optimal-transport flows for trajectory inference. Advances in neural information processing systems, 35, 29705-29718.
>
> - Koshizuka, T., & Sato (2023), I. Neural Lagrangian Schrödinger Bridge: Diffusion Modeling for Population Dynamics. In The Eleventh International Conference on Learning Representations.
>
> - Lipman, Y., Chen, R. T., Ben-Hamu, H., Nickel, M., & Le, M. (2022). Flow matching for generative modeling. arXiv preprint arXiv:2210.02747.
>
> - Tong, A., Fatras, K., Malkin, N., Huguet, G., Zhang, Y., Rector-Brooks, J., ... & Bengio, Y. (2023a). Improving and generalizing flow-based generative models with minibatch optimal transport. arXiv preprint arXiv:2302.00482.
>
> - Tong, A., Malkin, N., Fatras, K., Atanackovic, L., Zhang, Y., Huguet, G., ... & Bengio, Y. (2023b). Simulation-free schrödinger bridges via score and flow matching. arXiv preprint arXiv:2307.03672.
>
> - Chen, Y., Georgiou, T. T., & Pavon, M. (2022). The most likely evolution of diffusing and vanishing particles: Schrödinger bridges with unbalanced marginals. SIAM Journal on Control and Optimization, 60(4), 2016-2039.
>
> - Buze, M., & Duong, M. H. (2023). Entropic regularisation of unbalanced optimal transportation problems. arXiv preprint arXiv:2305.02410.
>
> ---

---

> ### Author Response · Authors · 2024-11-22
>
> Dear Reviewer FFk6,
>
> Thank you very much for your valuable comments and insightful suggestions on our manuscript.  We are also excited to receive your encouraging feedback. We have carefully addressed the concerns raised in your review and revised our manuscript substantially. We have uploaded the revised manuscript, and all changes made to the manuscript are indicated in **blue** throughout the revised main text and appendix.  As the discussion period is approaching an end, could you please let us know if our revisions have satisfactorily resolved your concerns or if there are any remaining issues or questions you would like us to address before the rebuttal deadline? We are committed to providing timely and detailed responses to ensure that all aspects of our work are clarified. We are more than happy to take the valuable opportunity to discuss our work with you further or make any further adjustments as needed.
>
> Thank you very much for your time and consideration. We are looking forward to your reply.
>
> Yours sincerely,
>
> The Authors

---

> ### Author Response · Authors · 2024-11-24
> **A gentle reminder to reviewer**
>
> Dear Reviewer FFk6,
>
> Thank you very much for your valuable comments and insightful suggestions on our manuscript. We deeply appreciate the time and effort you have invested in reviewing our work. We have worked hard to carefully address the concerns raised in your review and have substantially revised our manuscript accordingly.
>
> As the discussion period is approaching its end on **November 26**, we kindly ask if you could please let us know whether our revisions have satisfactorily resolved your concerns or if there are any remaining issues or questions you would like us to address before the rebuttal deadline. We would greatly appreciate it if you could reevaluate our paper based on the revisions and consider updating your evaluation accordingly.  **Your feedback is invaluable to us and plays a crucial role in enhancing the quality of our work.**
>
>
> Thank you for your consideration and we are looking forward to your replies!
>
> Yours sincerely,
>
> The Authors

---

> > ### Comment · Reviewer_FFk6 · 2024-11-25
> >
> > I thank the authors for their time and effort in revising their work. I believe the additional experiments strengthen this work, and the presentation, while improved, still needs work. While the work is improved, I still feel there are a few items that are not quite clear and well supported in this work.
> >
> > Specifically around (1) the effectiveness of algorithm given lack of clarity around the parameters and (2) the clarity of presentation of the algorithm. Specifics detailed below.
> >
> > ### Parameter settings
> >
> > > we have observed that these parameters exhibit robustness across diverse datasets, thereby minimizing the need for extensive adjustments.
> > >
> >
> > To me, Table 5 does not exhibit robustness across diverse datasets. The fact that the number of epochs for the second pre-training stage is both highly variable and non-monotonic is highly irregular to me. Parameter robustness across diverse datasets is highly desireable in practice.
> >
> > I think it would be highly valuable to include an exploration of the effects of these parameters to support the claim that DeepRUOT exhibits robustness to parameter choices across diverse datasets.
> >
> > At a minimum it would be great to include in particular
> >
> > - how the number of epochs per pretraining phase were selected for the various datasets
> > - Why $\lambda_w$ is set lower for the regulatory network and mouse hematopoiesis datasets.
> >
> > ### Clarity
> >
> > I still find the updated nomenclature around training stages extremely confusing. I see both “Initial Training Phase” vs. “Secondary Training Phase” and Pre-Training Stage (Phase I), Pre-Training Stage (Phase II), and Post-Training Stage in Table 5.
> >
> > I am also not sure, but there seems to be two separate stages within the pretraining phase (lines 2-5 and lines 8-9 Algorithm 1) within the first pretraining stage that to my understanding do not line up with the phases in Table 5.
> >
> > Some specific suggestions on the algorithm formulation which I still found unclear:
> >
> > - $L_{\text{Recons}}, L_{\text{Mass}}, \text{ and } L_{\text{OT}}$ are all abused to specify both the loss per time point and the sum of losses over time. It would be clearer to be consistent here.
> > - $v_\theta$ is both used to describe the vectorfield and what I assume is the flow defined by the vectorfield from time $t-1$ to time $t$. Likewise with $g_\theta$. It would be clearer to use different notation for the flow especially in algorithm 1.
> > - Personally I would find it easier to understand a pre-training and training stage (rather than pre and post) as these suggest before and after some main training stage but this does not exist here. Or perhaps referring to them as training and fine-tuning steps would work equally well.
> >
> > I think the method would be much better motivated if it was clear how each piece and stage of the algorithm is necessary to achieve satisfactory empirical performance. While I commend the authors’ efforts on inclusion of other baselines, this is what I was trying to get at with my previous comment:
> >
> > > The paper would be much stronger if it was made clear which parts of the training procedure are important practically potentially through an ablation study on different components of the training procedure.
> > >
> >
> > I suggest that an ablation of training stages would make it much more clear what each part of the training procedure is necessary for.
> >
> > - 0 epochs of pretraining
> > - 0 epochs of second stage of pretraining (score training portion)
> > - 0 epochs of post training
> >
> > I given my experience, I would guess that post training is the only part that is theoretically necessary, but is extremely unstable to train. Pretraining of separate components helps with this instability, and is necessary for good empirical performance. It would be excellent to show this (or whatever is the case) empirically.
> >
> > It is also possible that I misunderstood exactly which parts of the training are necessary both theoretically and empirically. I would love if the authors could clear this up.
> >
> > ### Theory
> >
> > > In the original formulation, computing the cross term  required calculating the derivative of the function  with respect to and performing vector multiplications with . **We have clarified this distinction in the revised manuscript to highlight the computational advantages of our formulation** (Page 5, Line 239-242; Appendix D.2, Page 26, Line 1378-1412).
> >
> > I cannot find this clarification. In the specified lines. Where can it be found?
> >
> > ### Overall
> >
> > My only remaining major concern is the necessity and usefulness of all parts of the training procedure. I believe all other parts are potentially fixable with minor edits. If the authors can show which parts of the training procedure are empirically necessary, and clarify which parts are theoretically necessary, It would substantially improve my view of the completeness of this work.

---

> ### Author Response · Authors · 2024-11-25
>
> Dear Reviewer FFk6,
>
> We extend our heartfelt gratitude for your insightful feedback and constructive comments. We are pleased that we have addressed some of the issues you raised in your previous comments. Your thorough evaluation and thoughtful suggestions have been invaluable in enhancing the quality of our manuscript. Below, we provide some initial responses to your new comments:
>
> > I given my experience, I would guess that post training is the only part that is theoretically necessary, but is extremely unstable to train. Pretraining of separate components helps with this instability, and is necessary for good empirical performance. It would be excellent to show this (or whatever is the case) empirically.
>
> You are absolutely correct in your assessment. As you insightfully noted, the post-training stage (or as you suggested, the training stage) theoretically ensures that we are addressing a regularized unbalanced optimal transport (RUOT) problem. However, directly training this model presents significant challenges and often leads to convergence issues. This instability is precisely why we incorporated the pretraining stage—to mitigate these difficulties and enhance empirical performance. We wholeheartedly agree that providing empirical evidence to support this is essential. Therefore, we are pleased to include the results of additional experiments with:
>
>
> - 0 epochs of pretraining
> - 0 epochs of the second stage of pretraining (score training portion)
> - 0 epochs of post-training
>
> These experiments will substantiate your insightful observations and demonstrate the importance of the pretraining stages in our methodology.
>
> > Why $\lambda_w$  is set lower for the regulatory network and mouse hematopoiesis datasets.
>
> We sincerely apologize for any confusion caused by the setting of the $\lambda_w$ parameter. Our empirical studies have revealed that a smaller $\lambda_w$ value generally promotes greater stability across most datasets. In certain instances, larger $\lambda_w$ values introduced instability. Through our further investigations, we have found that setting $\lambda_w = 0.1$ maintains stability in all datasets. So to alleviate sensitivity concerns, we propose adopting a default $\lambda_w$ value of 0.1 in future implementations. We apologize not conducting a more comprehensive examination of $\lambda_w$ in our initial submission and are grateful for your attention to this detail. To present this, we are pleased to show additional results that explore the adjustment of the $\lambda_w$ parameter.
>
> > how the number of epochs per pretraining phase were selected for the various datasets
>
> As you correctly pointed out, the pretraining stage is crucial for stabilizing the training process. The number of epochs was determined based on the complexity and scale of the problem, including factors such as the number of time points and dataset size. However, our experiments have shown that increasing the number of pretraining epochs does not adversely affect the training process and can, in fact, improve performance. Consequently, we recommend conducting additional pretraining epochs to enhance training outcomes. We will show further results varying the number of pretraining epochs to illustrate this point clearly.
>
>
> > Clarifications and Manuscript Improvements
>
> We deeply appreciate your attention to detail and acknowledge the areas where our manuscript lacked clarity. We will undertake a comprehensive revision of the manuscript to address all clarity concerns, ensuring that our explanations are precise, coherent, and accessible to the readers.
>
> We sincerely hope that these proposed modifications meet your expectations and adequately address your valuable feedback. If you have any further suggestions or concerns regarding our **revision plan**, please do not hesitate to let us know. We are committed to providing a thoroughly revised and detailed version as promptly as possible. We are diligently working on the revisions and hope that the final manuscript meets your satisfaction.
>
> Once again, thank you very much for your constructive and encouraging feedback. Your expertise and guidance are immensely appreciated.
>
> Yours sincerely,
>
> The Authors

---

> > ### Comment · Reviewer_FFk6 · 2024-11-25
> >
> > I thank the authors for the clarity on their planned revisions. I believe these changes will greatly improve the clarity and impact of the paper, especially for non-experts in this subfield. In my opinion with these promised changes this work is a useful contribution to the trajectory inference subfield. I would ask the authors to please add these changes or at least changes that clarify which steps are important to stabilize the PDE consistency training.
> >
> > Given the above understanding I suggest to denote the two algorithms as pre-training and training, or warmup and training stages. I also suggest that the main training algorithm (Currently Alg. 2), be included in the main text if at all possible, as it is the algorithm that actually enforces the desired objective.
> >
> > As such I raise my score 3 --> 8.
> >
> > P.S. It seems the discussion period has been extended. I do not support such a change, and I do not wish to make the authors rush changes. I am satisfied with the current plan, and trust the authors to execute it. That being said, additional changes will help strengthen my argument for acceptance.

---

> > > ### Author Response · Authors · 2024-11-26
> > >
> > > Dear Reviewer FFk6,
> > >
> > > We would like to extend my heartfelt gratitude for your thoughtful and constructive feedback on our manuscript. We are immensely grateful for your appreciation of our planned revisions and for increasing your score. Your recognition of our work’s potential impact on the trajectory inference subfield is truly encouraging.
> > >
> > > We deeply value the specific suggestions you have provided and will try our best to incorporate them to enhance the clarity and robustness of our paper:
> > >
> > > -  Following your recommendation, we will rename the two algorithms to “Pre-training and Training” stages  to ensure better clarity and comprehension for all readers.
> > >
> > > - We agree wholeheartedly that integrating the main training algorithm (currently Algorithm 2) into the main text is essential. By doing so, we aim to provide a clearer illustration of how the desired objective is enforced, thereby strengthening the overall presentation of our methodology.
> > >
> > > -  We will diligently highlight and clarify the critical steps involved in stabilizing the PDE consistency training.
> > >
> > > Your support and trust in our work mean a great deal to us, and we will try our best to deliver an improved version that meets the expectations of the broader research community.
> > >
> > > Once again, thank you for your insightful comments, your time, and your support. We deeply appreciate your contribution to enhancing the quality of our research.
> > >
> > > Yours sincerely,
> > >
> > > The Authors

---

> ### Author Response · Authors · 2024-11-27
> **Second Revision (1/3)**
>
> Dear Reviewer FFk66,
>
> We would like to extend our deepest gratitude for your thoughtful and constructive feedback on our manuscript. Your insightful comments and suggestions have been invaluable in guiding us to enhance the quality and clarity of our work.
>
> We have substantially revised our manuscript according to your valuable feedback and have uploaded the updated version for your consideration. To facilitate your review, all new changes in the revised manuscript are highlighted in **red**.  The major revisions we have undertaken are listed as follows:
>
> - Following your suggestions, we have **conducted additional ablation experiments on different stages of our training process and included the results in the final analysis** (Appendix B.5 & Table 4,  Page 23, Line 1194-1196).
> - We have also **discussed which components are theoretically significant and which aspects empirically contribute to the performance improvements** (Appendix B.5,  Page 23, Line 1214-1227).
> - We have **addressed the number of epochs** used during the pretraining stage by conducting new experiments to illustrate the impact of epoch quantity on the outcomes (Appendix C.2 & Table 5 & Table 6, Page 24 & Page 25, Line 1277-1339).
> - We have **provided guidelines on selecting an appropriate number of epochs** (Appendix C.2 & Table 4 & Table 5, Page 24-25, Line 1277-1339).
> - We have **elaborated on the parameter $\lambda_w$, incorporating new experiments that demonstrate its influence on the results** (Appendix C.2 & Table 7, Page 25-26, Line 1338-1359).
> - To improve the clarity, we have **clarified relevant terminology and ensured that all tables correspond consistently** with the narrative throughout the text (Section 5.2, Page 6, Line 310-312 & 316-320; Section 5.3, Page 7, Line 356-358; Algorithm 1, Page 8, Line 378-403; Appendix C.2, Page 24, Line 1277-1286; Table 5. Page 25, Line 1296-1310).
> - We have **revised the algorithm related to the $\mathcal{L}\_{\text{Recons}}, \mathcal{L}\_{\text{mass}}, \mathcal{L}\_{\text{OT}}$ in our algorithm** to maintain consistency with the definitions provided (Algorithm 1, Page 8, Line 387;).
> - We have **introduced new notation** ($\phi_\theta^{\boldsymbol{v}}$ and $\phi_\theta^{g}$) to clearly distinguish Neural ODE update process with vector field (Section 5.2, Page 6, Line 310-312 & 316-320; Algorithm 1, Page 8, Line 378-403;).
>
> - We **have also replaced the previously used terms**“pre-training stage” and “post-training stage” with “pre-training stage” and “training stage,” respectively.
>
> - We have **merged the previously separate Algorithm 1 and Algorithm 2 into the new algorithm in the main text** (Algorithm 1, Page 8, Line 378-403).
>
> - We apologize for the missing part concerning the theoretical aspects you raised. We **have now included a more detailed description in the manuscript** (Page 5, Line 238-243).
>
>
> Once again, thank you very much for your constructive and encouraging feedback. Your expertise and guidance are immensely appreciated. Below, we provide a detailed elaboration of some points.

---

> ### Author Response · Authors · 2024-11-27
> **Second Revision (2/3)**
>
> > Following your suggestions, we have **conducted additional ablation experiments on different stages of our training process and included the results in the final analysis** (Appendix B.5 & Table 3,  Page 23, Line 1194-1196).  We have also **discussed which components are theoretically significant and which aspects empirically contribute to the performance improvements** (Appendix B.5,  Page 23, Line 1218-1231).
>
> -  As you insightfully commented, the pre-training stage is critical within our framework, while the training stage contributes to the refinement of final results, ensuring we are addressing the regularized optimal transport problem, which is theoretically necessary.  The pre-training stage significantly enhances the performance of our algorithm. So, we investigated the impact of different stages in our training process on the final results. To assess their respective contributions, we conducted experiments under three scenarios according to your suggestions: (1) omitting the pre-training stage, (2) excluding score matching during the pre-training stage, and (3) removing the training stage.
> -  As presented in Table 4 (Page 23, Line 1194-1196), we observed that the algorithm failed to converge in the first two scenarios. Specifically, scenario (2) yielded poorer performance due to inaccurate score matching, which caused particle trajectories to deviate from the correct dynamical paths. In scenario (1), the necessity to balance multiple loss functions resulted in the vector field and score not being adequately learned, leading particles are inclined to remain near their initial positions. In scenario (3), the absence of the training stage slightly degraded the performance metrics, especially at later time points.
> -  These experiments substantiate your insightful observations and demonstrate the importance of the pretraining stages in our methodology.
>
> > We have **addressed the number of epochs** used during the pretraining stage by conducting new experiments to illustrate the impact of epoch quantity on the outcomes (Appendix C.2 & Table 4 & Table 5, Page 24 & Page 25, Line 1277-1339). We have **provided guidelines on selecting an appropriate number of epochs** (Appendix C.2 & Table 4 & Table 5, Page 24-25, Line 1277-1339).
>
> - As you insightfully pointed out, the primary hyperparameters requiring adjustment during the pre-training stage are mainly the number of training epochs selected during the $\lambda_m$ resetting process, specifically the number of epochs trained **before** $\lambda_m$ resetting (denoted as **first epoch number** below) and the number of epochs trained **after** $\lambda_m$ resetting (denoted as **second epoch number** below).
> - As previously discussed, the pre-training stage is crucial for stabilizing the training process. Determining the number of epochs relies on the complexity and scale of the problem, including factors such as the number of time points and the size of the dataset. For the first epoch number, a default of 30 epochs is recommended. However, in the case of the Gene Regulatory Network dataset, we opted for 20 epochs due to its larger number of time points and greater data volume. Similarly, for the EMT dataset, 20 epochs were chosen as it had already achieved convergence.
> - In Table 6 (Page 25, Line 1331-1339), we evaluated the GRN dataset by maintaining the second epoch number as ten while varying the first epoch number among 10, 20, and 30, with the other hyperparameters chosen as Table 5. The results indicated that the first epoch number value of 10, which was insufficient,  deteriorated performance, whereas the first epoch number values of 20 and 30 showed negligible differences. This suggests that increasing the number of pre-training epochs does not adversely affect the training process; hence, we recommend training for a higher number of epochs when feasible.
> - The selection of the second epoch number varies across datasets based on convergence criteria. For instance, in the GRN dataset, training with the second epoch number set to 10 was sufficient for convergence, leading us to terminate further training. Conversely, for the EMT dataset, convergence was achieved during the first stage, negating the need for additional second epoch number training. Generally, a higher number of epochs in this stage enhances training efficacy. In the case of Gaussian Mixtures, the number of the second epoch number increases with dimensionality. However, for 50D and 100D, the second epoch number decreases because, for 100D, training can extend beyond 120 epochs for improved results, while 50D converges around 100 epochs, making further training unnecessary. Therefore, for the adjustment of the second epoch number, we recommend that users train until convergence based on their specific data requirements.

---

> ### Author Response · Authors · 2024-11-27
> **Second Revision (3/3)**
>
> >  We have **elaborated on the parameter $\lambda_w$, incorporating new experiments that demonstrate its influence on the results** (Appendix C.2 & Table 6, Page 25-26, Line 1340-1362).
>
> - We first sincerely apologize for not conducting a more comprehensive examination of  $\lambda_w$ and for any confusion about setting  $\lambda_w$ in our initial submission. We are grateful for your attention to this detail.
> -  We found that $\lambda_w$ may relate to the noise amplitude  $\sigma$. Specifically, for the Gene Regulatory Network (GRN) and Mouse Hematopoiesis datasets, where $\sigma$ is set to 0.25, a smaller $\lambda_w$ is necessary to maintain computational stability; setting $\lambda_w$ to 1 in these cases resulted in instability. For the remaining datasets, we have found that selecting $\lambda_w$ within the range of [0.1, 1] is feasible, as variations within this range do not significantly impact the training outcomes.
> In Table 7 (Page 26, Line 1350-1358), we present the results of training with $\lambda_w$ values of 0.1, 0.4, 0.7, and 1.0 on the EMT dataset. We computed the values of the trained results ($\boldsymbol{v}$, $g$, $s$) at each data point, using the results obtained with $\lambda_w = 1$ as the baseline. The distances between the outcomes of other $\lambda_w$ settings and the baseline were measured using Euclidean distance for inferred $\boldsymbol{v}$ and mean squared error (MSE) for $g$ and $s$. As illustrated in  Table 6 (Page 26, Line 1350-1358), we observed that as long as $\lambda_w$ ensures computational stability, the outcomes for different $\lambda_w$ values are consistent. Therefore, we recommend selecting a smaller $\lambda_w$, with a default value of 0.1, to achieve stable and consistent training results.
>
> > To improve the clarity, we have **clarified relevant terminology and ensured that all tables correspond consistently** with the narrative throughout the text (Section 5.2, Page 6, Line 310-312 & 316-320; Section 5.3, Page 7, Line 357-358; Algorithm 1, Page 8, Line 378-403; Appendix C.2, Page 24, Line 1277-1286; Table 4. Page 25, Line 1301-1312).
>
> - We have revised the description of the algorithm in the manuscript to ensure that the stages of the algorithm (Algorithm 1, Page 8, Line 378-403) are consistent with the processes illustrated in Table 5 (Page 25, Line 1296-1310). Additionally, we have introduced the term “hyperparameter scheduling” to describe the process of parameter adjustments during the pretraining stage. These changes enhance the clarity of our presentation and provide a more precise alignment between the algorithmic stages and the corresponding processes outlined in the table.
>
> Finally, we would like to express our sincere gratitude for your valuable suggestions and the time you have dedicated to reviewing our manuscript. We hope that these revisions meet your expectations, and should you have any further suggestions, please do not hesitate to let us know. We are committed to making any necessary modifications until the very last moment before the deadline. Your insightful feedback has significantly contributed to enhancing the quality and clarity of our work.
>
> Yours sincerely,
>
> The Authors

---

### Official Review · Reviewer_RQfu · 2024-11-03

**Soundness:** 3
**Presentation:** 3
**Contribution:** 3
**Rating:** 8
**Confidence:** 3

**Summary:**

This paper proposes a new approach for learning dynamics from the observation of distributions at different time steps. The paper proposes to use an unbalanced optimal transport approach that allows modeling particle growth or death over time, which is an important aspect of single cell dynamics modeling. The authors compare their approach on datasets from syntethic gene regulatory networks and single cell trajectories dynamics. The paper shows favorable performance results compared to previous methods.

**Strengths:**

- Population dynamics from snapshot data is an important problem, especially in biological application such as single cell trajectories inference.
- The proposed method appears to surpass the performance of previous methods.
- The related works section is extensive.

**Weaknesses:**

- One of the most important weaknesses of this paper at this stage is its clarity. In its current form, the paper is poorly written, and the ideas are poorly motivated. The main limitation from prior works identified is the unabalanced aspect e.g. "However, computational methods for learning high-dimensional unbalanced stochastic dynamics from snapshots without prior knowledge of growth and death or additional information are still lacking.". Yet, the paper only indirectly addresses that issue, which confuses the reader. Despite the favorable experimental results, the current presentation of this paper prevents it from being appreciated by the community.

- Section 5, which is the core of your method, is extremly confusing. For instance, the reconstruction loss is not introduced previously such that it's extremely challenging to follow the development until the final loss in Equation 10. The algorithm 1 does not provide any help either in much needed clarification. The algorithm is too laconic and does not refer to corresponding parts of the text.

- It's not clear to me why the connection with the Schrodinger bridge is necessary. As far as my reading goes, it seems that you only need to leverage the dynamic version of regularized unbalanced OT. As such, it does not seem that the connection with Schrodinger bridges helps the reader understand the method, and you don't leverage that connection later on in your method.

- Theorem 4.1 uses $v$ which is not introduced previously. It's later undertstood that it corrresponds to the vector field of the corresponding ODE but this hinders the clarity of the paper.

- The presentation of the results in Figure 2 raises questions. Why does panel (b), with the true dynamics has ground truth, predicted and trajectories, if it is the ground truth dynamics ? What is the "predicted" in this case ?

- Because the most important contribution of this paper relies on modeling the growth rate, an ablation study is required where the growth rate part of the method is disabled (e.g. g = 0). This would allow to understand the impact of it on the final performance.

- Despite the extensive related works section, the experiments only compare with two baselines. Baselines such as Bunne et al. or Yutong et al. could also be considered.

Sha, Yutong, et al. "Reconstructing growth and dynamic trajectories from single-cell transcriptomics data."

**Questions:**

- Regarding the connection with SB, could you either make the impact of the connection more explicit, or considering removing that part from the paper for clarity ?

- Could you please restructure Section 5 to help the readers follow the reasoning until the final loss and algorithm ? Having a more detailed pseudo-code description would already help significantly.

- Can you clarify Figure 2 - the predictions dots in the ground truth panel ?

- Could you please add the results of an ablation experiment where you disable the learnable growth rate in your model ?

---

> ### Author Response · Authors · 2024-11-20
> **Response to Reviewer RQfu (1/4)**
>
> > Summary:
> This paper proposes a new approach for learning dynamics from the observation of distributions at different time steps. The paper proposes to use an unbalanced optimal transport approach that allows modeling particle growth or death over time, which is an important aspect of single-cell dynamics modeling. The authors compare their approach on datasets from synthetic gene regulatory networks and single cell trajectories dynamics. The paper shows favorable performance results compared to previous methods.
>
> > Strengths:
> Population dynamics from snapshot data is an important problem, especially in biological application such as single cell trajectories inference.
> The proposed method appears to surpass the performance of previous methods.
> The related works section is extensive.
>
>
> - Thank you for your careful reading and insightful suggestions. Thank you for your clear summary and positive comments. Trajectory inference is indeed a pivotal area in both machine learning and single-cell biology. Many methods have been developed in recent years.
>
> - In our study, we address this challenge while accounting for both stochastic and unbalanced effects. Our primary contribution lies in the development of DeepRUOT, a new deep-learning framework that leverages RUOT in conjunction with a computationally tractable Fisher regularization formulation. By employing neural networks to model drift score and growth function, our method effectively learns stochastic dynamics and unbalanced effects directly from the data without requiring any prior assumptions. We believe this capability makes DeepRUOT a more adaptable and comprehensive tool for studying complex systems.
>
> - Below you will find our detailed point-by-point responses. We have revised our manuscript accordingly to address the comments, and all changes made to the manuscript are indicated in **blue** throughout the revised main text and appendix.   The major revisions we have made are listed as follows:
>
>   - We have **thoroughly reformatted the manuscript** to improve its overall organization and clarity (Section 1, Page 2, Line 58-60, 73-74; Section 3, Page 3, Line 110-114, 153-155; Section 5, Page 5-7, Line 266-362; Algorithm 1, Page 8, Line 378-404;  Appendix A, Page 16, Line 836-847).
>    - We have **restructured the explanations within Section 5** to better introduce and motivate the overall loss function (Section 5.1 & Section 5.2 & Section 5.3, Page 5-7, Line 266-362; Algorithm 1, Page 8, Line 378-404).
>   - We have **revised the description of our algorithm** to make it more accessible and easier to understand (Section 5.3, Page 7, Line 339-363; Algorithm 1, Page 8, Line 378-404; Appendix A, Page 16, Line 836-847).
>   - We have **clarified the motivation** behind our work and its relation to the Schrödinger bridge problem (Section 3, Page 3, Line 110-114, 153-155).
>   - We have **included ablation studies** to systematically evaluate the contributions of different components of our model (Appendix B.5 & Table 4, Page 22-23, Line 1174-1228).
>   - We have **expanded our experimental comparisons** to include **five** baseline algorithms, specifically Balanced OT, Unbalanced OT, Balanced SB,  Unbalanced Action Matching (Unbalanced AM) and Unbalanced SB (Section 6 & Table 1, Page 8, Line 412-418, 420-426; Table 2, Page 10, Line 493-501; Appendix B.1, Page 18, Line 944-950; Figure 4, Page 19, 972-994; Appendix B.3 & Figure 7, Page 20-21, Line 1057-1115).
>   - We have **revised our figures** to enhance their clarity (Figure 2 (b), (d), Page 9, Line 432-446).
>
> Thank you once again for your valuable suggestions, which have significantly enhanced the quality and clarity of our work.

---

> ### Author Response · Authors · 2024-11-20
> **Response to Reviewer RQfu (2/4)**
>
> > Weaknesses:
> One of the most important weaknesses of this paper at this stage is its clarity. In its current form, the paper is poorly written, and the ideas are poorly motivated. The main limitation from prior works identified is the unbalanced aspect e.g. "However, computational methods for learning high-dimensional unbalanced stochastic dynamics from snapshots without prior knowledge of growth and death or additional information are still lacking.". Yet, the paper only indirectly addresses that issue, which confuses the reader. Despite the favorable experimental results, the current presentation of this paper prevents it from being appreciated by the community.
>
>
> - Thank you for your valuable feedback. We apologize for any confusion caused by the clarity of our manuscript. **In the revised version, we have reformatted the manuscript thoroughly to improve its clarity** (Section 1, Page 2, Line 58-60, 73-74; Section 3, Page 3, Line 110-114, 153-155; Section 5, Page 5-7, Line 266-362; Algorithm 1, Page 8, Line 378-404;  Appendix A, Page 16, Line 836-847). As previously mentioned, many existing methods in trajectory inference rely on ordinary differential equation (ODE) modeling; however, these approaches often neglect stochastic, which is crucial in both single-cell biology and machine learning. To further use stochastic differential equation (SDE), transport between arbitrary distributions has been approached through the Schrödinger bridge problem. However, these methods typically do not account for matching unbalanced distributions. As highlighted by Reviewer FFk603 and to our knowledge, computational methods for learning RUOT or such high-dimensional unbalanced stochastic dynamics from snapshots are relatively lacking, especially when there is no prior knowledge of unbalanced effects. Most existing methods require prior information about the form of the growth function  $g(\boldsymbol{x},t)$ , which is relatively difficult to specify in practice. Therefore, our approach leverages the RUOT framework and employs neural networks to model drift, score, and growth functions, enabling the effective learning of stochastic dynamics and unbalanced effects directly from data without any prior assumptions. **We believe this capability makes DeepRUOT a more adaptable and comprehensive tool for studying complex systems, thereby addressing a significant gap in the current literature and enhancing the utility of trajectory inference methods in both biological and machine learning applications.**
>
> > Section 5, which is the core of your method, is extremely confusing. For instance, the reconstruction loss is not introduced previously such that it's extremely challenging to follow the development until the final loss in Equation 10. The algorithm 1 does not provide any help either in much-needed clarification. The algorithm is too laconic and does not refer to corresponding parts of the text.
>
> - Thank you for your valuable feedback. We sincerely apologize for the confusion caused by the original presentation of our method. **To address your concerns, we have restructured the initial Algorithm description in a more detailed and comprehensive way** (Section 5.3, Page 7, Line 339-363; Algorithm 1, Page 8, Line 378-404; Appendix A, Page 16, Line 836-847). This revised algorithm description includes detailed steps and explicitly references the corresponding sections of the text. Additionally, **we have restructured the explanations within Section 5 to better introduce and motivate the overall loss** (Section 5.1 & Section 5.2 & Section 5.3, Page 5-7, Line 266-362; Algorithm 1, Page 8, Line 378-404), ensuring that the methodological progression is logical and easy to follow. These enhancements aim to improve the clarity and accessibility of our core method, facilitating a better understanding of DeepRUOT’s framework and its implementation.

---

> ### Author Response · Authors · 2024-11-20
> **Response to Reviewer RQfu (3/4)**
>
> > It's not clear to me why the connection with the Schrödinger bridge is necessary. As far as my reading goes, it seems that you only need to leverage the dynamic version of regularized unbalanced OT. As such, it does not seem that the connection with Schrödinger bridges helps the reader understand the method, and you don't leverage that connection later on in your method.
>
> - Thank you for your insightful feedback. We agree that the primary focus of our work is on leveraging the dynamic version of RUOT. However, the introduction of the Schrödinger bridge was intended to provide a foundational motivation for RUOT. Specifically, RUOT can be viewed as an extension of the Schrödinger bridge problem, where the latter primarily addresses stochasticity without accounting for unbalanced distributions. By drawing this connection, we aimed to highlight how RUOT naturally extends the Schrödinger bridge framework to incorporate growth and death processes.
>
> - **In our revised manuscript, we have clarified this motivation and its relation to the Schrödinger bridge** (Section 3, Page 3, Line 110-114, 153-155). We demonstrate that while the Schrödinger bridge focuses on mass-conserving transport, RUOT relaxes this constraint by introducing a growth/death term  $gp$ in the dynamical formulation:
>
> $$
> \partial_t p = -\nabla_{\boldsymbol{x}} \cdot \left(p \boldsymbol{b}\right) + \frac{1}{2} \nabla_{\boldsymbol{x}}^2 : \left(\sigma^2(t) \boldsymbol{I} p\right) + gp.
> $$
>
> - Additionally, we define a loss function that incorporates both the growth penalization and the Wasserstein metric. This introduces regularized unbalanced optimal transport formulation.
>
>
> > Theorem 4.1 uses $v$ which is not introduced previously. It's later understood that it corresponds to the vector field of the corresponding ODE but this hinders the clarity of the paper.
>
> - Thank you for your valuable feedback. **We have explicitly introduced  $v$ now to enhance the clarity (Page 5 Line 239).**
>
> > The presentation of the results in Figure 2 raises questions. Why does panel (b), with the true dynamics has ground truth, predicted and trajectories, if it is the ground truth dynamics ? What is the "predicted" in this case ?
>
> - Thank you for your valuable clarification. We apologize for the confusion caused by the original presentation of Figure 2(b). As you correctly noted, Figure 2(b) represents the ground truth dynamics, which is simulated by the dynamics governed by SDEs (Appendix B.1, Page 17, Line 904-913), where in this case all marked points, including crosses and dots, are generated by the true dynamics, and the trajectory illustrates the actual dynamical path. This panel was intended to correspond to panels (d) and (e). To rectify this, **we have relocated the relevant panels from (b) to (e) to ensure accurate and clear representation** (Figure 2 (b), (d), Page 9, Line 432-446).
>
> > Because the most important contribution of this paper relies on modeling the growth rate, ablation studies is required where the growth rate part of the method is disabled (e.g. g = 0). This would allow to understand the impact of it on the final performance.
>
> - Thank you for your insightful suggestion. We fully agree that evaluating the performance with the growth rate disabled (i.e., setting  $g = 0$ ) is crucial for understanding its contribution to the overall framework. **In our original manuscript, we indeed conducted a comparison where  $g = 0$.** Under this condition, our algorithm reduces to the framework of regularized optimal transport (as we discussed on Page 5, Line 223-224), which is equivalent to the Schrödinger bridge problem (Gentil et al., 2017; Baradat & Lavenant, 2021). Thus, the Schrödinger bridge algorithm (SF2M) corresponds to the scenario where  $g = 0$ in our approach. Our experiments demonstrated that omitting the growth factor leads to inaccurate trajectory reconstructions, highlighting the necessity of modeling unbalanced dynamics. **We have clarified this in the revised manuscript** (Appendix B.5, Page 23, Line 1205-1214).
>
> - **To further illustrate this insightful point, we have now included an explicit ablation study where  $g = 0$** (Appendix B.5 & Table 4, Page 22-23, Line 1174-1228).. The results from this experiment align with those of the SF2M method, confirming that disabling the growth rate adversely affects performance by ignoring essential growth and death processes. This comparison reinforces the importance of the growth rate component in accurately capturing the unbalanced stochastic dynamics, thereby validating the enhanced performance of DeepRUOT.

---

> ### Author Response · Authors · 2024-11-20
> **Response to Reviewer RQfu (4/4)**
>
> > Despite the extensive related works section, the experiments only compare with two baselines. Baselines such as Bunne et al. or Yutong et al. could also be considered.
>
> - Thank you for your valuable suggestion. We fully agree that comparing our method with a broader range of baselines is essential to thoroughly demonstrate its effectiveness. **In our original manuscript, we indeed included a comparison with Unbalanced Optimal Transport**, as our algorithm reduces to the framework proposed by Sha et al. (2024) when $\sigma = 0$. Since we observed that there exhibited certain instabilities in our dataset of their method, to ensure a fair comparison, we independently reimplemented their method and conducted additional evaluations. **We have clarified this point in the revised manuscript (Page 24, Line 1255-1259).** Building on your recommendation, we have now extended our experimental comparisons to include Unbalanced Action Matching (Unbalanced AM, [Neklyudov et al. 2023]) and Unbalanced Schrödinger Bridge (Unbalanced SB, [Pariset et al. 2023]). **These additions bring our comparisons to five baseline algorithms: Balanced OT, Unbalanced OT, Unbalanced AM, Balanced SB, and Unbalanced SB**  (Section 6 & Table 1, Page 8, Line 412-418, 420-426; Table 2, Page 10, Line 493-501; Appendix B.1, Page 18, Line 944-950; Figure 4, Page 19, 972-994; Appendix B.3 & Figure 7, Page 20-21, Line 1057-1115). Our comprehensive evaluation demonstrates that DeepRUOT consistently outperforms all these baselines, reaffirming the robustness and superiority of our approach in modeling unbalanced stochastic dynamics.
>
>
> > Questions:
> Regarding the connection with SB, could you either make the impact of the connection more explicit, or considering removing that part from the paper for clarity ?
> Could you please restructure Section 5 to help the readers follow the reasoning until the final loss and algorithm ? Having a more detailed pseudo-code description would already help significantly.
> Can you clarify Figure 2 - the predictions dots in the ground truth panel ?
> Could you please add the results of an ablation experiment where you disable the learnable growth rate in your model ?
>
> - Thank you for your insightful questions and constructive feedback. We have thoroughly addressed each of your points in the detailed responses above. We sincerely hope that the revisions will meet your expectations and that you find our manuscript suitable for publication. We are committed to addressing any additional comments and will gladly make further revisions if you find them needed.
>
> ---
>
> References
>
> - Pariset, M., Hsieh, Y. P., Bunne, C., Krause, A., & De Bortoli, V. (2023). Unbalanced Diffusion Schrödinger Bridge. arXiv preprint arXiv:2306.09099.
>
> - Neklyudov, K., Brekelmans, R., Severo, D., & Makhzani, A. (2023, July). Action matching: Learning stochastic dynamics from samples. In International conference on machine learning (pp. 25858-25889). PMLR.
>
> - Sha, Y., Qiu, Y., Zhou, P., & Nie, Q. (2024). Reconstructing growth and dynamic trajectories from single-cell transcriptomics data. Nature Machine Intelligence, 6(1), 25-39.
>
> - Gentil, I., Léonard, C., & Ripani, L. (2017). About the analogy between optimal transport and minimal entropy. In Annales de la Faculté des sciences de Toulouse: Mathématiques (Vol. 26, No. 3, pp. 569-600).
>
> - Baradat, A., & Lavenant, H. (2021). Regularized unbalanced optimal transport as entropy minimization with respect to branching brownian motion. arXiv preprint arXiv:2111.01666.
>
> ---

---

> ### Author Response · Authors · 2024-11-22
>
> Dear Reviewer RQfu,
>
> Thank you very much for your valuable comments and insightful suggestions on our manuscript. We are also excited to receive your valuable feedback. We have carefully addressed the concerns raised in your review and revised our manuscript substantially. We have uploaded the revised manuscript, and all changes made to the manuscript are indicated in **blue** throughout the revised main text and appendix.  As the discussion period is approaching an end, could you please let us know if our revisions have satisfactorily resolved your concerns or if there are any remaining issues or questions you would like us to address before the rebuttal deadline? We are committed to providing timely and detailed responses to ensure that all aspects of our work are clarified. We are more than happy to take the valuable opportunity to discuss our work with you further or make any further adjustments as needed.
>
> Thank you again for your time and consideration. We are looking forward to your reply.
>
> Yours sincerely,
>
> The Authors

---

> ### Author Response · Authors · 2024-11-24
> **A gentle reminder to reviewer**
>
> Dear Reviewer RQfu,
>
> Thank you very much for your valuable comments and insightful suggestions on our manuscript. We deeply appreciate the time and effort you have invested in reviewing our work. We have worked hard to carefully address the concerns raised in your review and have substantially revised our manuscript accordingly.
>
> As the discussion period is approaching its end on **November 26**, we kindly ask if you could please let us know whether our revisions have satisfactorily resolved your concerns or if there are any remaining issues or questions you would like us to address before the rebuttal deadline. We would greatly appreciate it if you could reevaluate our paper based on the revisions and consider updating your evaluation accordingly.  **Your feedback is invaluable to us and plays a crucial role in enhancing the quality of our work.**
>
>
> Thank you for your consideration and we are looking forward to your replies!
>
> Yours sincerely,
>
> The Authors

---

> ### Author Response · Authors · 2024-11-26
> **A gentle reminder to reviewer**
>
> Dear Reviewer RQfu,
>
> Thank you very much for your valuable comments and insightful suggestions on our manuscript. We deeply appreciate the time and effort you have invested in reviewing our work. We have worked hard to carefully address the concerns raised in your review and have substantially revised our manuscript accordingly.
>
> As the discussion period is approaching its end on **November 27**, we kindly ask if you could please let us know whether our revisions have satisfactorily resolved your concerns or if there are any remaining issues or questions you would like us to address before the rebuttal deadline. We are unable to make further modifications to our manuscript after **November 27** and are committed to ensuring that it meets the highest standards. We would greatly appreciate it if you could reevaluate our paper based on the revisions and consider updating your evaluation accordingly. Your feedback is invaluable to us and plays a crucial role in enhancing the quality of our work.
>
> Thank you for your consideration and we are looking forward to your replies!
>
> Yours sincerely,
>
> The Authors

---

> ### Author Response · Authors · 2024-11-27
> **A gentle reminder to reviewer RQfu (Second Revision)**
>
> Dear Reviewer RQfu,
>
> Thank you very much for your valuable comments and insightful suggestions on our manuscript. We deeply appreciate the time and effort you have invested in reviewing our work.
>
> We have diligently addressed the concerns raised in your review and have substantially revised our manuscript accordingly. Specifically, we have incorporated Reviewer FFk6’s comments and revised our manuscript once again. The second round of modifications is marked in **red** for your convenience. **Additionally, we have updated the page numbers in our previous responses to facilitate your review.**
>
> As the discussion period is set to conclude shortly, we kindly ask if you could please let us know whether our revisions have satisfactorily resolved your concerns or if there are any remaining issues or questions you would like us to address before the rebuttal deadline. We are committed to making any necessary modifications until the very last moment before the deadline.
>
> We would greatly appreciate it if you could reevaluate our paper based on the revisions and consider updating your evaluation accordingly. Your feedback is invaluable to us and plays a crucial role in enhancing the quality of our work.
>
> Thank you once again for your time and assistance.
>
> Yours sincerely,
>
> The Authors

---

> ### Author Response · Authors · 2024-11-29
> **A gentle reminder to reviewer RQfu**
>
> Dear Reviewer RQfu,
>
> We hope you are having a great day! We would like to express our sincere gratitude for your valuable feedback and insightful suggestions on our manuscript. Your thoughtful comments have been instrumental in significantly enhancing the quality of our work, and we sincerely appreciate the time and effort you have dedicated to reviewing our submission.
>
> We have carefully addressed all the concerns raised in your review. In our revised manuscript, we have highlighted the changes for your convenience and provided specific page numbers and line numbers corresponding to each modification. Additionally, all your questions and comments have been thoroughly addressed in our previous responses, including a detailed point-by-point response to each of your comments. Furthermore, we are profoundly grateful for the insightful assistance provided by Reviewer FFk6, which has greatly enhanced the clarity and overall quality of our manuscript. We hope that these revisions satisfactorily resolve your concerns.
>
> As the discussion period is set to conclude shortly (**December 2**), we would be extremely grateful if you could kindly review our revised manuscript at your convenience. Should you have any remaining questions or require further clarification, please do not hesitate to let us know. **We are very pleased and eagerly look forward to the opportunity to engage in further in-depth discussions about our work with you.**
>
> We fully understand the demands of your time, and we greatly appreciate your continued support and assistance throughout this process. Your feedback has been invaluable to us.
>
> Thank you once again for your time and consideration.
>
> Yours sincerely,
>
> The Authors

---

> > ### Comment · Reviewer_RQfu · 2024-12-03
> > **Thank you for your response**
> >
> > I appreciate the authors' efforts in addressing my comments and have found the edits provided to the paper to greatly improve the clarity.
> >
> > I also highly appreciate the inclusion of several other baselines which further demonstrate the advantages of this approach.
> >
> > I have increased my score accordingly.

---

> > > ### Author Response · Authors · 2024-12-03
> > > **Thank you!**
> > >
> > > Dear Reviewer RQfu,
> > >
> > > Thank you very much for raising the score! We would like to extend our deepest gratitude for your thoughtful and constructive feedback. Your insightful comments and suggestions have been instrumental in guiding us to enhance the quality and clarity of our work. Additionally, your support and trust in our research mean a great deal to us.
> > >
> > > Thank you for your time, consideration, and invaluable contributions to improving our manuscript.
> > >
> > >
> > > Yours sincerely,
> > >
> > > The Authors

---

### Official Review · Reviewer_FkRe · 2024-11-04

**Soundness:** 3
**Presentation:** 3
**Contribution:** 4
**Rating:** 10
**Confidence:** 2

**Summary:**

The paper introduces DeepRUOT, a new deep-learning approach for solving regularized unbalanced optimal transport (RUOT) and inferring continuous unbalanced stochastic dynamics from observed snapshots. The method models dynamics without requiring prior knowledge of growth and death processes, allowing these to be learned directly from data through a Fisher regularization formulation. The authors explore theoretical connections between RUOT and the Schrödinger bridge problem while reformulating RUOT to transform stochastic differential equations into more computationally tractable ordinary differential equations (a constraint). When tested on both synthetic gene regulatory networks and real single-cell RNA sequencing data, DeepRUOT outperformed existing methods in constructing meaningful developmental landscapes.

**Strengths:**

- The experimental results seem very convincing
- The paper provides thorough theoretical foundations with formal theorems and proofs connecting RUOT to the Schrödinger bridge problem.

**Weaknesses:**

- it remains unclear how the loss function weights (cost, reconstruction error and the PINN loss) are set
- Figure 1 and its caption are not that helpful

Typos:
- Theorem 3.1: "probelm" -> "problem"
- page 8: "a increase" -> "an increase"

**Questions:**

- is not knowing the death/growth rates (what you call prior knowledge) such a big advantage?
- Why did you project the data onto a 2d manifold for the "Real Single-Cell Population Dynamics" experiment? Isn't it possible to be done on a higher dimensional space, e.g. gene expression of PCA space?
- The method involves multiple neural networks (v, g, s) and complex loss functions, which could require substantial computational resources as well as necessary tuning efforts, potentially limiting its practical application in real-time analysis. Would be great to get a comment on this.

---

> ### Author Response · Authors · 2024-11-20
> **Response to Reviewer FkRe (1/3):**
>
> > Summary:
> The paper introduces DeepRUOT, a new deep-learning approach for solving regularized unbalanced optimal transport (RUOT) and inferring continuous unbalanced stochastic dynamics from observed snapshots. The method models dynamics without requiring prior knowledge of growth and death processes, allowing these to be learned directly from data through a Fisher regularization formulation. The authors explore theoretical connections between RUOT and the Schrödinger bridge problem while reformulating RUOT to transform stochastic differential equations into more computationally tractable ordinary differential equations (a constraint). When tested on both synthetic gene regulatory networks and real single-cell RNA sequencing data, DeepRUOT outperformed existing methods in constructing meaningful developmental landscapes.
>
> > Strengths:
> The experimental results seem very convincing
> The paper provides thorough theoretical foundations with formal theorems and proofs connecting RUOT to the Schrödinger bridge problem.
>
>
> - Thank you for your careful reading and insightful suggestions. Thank you for your clear summary and positive comments.  Trajectory inference is indeed a pivotal area in both machine learning and single-cell biology. In our study, we address the challenge of learning stochastic dynamics from snapshots while accounting for unbalanced effects—an aspect that is critically important in both single-cell analysis and broader machine learning applications. We believe DeepRUOT offers a flexible tool for modeling and inferring continuous unbalanced stochastic dynamics from observed snapshots without requiring prior knowledge. Your recognition of the importance and innovation of our work is greatly appreciated.
>
> - Below you will find our detailed point-by-point responses. We have revised our manuscript accordingly to address the comments, and all changes made to the manuscript are indicated in **blue** throughout the revised main text and appendix.  Thank you once again for your insightful review and encouraging feedback.
>
>
>
> > Weaknesses:
> it remains unclear how the loss function weights (cost, reconstruction error and the PINN loss) are set
> Figure 1 and its caption are not that helpful
>
>
>
> - Thank you for your valuable suggestions. To address your concerns,  **we have expanded the discussion on loss function weighting** (Appendix C.2, Page 24, Line 1265-1280; Table 5, Page 25, Line 1296-1316). Specifically, we provide detailed values (Table 5, Page 25, Line 1296-1316) and the rationale behind the selection of these weights (Page 24, Line 1270-1276). During different training stages, we employ varying parameters to optimize performance; however, we have observed that these parameters exhibit robustness across diverse datasets, thereby minimizing the need for extensive adjustments.
>
> - Additionally, **we have revised Figure 1 to include a more comprehensive description of the loss components** (Figure 1, Page 6, Line 294-308). Additionally, we have incorporated the corresponding partial differential equations (PDEs) that govern the control mechanisms within our algorithm. These enhancements aim to provide a better understanding of our methodological framework.
>
>
>
> >Typos:
> Theorem 3.1: "probelm" -> "problem"
> page 8: "a increase" -> "an increase"
>
> - Thank you for your careful observations. **We have corrected “probelm” to “problem” in Theorem 3.1 (Page 3, Line 138) and changed “an increase” to “an increase” on page 8 in the revised version (Page 8, Line 392-393).** We have conducted a comprehensive review of the manuscript to identify and rectify all other typographical errors.

---

> > ### Author Response · Authors · 2024-11-20
> > **Response to Reviewer FkRe (2/3)**
> >
> > > Questions:
> > is not knowing the death/growth rates (what you call prior knowledge) such a big advantage?
> >
> > - Thank you for your insightful question. Indeed, the ability to infer stochastic dynamics without predefined growth or death rates constitutes a significant advantage in trajectory inference. In the context of single-cell analysis, cellular proliferation and apoptosis are fundamental biological processes, particularly during early developmental stages where rapid cell division leads to substantial increases in cell numbers. Accurately modeling these growth and death processes is inherently challenging due to their complexity and the difficulty of specifying appropriate functions  $g(\boldsymbol{x},t)$ a priori. Existing models often rely on detailed prior information such as counts of certain proliferation-related genes, which is not always available nor quantitatively accurate.
> > - For instance, as demonstrated in our synthetic gene regulatory network **(Section 6 & Figure 2, Page 8, Line 383-429)**, if the function $g(\boldsymbol{x},t)$ is incorrectly specified, it can lead to false dynamics, thereby compromising the validity of the inferred trajectories. This underscores the critical importance of accurately modeling growth and death processes. By allowing the model to learn these functions from the data, DeepRUOT mitigates the risk of introducing inaccuracies due to incorrect assumptions, thereby enhancing the reliability and robustness of trajectory inference in complex biological systems.
> > - **We believe that this flexible modeling approach significantly advances the field of trajectory inference, particularly in scenarios where prior knowledge of growth and death rates is limited or unavailable.**
> >
> > > Why did you project the data onto a 2d manifold for the "Real Single-Cell Population Dynamics" experiment? Isn't it possible to be done on a higher dimensional space, e.g. gene expression of PCA space?
> >
> > - Thank you for your insightful question. In our “Real Single-Cell Population Dynamics” experiment, we initially projected the data onto a 2D manifold to maintain consistency with the methods used by Sha et al. (2024). However, recognizing the benefits of higher-dimensional analysis, **we have expanded our study to include a new time-series scRNA-seq dataset from an A549 cancer cell line exposed to TGFB1 to induce EMT** (Page 9, Line 481-485; Appendix B.4 & Figure 8, Page 21-22, Line 1130-1186). In this extended analysis, we first reduced the gene expression data to 10 principal components using PCA, which served as the input for our algorithm. Subsequently, we visualized the results in the first and second principal component spaces. Our findings indicate that DeepRUOT remains effective and applicable in higher-dimensional settings, demonstrating its versatility and robustness in modeling complex single-cell dynamics without being limited to 2D projections.
> >
> > > The method involves multiple neural networks (v, g, s) and complex loss functions, which could require substantial computational resources as well as necessary tuning efforts, potentially limiting its practical application in real-time analysis. Would be great to get a comment on this.
> >
> > - Thank you for the insightful comments. For the tuning efforts, as previously mentioned, during different training stages, we employ varying parameters to optimize the performance of loss weighting. Importantly, we have observed that these parameters exhibit robustness across diverse datasets, thereby minimizing the necessity for extensive adjustments. This robustness ensures that our method remains user-friendly and does not require significant tuning efforts for different applications. **We have elaborated on this aspect in the revised manuscript** (Appendix C.2, Page 24, Line 1265-1280; Table 5, Page 25, Line 1296-1316).
> >
> > - Regarding computational cost, our approach is comparable to existing frameworks such as TrajectoryNet (Tong et al., 2020), MIOflow (Huguet et al., 2022), and Koshizuka et al. (2023), which involve simulating ordinary or stochastic differential equations (ODEs/SDEs) and performing optimal transport for distribution matching or integration along trajectories. Notably, the score matching in our method is conducted via conditional flow matching (Lipman et al., 2022; Tong et al., 2023a, 2023b), which is simulation-free and thus highly efficient. Consequently, the incorporation of multiple neural networks and loss functions in our approach does not introduce significant additional computational overhead since the cost of each component is similar to previous works. This ensures that our method remains both scalable and efficient. **We have commented on these aspects in the revised manuscript** (Page 19, Line 1006-1017).

---

> > > ### Author Response · Authors · 2024-11-20
> > > **Response to Reviewer FkRe (3/3)**
> > >
> > > References
> > >
> > > - Sha, Y., Qiu, Y., Zhou, P., & Nie, Q. (2024). Reconstructing growth and dynamic trajectories from single-cell transcriptomics data. Nature Machine Intelligence, 6(1), 25-39.
> > >
> > >
> > > - Tong, A., Huang, J., Wolf, G., Van Dijk, D., & Krishnaswamy, S. (2020, November). Trajectorynet: A dynamic optimal transport network for modeling cellular dynamics. In International conference on machine learning (pp. 9526-9536). PMLR.
> > >
> > > - Huguet, G., Magruder, D. S., Tong, A., Fasina, O., Kuchroo, M., Wolf, G., & Krishnaswamy, S. (2022). Manifold interpolating optimal-transport flows for trajectory inference. Advances in neural information processing systems, 35, 29705-29718.
> > >
> > > - Koshizuka, T., & Sato (2023), I. Neural Lagrangian Schrödinger Bridge: Diffusion Modeling for Population Dynamics. In The Eleventh International Conference on Learning Representations.
> > >
> > > - Lipman, Y., Chen, R. T., Ben-Hamu, H., Nickel, M., & Le, M. (2022). Flow matching for generative modeling. arXiv preprint arXiv:2210.02747.
> > >
> > > - Tong, A., Fatras, K., Malkin, N., Huguet, G., Zhang, Y., Rector-Brooks, J., ... & Bengio, Y. (2023a). Improving and generalizing flow-based generative models with minibatch optimal transport. arXiv preprint arXiv:2302.00482.
> > >
> > > - Tong, A., Malkin, N., Fatras, K., Atanackovic, L., Zhang, Y., Huguet, G., ... & Bengio, Y. (2023b). Simulation-free schrödinger bridges via score and flow matching. arXiv preprint arXiv:2307.03672.
> > >
> > > ---

---

> ### Author Response · Authors · 2024-11-22
>
> Dear Reviewer FkRe,
>
> Thank you very much for your valuable comments and insightful suggestions on our manuscript. We are also excited to receive your encouraging feedback. We have carefully addressed the concerns raised in your review and revised our manuscript substantially. We have uploaded the revised manuscript, and all changes made to the manuscript are indicated in blue throughout the revised main text and appendix.  If there are any remaining issues or questions you would like us to address before the rebuttal deadline, please do not hesitate to let us know. We are more than happy to take the valuable opportunity to discuss our work with you further or make any further adjustments as needed.
>
> Thank you again for your time and positive feedback.
>
> Yours sincerely,
>
> The Authors

---

> ### Comment · Reviewer_FkRe · 2024-12-03
> **Official Comment by the Reviewer**
>
> Thank you for the comments in the rebuttal.
>
> I increased my rating to 10 and the contribution to 4. I went through the paper again, and I am convinced that the theoretical and especially the empirical results are very convincing (!) and are strong compared to other methods I am aware of. Based on my experience, finding a tool that would produce similar results as observed in your Figure 3 is hard.
> Therefore, I think this is an important contribution to the computational biology community, and I would consider highlighting this work during the conference.

---

> ### Author Response · Authors · 2024-12-03
> **Thank you!**
>
> Dear Reviewer FkRe，
>
> Thank you very much for your encouraging and positive feedback! Thank you very much for raising the score! We are truly grateful for your recognition of the significance of our work and your appreciation of our results.
>
> Your positive assessment is immensely motivating for us. We are thrilled by the prospect of applying our algorithm to a broader range of applications within computational systems biology and single-cell omics. Furthermore, we are also excited about exploring its potential to extend beyond the biological sciences into wider machine-learning applications.
>
> Thank you again for your generous support, insightful feedback, and the time you have spent reviewing our manuscript.  Your feedback is invaluable to us and plays a crucial role in enhancing the quality of our work.
>
> Yours sincerely,
>
> The Authors

---

### Author Response · Authors · 2024-11-13
**Quick and Initial Response**

We would like to express our sincere gratitude to the Area Chairs and all the reviewers for their careful reading and thorough evaluation of our manuscript.  We are also very grateful for the clear summaries and positive comments about our work.

While all reviewers have positively acknowledged our work or recognized the great potential of our algorithm, they have also raised some concerns. Specifically, some reviewers have pointed out issues related to the clarity of our presentation, while others have recommended further empirical studies in comparison to existing unbalanced dynamic Optimal Transport (OT) methods and for a more comprehensive empirical understanding of the various components involved in training and loss weighting.

We find these suggestions highly insightful and believe that addressing them will significantly enhance the clarity and overall quality of our manuscript. Importantly, we consider these concerns to be technical rather than fundamental, and we think we will address them effectively. Accordingly, we are committed to undertaking substantial revisions of our paper. We will conduct new experiments and provide more detailed explanations to support our findings.

We will prepare a point-by-point response to address each of the reviewers’ concerns comprehensively and make corresponding improvements to the manuscript. We kindly ask for your patience and understanding as we work diligently to implement these enhancements. Thank you for taking the time to read our paper and for providing invaluable feedback. We hope that our forthcoming revisions and responses will meet your expectations and warrant a reevaluation of our manuscript for acceptance.

---

### Author Response · Authors · 2024-11-21
**Response to Reviewers (Revised Version)**

Dear Reviewer,

Thank you very much for your time and effort, as well as for your valuable comments and insightful suggestions on our manuscript. We have carefully addressed each of your comments in our revised manuscript and have provided a point-by-point response to address each of your concerns comprehensively. We hope that the revisions meet your expectations and that you find our manuscript suitable for acceptance. We would greatly appreciate it if you could reevaluate our paper based on the revisions and consider updating your evaluation accordingly. Should you have any further comments, we are more than happy to continue the discussion and make further revisions if needed.

Thank you for your consideration.

Best regards,

The Authors

---

### Meta-Review · Area_Chair_WAd2 · 2024-12-22

**Metareview:**

This paper uses deep learning to learn stochastic dynamics from discretely observed data. They connect the RUOT problem to Schrödinger bridge, which reviewers found novel and in particular comes alongside a careful treatment of unbalanced effects. The methodology is novel and is also motivated by a compelling application to single-cell RNA data. This is a strong submission with several novel contributions

**Additional Comments On Reviewer Discussion:**

Good engagement during response period

---

### Decision · Program_Chairs · 2025-01-22

Accept (Oral)